

# Turbulent dissipation from AMAZOMIX off the Amazon shelf along

# internal tides paths

Fabius Kouogang[1,4], Ariane Koch-Larrouy[1], Jorge Magalhaes[2], Alex Costa da Silva[4], Daphne Kerhervé[1], Arnaud Bertrand[5], Evan Cervelli[3], Jean-François Ternon[5], Pierre Rousselot[6], James Lee[7], Marcelo Rollnic[7], Moacyr Araujo[4]

[1]LEGOS, Université de Toulouse, CNRS, OMP, IRD, Toulouse, France

[2]Department of Geoscience, Environment and Spatial Planning (DGAOT), Faculty of Sciences, University of Porto, Porto, Portugal

[3]Rockland Scientific Inc, Lunenburg, Nova Escócia, Canadá

[4]Departamento de Oceanografia, Universidade Federal de Pernambuco, DOCEAN/UFPE, Recife, Brazil

[5]MARBEC, Université de Montpellier, CNRS, Ifremer, IRD, Sète, France

[6]IMAGO, Université de Bretagne Occidentale, CNRS, Ifremer, IRD, Brest, France

[7]Departamento de Oceanografia, Universidade Federal do Para, UFPA, Belem, Brazil

*Correspondence to*: Fabius Kouogang (fabius.cedric@yahoo.fr)

**Abstract.**

The Amazon shelf-break is a key region of the ocean where strong internal tides (ITs) are generated, which may have a key role to play on both Climate and Ecosystem, via its vertical mixing. AMAZOMIX survey (2021) collected microstructure and hydrographic (ADCP/CTD-$O_2$) profiles to quantify mixing, associated processes and their impact on marine ecosystems. Measurements are obtained over M2 tidal period (12h) inside and outside of both the ITs generation sites and propagation beams, respectively at mode-1 distances (90km and 210km) from the shelf-break to evaluate the IT impact on mixing.

Hydrography analysis showed strong step-like characteristics (~20-40 m thick) and vertical displacements (20-60 m) triggered by ITs, as well as the signatures of high modes up to 5-6 on generation sites and IT pathways.

The results of the microstructure analysis coupled with those of the hydrography revealed important mixing associated with a competition of processes between the semidiurnal shear of ITs and the baroclinic shear of the mean current (BC). Closer to the generation sites, mixing is stronger within $[10^{-6}, 10^{-4}]$ W.kg$^{-1}$, with a greater contribution (~65 %) from ITs shear than BC shear. It is reduced but nevertheless considerable between $[10^{-8}, 10^{-6}]$ W.kg$^{-1}$ along the IT pathways, owing to equal contributions from ITs and BC shear. At a distance of ~225 km, mixing was still higher within $[10^{-7}, 10^{-6}]$ W.kg$^{-1}$ because of the increased contribution (~65 %) of ITs shear, where IT beams may intersect and interact with background circulation. Mixing in no-tidal fields was fairly minimal ($[10^{-8}, 10^{-7}]$ W.kg$^{-1}$), owing to a minor contribution (~50.4 %) of BC shear from the North Brazil Current.



Finally, the nutrient flux estimations showed that ITs mixing could reach the surface (by a large tidal diffusivity of $[10^{-4}, 10^{-1}]$
$m^{-2}.s^{-1}$). This resulted in high vertical fluxes of nitrate ($[10^{-2}, 10^{-0}]$ mmol N $m^{-2}.s^{-1}$) and phosphate ($[10^{-3}, 10^{-1}]$ mmol P $m^{-2}.s^{-1}$),
which can stimulate chlorophyll production, biodiversity and cool surface water, so influencing the whole ecosystem and
climate in this river-ocean continuum region. This study provides a guide for the mixing parameterization in future numerical
simulation (e.g., in physical-biogeochemical coupled models) in the Amazon region in order to include the impact of the IT
turbulence on the whole ecosystem (i.e., from physics to biological production).

**1 Introduction**

Turbulent mixing in the ocean plays an important role for sustaining the thermohaline and meridional overturning circulation
(Kunze, 2017) and for closing the global ocean energy budget. These processes have strong implications for the climate,
through the influence on heat and carbon transport, and nutrient supply for photosynthesis (Huthnance, 1995; Munk and
Wunsch, 1998). Mixing processes can result from wind in the surface layer of the ocean, internal waves and shear instability
in the ocean interior, and bottom friction close to the bottom layer (Miles, 1961; Thorpe, 2018; Ivey et al., 2020; Inall et al.,
2021). Barotropic tides interacting with steep shelf-break topography trigger internal waves at tidal frequency and harmonics
called internal tides (ITs) that may propagate and generate mixing. These ITs induce large vertical displacements of water
masses up to tens of metres. After their generation at shelf-break, ITs higher-modes (more unstable) can dissipate locally
whereas ITs lower-modes may propagate far from generation sites. They can dissipate where the energy beam reflects at the
bottom, at the surface or at the thermocline levels (Bordois, 2015; Zhao et al. 2016) or where energy flux interferes with each
other (Zhao et al. 2012). ITs can also dissipate when they interact with strong baroclinic eddies or currents (Rainville and
Pinkel, 2006; Whalen et al., 2012). Furthermore, ITs may disintegrate in packets of higher-mode nonlinear internal solitary
waves (ISWs) that can propagate and dissipate offshore (Jackson et al., 2012).
Previous and recent studies have reported that ITs-induced turbulent mixing can have impacts on the surface, such as on sea
surface temperature (Ray and Susanto, 2016; Nugroho et al. 2018; Assene et al., 2024), chlorophyll content (M'Hamdi et al.,
2024; in preparation) and marine ecosystems (Wang et al. 2007; Muacho et al., 2014; Zaron et al. 2023), as well as on the
atmospheric convection and the rainfall structure (Koch-Larrouy et al., 2010, Sprintall et al. 2014).
In the western tropical Atlantic, the Amazon River-Ocean Continuum (AROC) constitutes a key region of the global oceanic
and climate system (Araujo et al., 2017; Varona et al., 2018). This region is characterised by the presence of a system of
western boundary currents including North Brazil Current (NBC). NBC flowing northwestward has its core velocities ($\sim 1.2$
$m.s^{-1}$) stable from the surface to a depth of 100 m (Johns et al., 1998; Bourlès et al., 1999; Barnier et al., 2001; Neto and Silva,
2014). There is also a system of Amazonian Lenses of water (AWL) induced by continental inputs, which can influence both
boundary layer and mixed layer patterns (Silva et al., 2005; Prestes et al., 2018).
In the AROC region, the Amazon shelf-break is a hotspot for generation, propagation and dissipation of ITs and ISWs as a
result of non-linear processes (Geyer, 1995; Magalhães et al., 2016; Ruault et al., 2020; Tchilibou et al., 2022; Figure 1). Using



SAR images, previous studies (Magalhaes et al., 2016) identified ISWs along the path of ITs propagating from two sites (i.e.,
sites Aa and Ab; Fig. 1a). Conversely, other sites did not have any ISW propagation (i.e., sites F and D; Fig. 1a and 1b) (see
Magalhaes et al., 2016 for definition). Using numerical modeling, Tchilibou et al. (2022) showed that about 30% of the M2
(dominant tidal component, Le Bars et al. 2010) ITs energy is dissipated locally (for higher-modes ITs) at sites F, Aa, Ab and
D (Fig. 1a), and that the remaining lower-modes ITs energy can be dissipated remotely. Dissipation away from the generation
sites (F, Aa, Ab and D; Fig. 1a) can result from the shear instabilities that are due to processes of ITs-ITs and/or ITs-
eddy/current interactions. Despite the presence of ITs, no direct measurements of dissipation rates have been conducted. In
addition, the impacts of IT dissipation on vertical nutrient fluxes in the AROC region are still unexplored.
The mixing induced by these internal waves in the region was observed during the AMAZOMIX cruise (Bertrand et al., 2021).
It has been designed to have stations/transects inside and outside ITs fields (Fig. 1a) and to measure ITs dissipation and study
their impact on the AROC ecosystem. Direct microstructure measurements of temperature, salinity and velocity were
conducted at the different repeated stations/transects over one M2 tidal cycle (~12h). In this study, we will quantify mixing,
identify the associated processes, and investigate their impact on nutrient fluxes off the Amazon shelf. We will calculate
turbulent kinetic energy (TKE) dissipation rates and vertical eddy diffusivities using in situ microstructure and hydrography
data. The baroclinic shear of currents will then be calculated from current data collected between stations and transects. Finally,
we will use vertical diffusivities estimations to determine vertical nutrient fluxes at the base of the mixed layer.



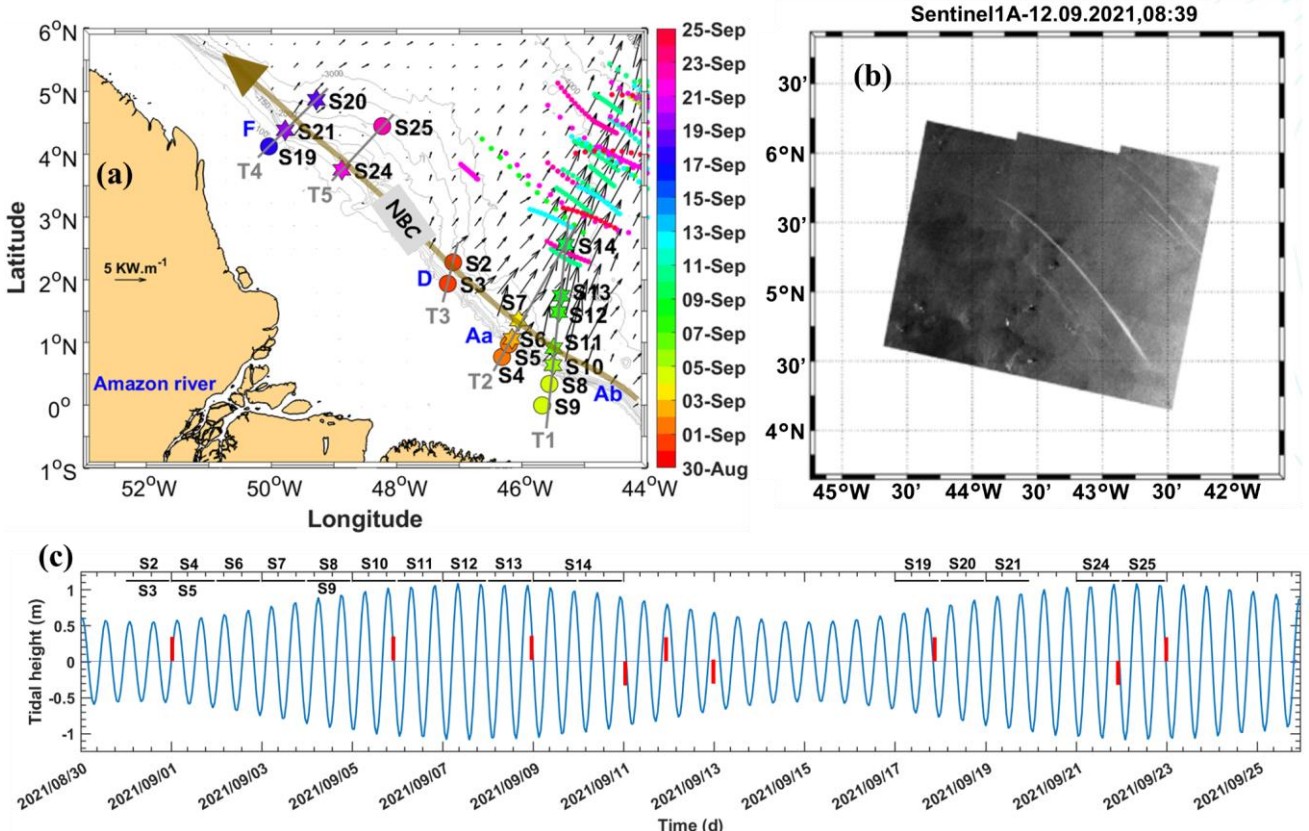


*Figure 1: a) Map of part of the AMAZOMIX 2021 cruise with bathymetric grey lines (100 m, 750 m, 2000 m, 3000 m and 4000*
*m isobaths) off the Amazon shelf. Colored circles and stars represent short and long CTD-O₂/L-S-ADCP stations casts,*
*respectively. Colored bar represents the time. Solid grey lines represent SADCP transects (T1 to T5). Black arrows represent*
*25h-mean depth-integrated baroclinic energy flux (on September 2015 from NEMO model) from ITs generation sites (Aa, Ab,*
*D and F) along the critical slope (grey contours) of the shelf-break. Solid brown line indicates the background circulation*
*with NBC pathways. Shattered colored lines show ISW signatures. T1 to T4 and T5 represent the inside-ITs and outside-ITs*
*transects, respectively. b) 1A Sentinel image acquired 12th September 2021 showing ISW signatures. c) Tidal range at*
*AMAZOMIX stations with ISW signatures dates in red bars.*

## 2 Data and Methods

### 2.1 Data collection

The AMAZOMIX cruise (Bertrand et al., 2021) was performed on the shelf/slope areas off the AROC during August-October
2021, on board the IRD vessel RV *ANTEA*. 12h-long stations were conducted with repeated (between 4 and 5 for each site)
casts of Conductivity-Temperature-Depth-Oxygen (CTD-O₂)/Lowered Acoustic Doppler Current Profiler (LADCP) and



Velocity Microstructure Profiler (VMP) in previously defined sites to measure the TKE dissipation rates over one complete
tidal cycle, and extract the tidal (M2) contribution from the total current. A high-resolution NEMOv3.6 model (1/36°) (Nucleus
for European Modeling of the Ocean; Madec et al., 2019) that provides realistic maps of ITs generation and propagation (Fig.
1a; Tchilibou et al., 2022; Assene et al., 2024) was used to estimate the position of 12h stations and estimate the mean
background stratification at these stations. Stations (Fig. 1a and 1c) were located inside the ITs fields, named "IN-ITs" (sites
Aa, Ab and D: S2 to S14; site F: S19 to S21), and outside the ITs fields (S24 and S25), named "OUT-ITs", on the shelf-break
generation (sites Aa, Ab, D and F) and propagation along 5 transects (T1 to T5, Fig. 1) including stations.
CTD-$O_2$ measurements were obtained using a Seabird 911 Plus with dual sensors mounted in the rosette equipped with 11
Niskin bottles used to sample water down to a depth of 1000 m. Concentrations of nutrients, including nitrate and phosphate,
were determined from nutrient samples taken in 30 ml Nalgene bottles and stabilised in an oven at 80 °C for 2h30. The 24 Hz
CTD-$O_2$ sensors were calibrated before and after the cruise. The standard deviation of temperature (salinity; oxygen) was
0.003 °C (0.003 PSU; 0.05 ml.l$^{-1}$) according to adjusted data. CTD-$O_2$ data were averaged over 1-m bins to filter out spikes
and missing points, and aligned in time to correct the lag effects. Two LADCPs RDI 300 kHz were mounted on the rosette,
one looking down and other one looking up, to provide vertical currents profiles with a 8 m resolution. In addition, 75 kHz
ship-ADCP (SADCP) profiles (with vertical resolution adjusted to the bottom depth: 8 m at S6, S7, S10-S14, S20, S21 and
S24, for bottom depth > 150 m; and 4 m at the rest) were continuously recorded during the cruise. All measured data were
processed and quality controlled according to the standard protocols of the GO-SHIP Repeat Hydrography Manual. A total
number of 71 CTD-$O_2$/LADCP profiles were acquired during the AMAZOMIX cruise.
To characterize mixing, the TKE microstructure profiles were obtained from high frequency (resolution: ~2 mm)
measurements of temperature and velocity shear using a VMP-250 profiler (Rockland Scientific International, Inc.) operating
at depth range of 1000 m.  The instrument has two high-resolution thermistors (FP07) and two high-resolution velocity shear
probes (probe 1 and 2, with accuracy of 5 % of the total signal) with their sampling frequency of 1024 Hz. The VMP-250
profiler was deployed and recovered with use of an electric winch and rope tether. The VMP was alternatively deployed
between the CTD-$O_2$/LADCP profiles at 33 stations for a total of 202 profiles. Only 18 stations (S2 to S14, S19 to S21, S24
and S25) for a total of 109 VMP profiles and 54 CTD-$O_2$/LADCP profiles will be processed and used in this study.
**2.2 Methods**
**TKE dissipation rates**
VMP data were processed using ODAS Matlab library (developed by Rockland Scientific International, Inc) in order to
determine TKE dissipation rate (ε). The VMP processing methods are briefly explained here, and conform to the
recommendations of ATOMIX (Analyzing ocean turbulence observations to quantify mixing) group (reported in Lueck et al.,
2024) and tested against the benchmark estimates (presented in Fer et al., 2024). First, the VMP data are converted into physical
shear units. The time series are prepared, and sections (continuous parts of the time series) within VMP data are chosen for
dissipation estimation. Before spectral estimation, the aberrant shear caused by vessel wake contamination is eliminated from



the timeseries. Shear probe collisions (with plankton and other matter) are removed from the shear signals using the de-spiking
routine. The record from each section is high-pass filtered. Shear spectra are estimated using record lengths (diss_length) and
Fast Fourier Transform segments (fft_length) of 2s that are cosine windowed and overlapped (overlap_sec) by 50%. Vibration-
coherent noise is removed. Different diss_length and overlap_sec were chosen and tested based on the environment (deep and
shallow water) following Fer et al. (2024). After testing, the diss_length (overlap_sec) was shortened to 4s (2s) for shallow
stations rather than 8s (4s) for deep stations, because of the evidence of the presence of overturns from AMAZOMIX acoustic
measurements in deep water stations (Koch-Larrouy et al., 2024; in preparation) and to optimize the spatial resolution of
dissipation estimates in shallow water stations. Finally, ε is determined using the spectral integration method and by
comparison with the Nasmyth empirical spectrum (Nasmyth, 1970).
The estimation of ε used high pass filtered data. Fig. 2 shows an example of the high pass filtered shear probes data at station
S6 (Fig. 2a). The shear data are qualitatively supported by both the observed profiles (Fig. 2c) and the low-pass filtered, along-
path gradients (Fig. 2b). Hence, in high turbulence regions, the temperature gradient variance is enhanced (bipolar), and the
temperature is reversed as a result of the overturns. In low turbulence regions, the temperature gradient variance is inverted
and background temperature tends to decrease with depth.

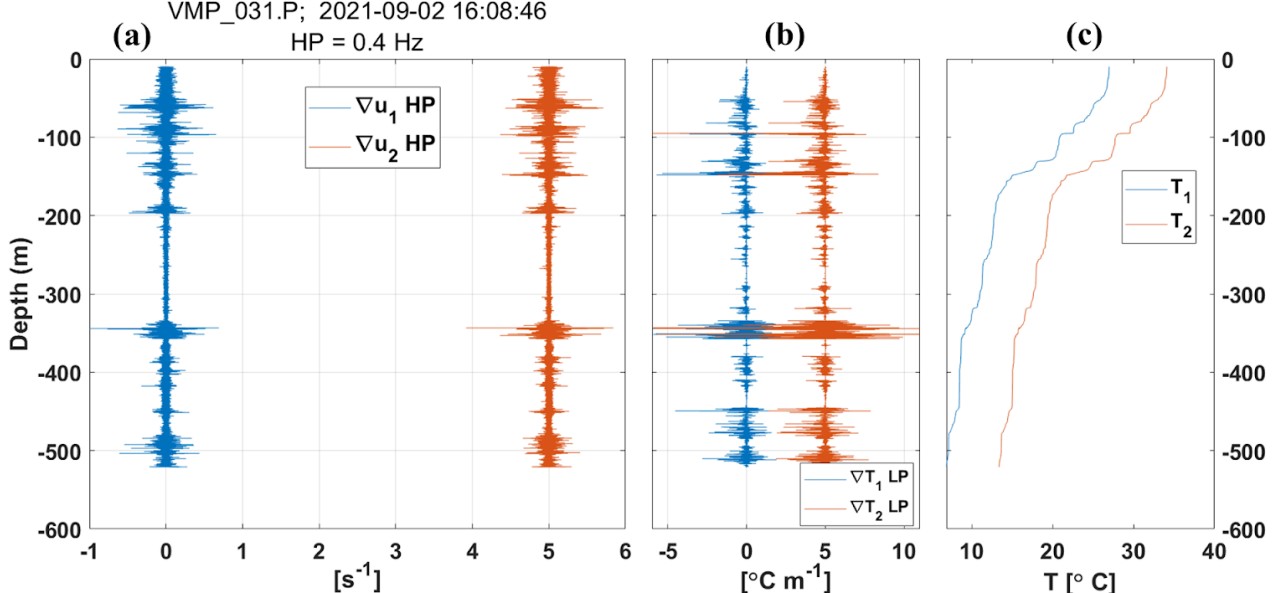

*Figure 2: Example of the VMP data at station S6. (a) Vertical profiles of data from shear probe 1 ($\nabla u_1$, blue), and from shear*
*probe 2 ($\nabla u_2$, red). HP indicates high-pass filtering of the shear probe data. (b) Vertical gradients of temperature from shear*
*probe 1 ($\nabla T_1$, blue) and probe 2 ($\nabla T_2$, red). LP indicates low-pass filtering on the along-path gradients. (c) Temperature*
*profiles $T_1$ (blue) and $T_2$ (red) based on nominal calibration coefficients.*



For each of the CTD-O$_2$ profiles, the Mixed Layer Depth (MLD) was computed using a density criterion of 0.03 kg.m$^{-3}$
difference (Montégut et al., 2004) with the surface density (with no major difference from density criterion of 0.01 and 0.02
kg.m$^{-3}$ by way of comparison) at AMAZOMIX stations/transects. Based in Lozovatsky et al (2006) and Sutherland et al (2014),
the miXed Layer Depth (XLD) is determined where ε falls to an assumed background level (first minimum value) in order to
capture the dissipation rate away from the influence of sea surface processes. The Upper (UTD) and Lower (LTD/LPD)
Thermocline/Pycnocline Depth was delimited as defined by Assunçao et al (2020). UTD corresponded to the depth where the
vertical temperature gradient $\partial\theta/\partial z = 0.1$ °C.m$^{-1}$, while LTD/LPD were the last depth below the UTD at which N$^2 \geq 10^{-4}$ s$^{-2}$.

**Vertical eddy diffusivity and turbulent flux of nutrients**
The vertical eddy diffusivity coefficient (K$_z$) is calculated from ε following Osborn (1980) formulation as defined by $K_z = \varepsilon\, \Gamma$
$N^{-2}$. Here N$^2$ is the buoyancy frequency squared was calculated using the sorted potential density profiles ($\sigma_\theta$) from CTD-O$_2$
profiles, via $N^2 = -\,(g/\rho_0)\,(d\sigma_\theta/dz)$ where $\rho_0$ is a reference density (1025 kg.m$^{-3}$) and g is the gravitational acceleration. $\Gamma$ is a
mixing efficiency defined as the ratio between the buoyancy flux and the energy dissipation, and is set to 0.2 which corresponds
to the critical Richardson number Ri = 0.17 based in Osborn (1980). ε was linearly interpolated into the depths of N$^2$.
The vertical turbulent flux F (unit: mmol.m$^{-2}$.s$^{-1}$) of nutrients (nitrate and phosphate) was estimated from K$_z$ as defined by:
$$F_{(NO_3^-,PO_4^{3-})} = -K_{(NO_3^-,PO_4^{3-})}\left(\frac{\partial C_{(NO_3^-,PO_4^{3-})}}{\partial Z}\right), \tag{1}$$

where $C_{(NO_3^-)}$ and $C_{(PO_4^{3-})}$ indicates the concentration of nitrate (NO$_3^-$) and phosphate (PO$_4^{3-}$) (unit: mmol.L$^{-1}$), respectively,
and $K_{(NO_3^-)}$ and $K_{(PO_4^{3-})}$ indicates its vertical diffusivity, respectively. Here we assume $K_{(NO_3^-,PO_4^{3-})}$ is equivalent to K$_z$. The
vertical profiles of nutrient concentrations (obtained from bottle sampling) were linearly interpolated into the depths of K$_z$.

**Baroclinic currents and energy**
To evaluate the processes that can explain the mixing measured, the baroclinic ($u'$, $v'$) components of horizontal velocity was
calculated ($u' = u - u_{bt}, v' = v - v_{bt}$) removing the barotropic ($u_{bt}, v_{bt}$) components of total horizontal current ($u, v$)
provided by SADCP time series with LADCP profiles glued below ~ 500 m of depth, as shown by the equations below.
$$[u',v'] = [u,v] - [u_{bt},v_{bt}], \tag{2}$$

$$[u_{bt},v_{bt}] = \frac{1}{H}\int_{-H}^{0}[u,v]\,dz \tag{3}$$

The M2 tidal component of baroclinic currents is evidenced by baroclinic (semi-diurnal) tidal velocity ($u''$, $v''$) by removing
($[u'',v''] = [u',v'] - [\overline{u'},\overline{v'}]$, with overbar the average over a tidal period) the baroclinic mean current ($\overline{u'},\overline{v'}$) profile, and
highlighted by baroclinic (semi-diurnal) tidal shear squared $S^{2''}$ ($S^{2''} = (\partial u''/\partial z)^2 + (\partial v''/\partial z)^2$) compared at the baroclinic mean
vertical shear squared $\overline{S^{2'}}$ ($\overline{S^{2'}} = (\partial\overline{u'}/\partial z)^2 + (\partial\overline{v'}/\partial z)^2$). The vertically sheared total baroclinic current can be converted to total
baroclinic energy following internal waves parameterization of dissipation rates (ε$_{MG}$) by Mackinnon and Gregg (2003) and
validated by Xie et al. (2013), as defined by:



$\varepsilon_{MG} = \varepsilon_0 \, (N/N_0) \, (S/S_0),$ (4)
with N the buoyancy frequency from CTD-O$_2$ profiles and S the vertical shear from baroclinic currents. $N_0=S_0=0.0052$ s$^{-1}$ and
$\varepsilon_0=2.2 \times 10^{-9}$ W.kg$^{-1}$ (adjustable constant). By simple substitution in this formulation, we computed the baroclinic tidal energy
($\varepsilon''_{MG}$) and baroclinic energy of mean circulation ($\overline{\varepsilon'}_{MG}$) in order to obtain the individual contribution ($\overline{\varepsilon'}_{MG}/(\overline{\varepsilon'}_{MG} + \varepsilon''_{MG})$ for
mean baroclinic energy and $\varepsilon''_{MG}/(\overline{\varepsilon'}_{MG} + \varepsilon''_{MG})$ for M2 semi-diurnal baroclinic energy) to the total baroclinic energy ($\overline{\varepsilon'}_{MG}$ +
$\varepsilon''_{MG}$).
To evaluate the bottom friction effect, the kinetic energy $\varepsilon_f = \frac{1}{2}\rho_s(u_f^2)$ close to the bottom boundary layer was computed using
friction velocity $u_f = u_b\sqrt{C_d}$. $C_d=2.5 \times 10^{-3}$ is the drag coefficient from a high-resolution NEMOv3.6 model (1/36°) (Nucleus
for European Modeling of the Ocean; Madec et al., 2019). Previous study of Huang et al. (2019) shown that the bottom
boundary thickness spatially varies between 15-123 m in Ocean Atlantic with median of about 30-40 m in the North Atlantic.
Bottom layer thicknesses were defined in our study area depending on the measured bathymetry from CTD-O$_2$ and near-bottom
currents from ADCP. $u_b$ is the total velocity over 15 m (40 m) thick above the seabed for shallow (deep) stations.

**Ray tracing calculation**
Analysing both the vertical profiles of the average currents and the spatial dimension along the IT pathways is another way to
better understand the mechanisms related to the measured mixing. The rays of ITs energy are generated at steep topography
regions (e.g., shelf-break) where ITs beams and the bottom slope match together (i.e. critical slopes) and then propagate within
the ocean interior. After seafloor reflection, these IT beams propagate upward and impinge on the seasonal pycnocline from
below (resulting in beam scattering) and create large ITs oscillations, which after steepening have been documented to
disintegrate into nonlinear ISWs - known as a "local generation" for the ISWs (New and Pingree, 1992). Theoretical analysis
of IT paths was computed using IT ray-tracing techniques previously used (New and Da Silva, 2002; Muacho et al., 2014), to
investigate the effectiveness and expected pathways of the IT beams off the Amazon shelf. One main hypothesis we made
using linear theory is to consider that the stratification is constant horizontally along the IT propagation path. Whereas in
reality it may vary, due to submesoscale and mesoscale variability. The IT ray tracing calculation assumes that in a
continuously stratified fluid, ITs energy can be described by characteristic pathways of beams (or rays) with a slope c to the
horizontal following:
$c = \pm \left(\frac{\sigma^2 - f^2}{N^2 - \sigma^2}\right)^{1/2},$ (5)
where σ is the M2 tidal frequency (1.4052x10$^{-4}$ rad.s$^{-1}$), and f is the Coriolis parameter. The $N^2$ from CTD-O$_2$ AMAZOMIX
data were first time-averaged to obtain the mean stratification at stations. Then, the monthly Amazon36 (2012-2016) $N^2$
obtained from NEMO model outputs (see Tchilibou et al., 2022; Assene et al., 2024, for model description) were smoothed
and stitched to the AMAZOMIX $N^2$ profiles below 1000m depth. IT ray-tracing diagrams were obtained along the 5 transects
(T1 to T5, Fig. 1a). Sensibility tests of IT rays with different seasons (August, September, October and April) were performed





varying the position of critical topography in order to get an envelope of ray paths consistent with characteristics of IT pathways.

## 3 Results

### 3.1 Mixing

#### 3.1.1 Thermohaline and IT features

In this section we analyze density profiles to gain information on the mixing and/or on waves propagation. First we observe step-like features in the density profiles (Fig. 3b and 3c). During the M2 tidal period, step-like structures of ∼20-40 m thick are found at depths ranging from 80-160 m at S10, S12, S13, and S14 (Fig. 3b). They are thicker along the IN-ITs transect T1 than along other transects (T2–T4) (Fig. A1.a and A1.b, Appendix). Then, between 60 and 170 m depth, large vertical displacements of 20 to 60 m are detected along T1-T2 and T4 (e.g., 40 m at S10, 48 m at S6, 52 m at S13, and 32 m at S14) (Fig. 3b and 3c). The smallest displacements (e.g., ~8 m at S25) are observed along OUT-ITs transect T5 (Fig. A1.b, Appendix). Finally, the vertical displacement of the IT wave may also be detected in the variability of the mixed layer depth (MLD), which ranges from 18 to 84 m over a semi-diurnal cycle (see Fig. 6.e). As a preliminary conclusion, both step-like structures and isopycnal displacements support the hypothesis of propagating ITs, that can have stronger energy along transects T1 and T2, weaker on T3, T4, and almost absent on T5 (Fig. 1a).



*Figure 3: (a) Horizontal (maximum) dissipation rates (ε, W.kg⁻¹, on log scale) from VMP during the AMAZOMIX 2021 cruise for all stations/transects T1 to T5. (b)-(c) Density profiles (in kg.m⁻³) from CTD-O₂ and (d)-(e) vertical dissipation rates (ε in W.kg⁻¹, on log scale) from VMP during the AMAZOMIX 2021 cruise for the transects/stations inside of the IT fields (b)-(d) T1 (S8-S14) and (c)-(e) T2 (S4-S7 and S14). For long stations (S6, S7 and S10-14), two density profiles are used to illustrate the step-like structures and isopycnal vertical displacements along the transects. Colour is used to distinguish each station in each transect. Dashed and solid back lines (on panels c and d) are for comparison. The density of S4 and S5 vary between 23.4-23.8 kg/m⁻³.*



234

### 3.1.2 TKE Dissipation rates and mixing

Now, we analyze the distribution of dissipation rates ($\varepsilon$) estimated below the mixing layer depth (hereinafter XLD), in order to characterize the mixing produced off the Amazon shelf. Note that the XLD (Table A1, Appendix) is generally deeper than the MLD.

Results show that from the continental shelf to the open sea $\varepsilon$ vary within the range of $[10^{-10}, 10^{-4}]$ W.kg$^{-1}$. Mapping the maximum value of $\varepsilon$ over the water column (Fig. 3a, 3d and 3e) reveals that the strongest $\varepsilon$ within $[10^{-6}, 10^{-4}]$ W.kg$^{-1}$ are observed on IN-ITs transects (T1-T3), and even larger $\varepsilon$ are found on the shelf-break at generation site of ITs (S3, S6 and S10). Smaller $\varepsilon$ values (between $[10^{-8}, 10^{-7}]$ W.kg$^{-1}$) are found away from the shelf-break (e.g., at S7, S11, S24 and S20), except at some deep-sea stations (e.g., at S14 and S25).

The vertical profile of $\varepsilon$ (Fig. 3d and 3e, and Fig. A1.c and A1.d, Appendix), show stronger $\varepsilon$ ($10^{-7}$-$10^{-6}$ W.kg$^{-1}$) in the thermocline layers (~120 m) for the stations close to the shelf-break and in the ITs influence (S6 and S10). Hotspots of mixing are found almost anywhere at S14 in the water column, for example, at 150, 300, 350, 500, 600 and 700 m depth. Finally for shelf stations in the ITs regions, S3 and S5, mixing increases close to the bottom.

As another preliminary conclusion, the distribution of $\varepsilon$ ranges by 2-3 orders of magnitude over depth, and indicates stronger mixing on the Amazon shelf and shelf-break compared with those located far from these areas, and even stronger in the regions of occurrences of ITs. In order to investigate the precise reasons for such heterogeneous distribution of $\varepsilon$ the aim of the next section will be to try to identify the different processes responsible for it, by looking at shear instability driven by the current measurements.

## 3.2 Processes contributing to mixing

In this section, we explore which processes among tides, general circulation, and friction are responsible for the high mixing activity observed off the Amazon shelf.

### 3.2.1 Baroclinic tidal current

The contribution of the ITs to the total baroclinic velocity structure is shown (Fig. 4a.1 to 4c.1, and Fig. A2.a.1 to A2.g.1, Appendix) by the temporal evolution of the baroclinic tidal current. The semi-diurnal M2 component of the baroclinic current is easily identified by alternating positive (red bands) and negative (blue bands) velocities along T1-T4. For the IN-ITs stations (Fig. 4a.1 to 4c.1, and Fig. A2.a.1 to A2.f.1, Appendix), M2 component signal is strong between depths of 100-300 m (e.g., at S10-S14) and 100-400 m (e.g., at S6) and 200-450 m (e.g., at S21). Whereas the signal is noisier at depth for OUT-ITs stations (e.g., at S24) along T5 (Fig. A2.g.1, Appendix). The baroclinic tidal velocity shows a superposition of several (03-05) tidal modes. A high number of modes are observed on the shelf-break (e.g., 04 modes at S6 and 05 modes at S10). While a low number of modes away from the shelf-break (e.g., 03 modes at S7, S12 and S14). The highest baroclinic tidal current velocities were observed (between 25-48 cm.s$^{-1}$) at sites Aa and Ab along T1-T2. Whereas lower tidal velocities (< 25 cm.s$^{-1}$) are found in site F along T4 (e.g., at S20 and S21) compared to OUT-ITs stations (e.g., at S24). The strong vertical shears (> $10^{-3}$ s$^{-2}$) of





the baroclinic tidal velocities are clearly localized in the pycnocline (between 80-180 m depth) for the IN-ITs stations (e.g., at S6, S10 and S12). While weak vertical tidal shears ($< 10^{-3}$ $s^{-2}$) are observed at greater depths. At these stations (e.g., S6, S10 and S14), the large vertical displacements of maximum $N^2$ were also encountered in pycnocline over tidal period. The dissipation rates ($\epsilon$) already presented in section 3.1.2, is also reported on Fig. 3. $\epsilon$ shown over the tidal period at long stations (e.g., at S6, S10 and S14) are found 2-3 orders of magnitude stronger in the pycnocline than at depth.

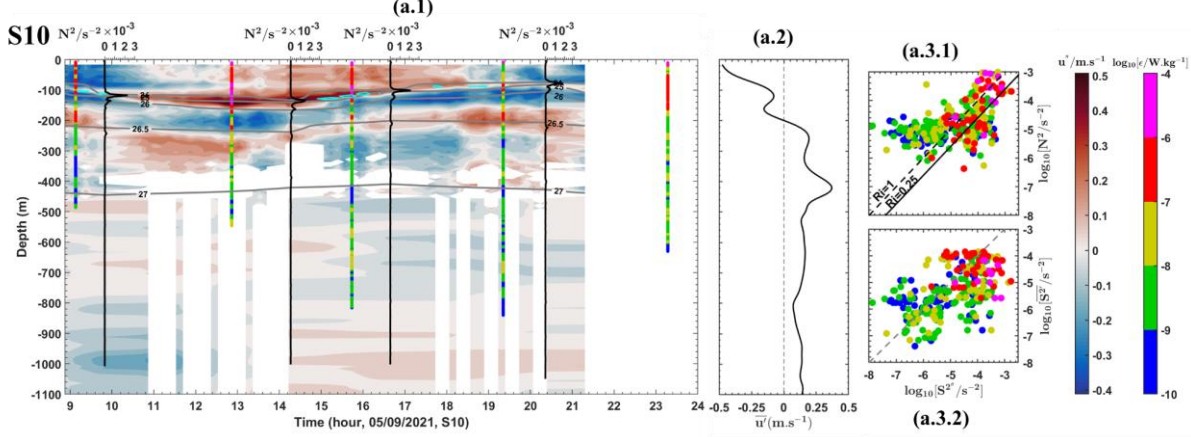

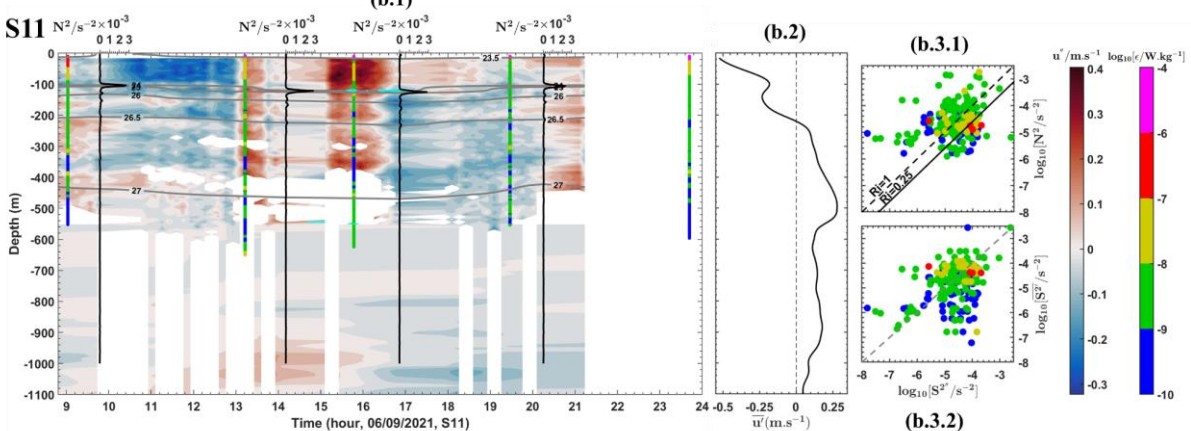





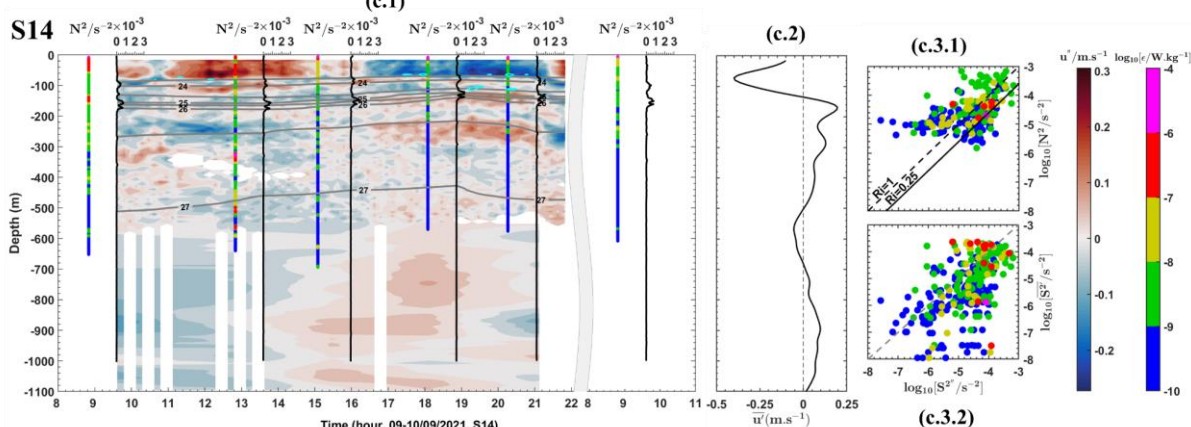

*Figure 4: (1st column: a.1 to c.1) Semi-diurnal (u'' in m.s$^{-1}$) baroclinic zonal currents from ADCP overlaid the semi-diurnal vertical shear squared (S$^{2''}$in s$^{-2}$ x10$^{-3}$ with the following contours values: 1.2 at S10, 0.6 at S11 and 0.8 at S14, in cyan contours) from ADCP, the buoyancy frequency squared (N$^2$ in s$^{-2}$, in vertical black lines) and potential density (in grey contours background shift) from CTD-O$_2$ and dissipation rates (ε in W.kg$^{-1}$ on log scale, in coloured bars) profiles from VMP. (2nd column: a.2 to c.2) Mean baroclinic (alongshore current) velocity ($\overline{u'}$ in m.s$^{-1}$) from ADCP. (3rd column: a.3.1 to c.3.1) ε as a function of S$^{2''}$ and N$^2$ overlaid Richardson number (with critical value Ri=0.25 in solid black line and Ri=1 in dashed black line). (3rd column: a.3.2 to c.3.2) ε as function of mean vertical shear squared ($\overline{S^{2'}}$ in s$^{-2}$) and S$^{2''}$ overlaid dashed grey line for comparison. (first line, panels a) for S10, (second line, panels b) for S11 and (third line, panels c) for S14. N$^2$ was linearly interpolated into the depths of S$^{2''}$ to have same vertical scales.*

### 3.2.2 Mean baroclinic current

Now, the contribution of the mean baroclinic current is diagnosed by the alongshore baroclinic velocities (taken parallel to the 200 m depth isobath) as a proxy of the main circulation pattern in the region. A surface alongshore northwestward flow crossing transects T1 and T2 (e.g., at S6, S7, and S10-S14) is observed stronger with mean (negative) velocities exceeding 40 cm.s$^{-1}$ in the upper layer of 120 m (Fig. 4a.2 to 4c.2, and Fig. A2.a.2 to A2.d.2, Appendix). It moves with decreasing velocities towards the subsurface layer. Below 130 m depth on these transects (T1 and T2), a flow with relatively low (positive) velocities (~ 25 cm.s$^{-1}$) is observed towards the southeast. In this layer, there are extremes of (positive) velocities (e.g., at S10, S11 and S13) and also direction reversals below 300 m (e.g., at S13 and S14). In contrast, a surface flow crossing T4 (Fig. A2.e.2 and A2.f.2, Appendix) is found moving southeastward in the first 100 meters depth. It shows mean (positive) velocities up to 35 cm.s$^{-1}$ (e.g., at S20) before decreasing in the subsurface layer. Below 100 m depth, a flow northwestward is observed on T4 and rapidly becomes unstable (e.g., at S21). In this layer, there is a reversal of direction and low flow velocities (< 15 cm.s$^{-1}$) towards the depths. For OUT-ITs T5 transect (e.g., at S24), a bidirectional flow is observed moving southeastward at the surface and northwestward in the subsurface layer with low (< 15 cm.s$^{-1}$) mean velocities (Fig. A2.g.2, Appendix).




### 3.2.3 Competitive processes to generate mixing


Our intention is now to clearly associate each mixing event to either tides or mean currents. For that purpose we will compare
for each mixing event the vertical shear induced by the baroclinic tidal current ($S^{2''}$) to those induced by the baroclinic mean
current ($\overline{S^{2'}}$). Practically, each ε was analyzed in ($N^2$, $S^{2''}$) and ($\overline{S^{2'}}$, $S^{2''}$) space. The hotspots of ε are observed where there is
important vertical shear instability. Indeed, along T1-T2 and T4 (Fig. 4a.3.2 to 4c.3.2, and Fig. A2.a.3.2 to A2.g.3.2,
Appendix), strong mixing between [$10^{-8}$,$10^{-5}$] W.kg-1 on the IN-ITs transects (e.g., at S6, S10, S14 and S21) are found where
$S^{2''}$ is stronger (e.g., up to 1.4x $10^{-3}$ s$^{-2}$ at S6, 1.2x $10^{-3}$ s$^{-2}$ at S10 and 0.8x $10^{-3}$ s$^{-2}$ at S14) than $\overline{S^{2'}}$. Tidal vertical shear was
large enough to cross the large stratification (with $N^2$ up to $10^{-3}$ s$^{-2}$ where Ri < 0.25) along T1 and T2 (Fig. 4a.3.1 to 4c.3.1,
and Fig. A2.a.3.1 to A2.d.3.1, Appendix). This was not true (with Ri > 1) where there are more ε between [$10^{-9}$,$10^{-8}$] W.kg$^{-1}$
(e.g., at S20, S21 and S24) along T4 and T5 (Fig. A2.e.3.1 to A2.g.3.1, Appendix). Others mixing events within [$10^{-9}$,$10^{-6}$]
W.kg$^{-1}$ were observed where $\overline{S^{2'}}$ is more relevant (up to $10^{-3}$ s$^{-2}$) than $S^{2''}$ or where there is low $S^{2''}$ shear (e.g., up to 0.4x$10^{-3}$
s$^{-2}$ at S21).
To better clarify which of the tidal vs mean vertical shear is dominating to explain the hotspots of mixing, we compare the
contribution of the semi-diurnal ($\varepsilon''_{MG}$) and mean ($\overline{\varepsilon'}_{MG}$) baroclinic energy to the total baroclinic energy, by simply transforming
the vertical shear into turbulent energy (Table A1, Appendix). On T1 and T2, the contribution of $\varepsilon''_{MG}$ was found dominant
(61.2 % at S6 and 65.94 % at S10) compared with that (38.8 % at S6 and 34.06 % at S10) of $\overline{\varepsilon'}_{MG}$ on the IT generation on the
shelf-break. The same is true far from the shelf-break for some IN-ITs stations on T1 (e.g., at S12, S13 and S14) and on T4
(e.g., at S20 and S21). But this is not the case for other IN-ITs stations (e.g., at S7 and S11) where the contribution of $\varepsilon''_{MG}$
suddenly drops, and for OUT-ITs stations (e.g., at S24) where the contribution (> 50 %) of $\overline{\varepsilon'}_{MG}$ dominates. However, the
contribution of $\varepsilon''_{MG}$ increases (from 51.85 % at S12 to 58.94 % at S14) along T1, while it decreases (from 59.13 % at S21 to
52.64 % at S20) along T4. The contribution to the baroclinic energy supports the hypothesis that mixing is dominated by ITs
on the IT generation sites on the shelf-break, by mean circulation far IT fields, and both by ITs and background circulation
away from the shelf-break.

### 3.2.4 IT ray tracing


Another way of looking at the two main processes that can explain the mixing measured is to look at the vertical profiles of
the mean total (alongshore) currents, with the spatial dimension along the transect of IT rays propagation. IT ray paths are
computed for the M2 tidal frequency and the rays for September are illustrated along the ITs-IN transects T1-T4 (Fig. 5a and
5b, and Fig. A3.a to A3.e, Appendix). IT rays are generated at the critical slope (between 32-104 km from the coast) on
Amazon shelf-break, then propagate downward into the deep ocean where they reflect for the first time (within 1250-3900 m
depth and at a distance between 54-222 km, not shown). After bottom reflection and eventual interaction with the pycnocline,
IT ray paths were observed reflecting at the surface seaward at a distance of about 115-400 km (e.g., at T1-T4). The curvature



of the IT rays is more pronounced when they reach the pycnocline depths delimited (between 93-207 m depth at T1-T4) by the upper (UTD) and lower (LTD) thermocline depth. Along T1 and T2 (Fig. 5a and 5b), alongshore flow towards northwest is found stronger at the surface with a maximum (negative) velocity up to 80 cm.s$^{-1}$ (e.g., at S11) above 150 m depth. This flow becomes unstable beyond 150-450 m depth. A flow instability was also observed along T4 (Fig. A3.e, Appendix). Large $\varepsilon$ are encountered where IT rays paths presumably interfere either between them or with the mean flow (Fig. 5a and 5b). Tracking IT rays along the transects (Fig. 5a and 5b, and Fig. A3.a to A3.e, Appendix), $\varepsilon$ are found larger (within $[10^{-8},10^{-5}]$ W.kg$^{-1}$) in the rays generation (e.g., at S3, S5 and S10) and propagation (e.g., at S6 and S21). Some large $\varepsilon$ are observed where IT rays radiated at the surface (e.g., at S12, S14 and S20) and below 300 m depth (at S14 along T1). These results indicate that turbulent dissipation occurred on the IT rays paths, and where rays interfere with each other and encounter the strong mean background circulation.

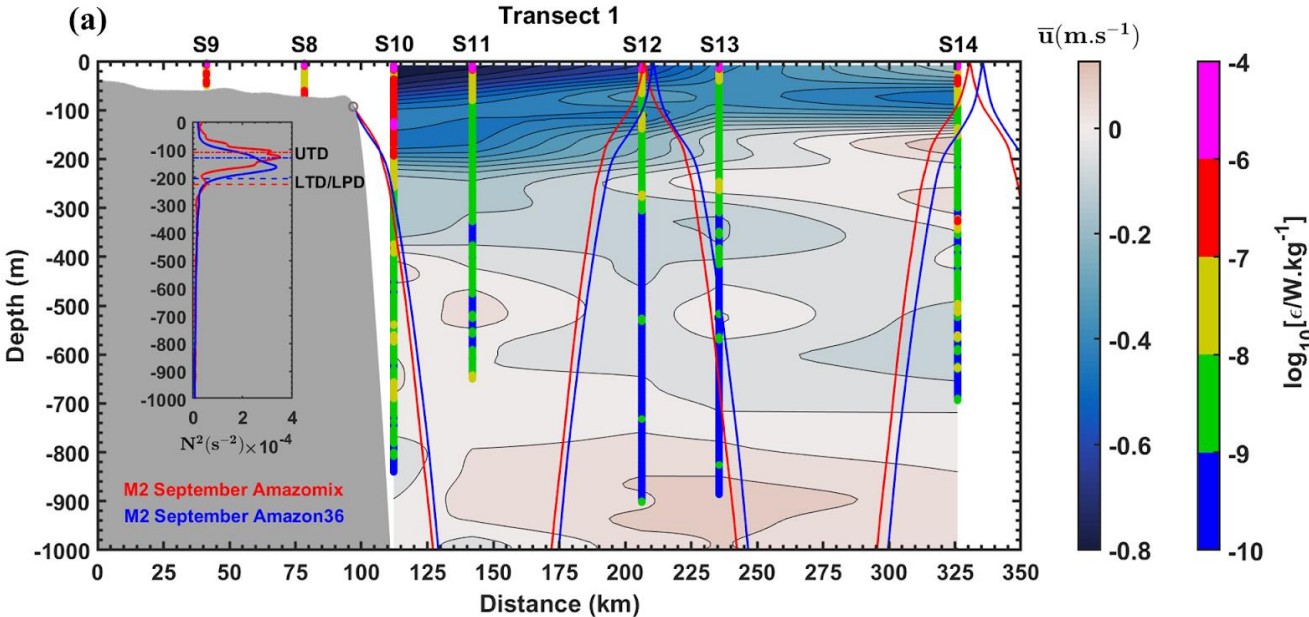



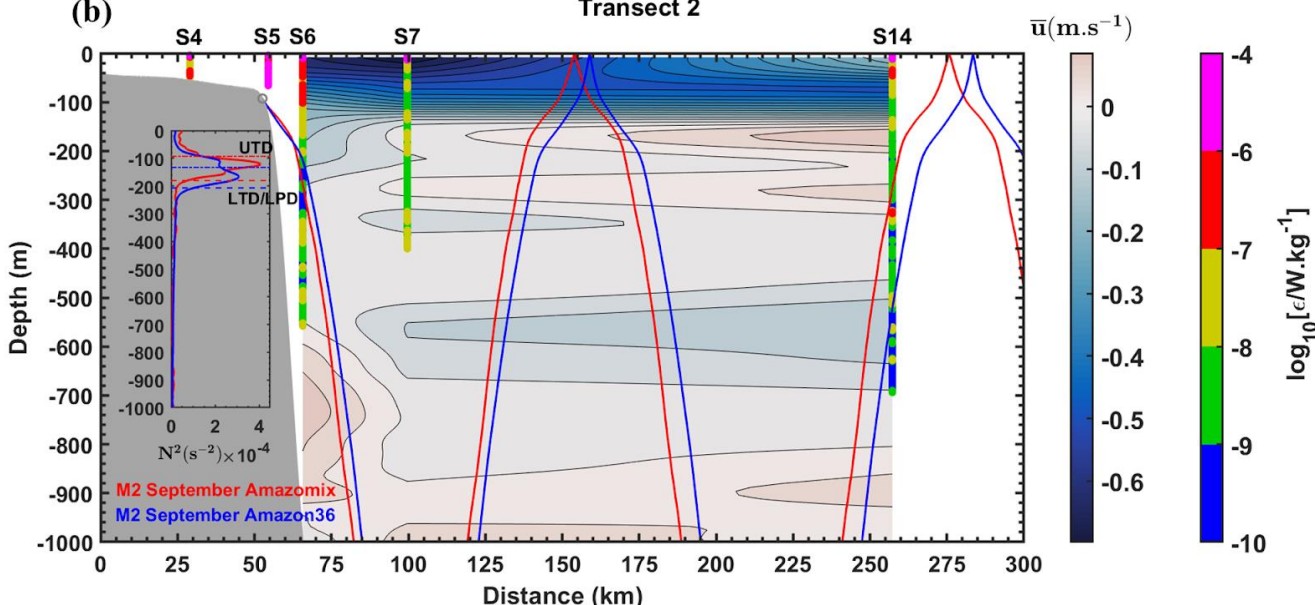

*Figure 5: Vertical sections showing: IT ray-tracing diagrams for the M2 tidal constituent assuming the mean (September) stratification (N² in s⁻², in overlaid panel) from AMAZOMIX CTD-O₂ (2021, in red line) and NEMO model (Amazon36, 2012-2016, in blue line), along the transects (a) T1 and (b) T2. Overlaid on the panels the profiles of dissipation rates (ε in W.kg⁻¹ on log scale, in coloured bars) from VMP, and the profiles of mean total (alongshore) current ($\bar{u}$ in m.s⁻¹, in background) for T1 and T2 from ADCP. UTD (in dotted lines) and LTD/LPD (in dashed lines) correspond to Upper and Lower Thermocline/Pycnocline Depth, respectively. Grey circles indicate the critical slope and grey areas represent local bathymetry.*

### 3.3 Vertical diffusivity and nutrients flux at the base of the mixed layer

The next steps of our study is dedicated to quantifying how much mixing could be responsible for nutrients uplift toward the euphotic layer, which may explains a boost in local primary production. Mixing coefficients were evaluated through vertical diffusivity ($K_z$). $K_z$ is first analysed in the water column (Fig. A4, Appendix), and then at the base of MLD in Fig. 6c-e for T1 and T2 (and on Fig. A6.c and A6.d for T3 to T5, Appendix). For IN-ITs stations on T1-T4 (Fig. A4, Appendix), $K_z$ is higher between $[10^{-2}, 10^{-1}]$ m².s⁻¹ in the mixed layers (e.g., at S10) and closer to the bottom layer. $K_z$ is found smaller but still important below 200 m depth, with levels ranging between $[10^{-4}, 10^{-3}]$ m².s⁻¹. For OUT-ITs stations on T5 (Fig. A4, Appendix), $K_z$ remains stronger up to $10^{-4}$ m².s⁻¹ below 200 m depth at S24, with two highest values exceeding $10^{-2}$ m².s⁻¹ below XLD at S25. Finally, at the base of MLD, strong $K_z$ is observed between $[10^{-3}, 10^{-0}]$ m².s⁻¹ on the Amazon shelf (e.g., at S5 and S9) and the generation sites (e.g., at S3, S6, and S10), as well as within $[10^{-4}, 10^{-3}]$ m².s⁻¹ along IT pathways. The question now is could it trigger a strong vertical flux of nutrients that may boost primary production? With simple consideration we calculate the



vertical flux of nutrients (nitrate and phosphate) at the base of MLD. Nitrate flux is analyzed in Fig. 6a-b for T1 and T2 (and in Fig. A6.a and A6.b for T3 to T5, Appendix).

Nitrate fluxes at the base of MLD are higher within $[10^{-2}, 10^{-0}]$ mmol N m$^{-2}$.s$^{-1}$ between 30-95 m depth along T1-T3 (e.g., at S2, S3, S5-S7, and S9-S11), and smaller but still large within $[10^{-3}, 10^{-2}]$ mmol N m$^{-2}$.s$^{-1}$ around 60-120 m depth on T4 (e.g., at S19-S21). Other stations inside (e.g., S4, S12-S14) and outside (e.g., S24 and S25) of IT fields reveal nitrate fluxes even smaller approximately $[10^{-5}, 10^{-3}]$ mmol N m$^{-2}$.s$^{-1}$ at the base of MLD.

Also, phosphate fluxes (Fig. A5, Appendix) at the base of MLD are stronger between $[10^{-3}, 10^{-1}]$ mmol P m$^{-2}$.s$^{-1}$ around 30-95 m depth along T1-T3 (e.g., at S2, S3, S4, S5, S7 and S10), and smaller but still significant within $[10^{-4}, 10^{-3}]$ mmol P m$^{-2}$.s$^{-1}$ between 60-120 m depth on T4 (e.g., at S19-S21). Phosphate fluxes along T4 are still smaller about $[10^{-6}, 10^{-4}]$ below the MLD and even exceeding $10^{-2}$ mmol P m$^{-2}$.s$^{-1}$ above the MLD at S25. Finally, phosphate fluxes are essentially similar in shape as nitrate fluxes, which is relevant because they both depend mainly on $K_z$ profiles (e.g., Fig. A6, Appendix). This indicates that significant vertical flux of nutrients was found where there was strong vertical diffusivity at the base of the mixed layers off the Amazon shelf.

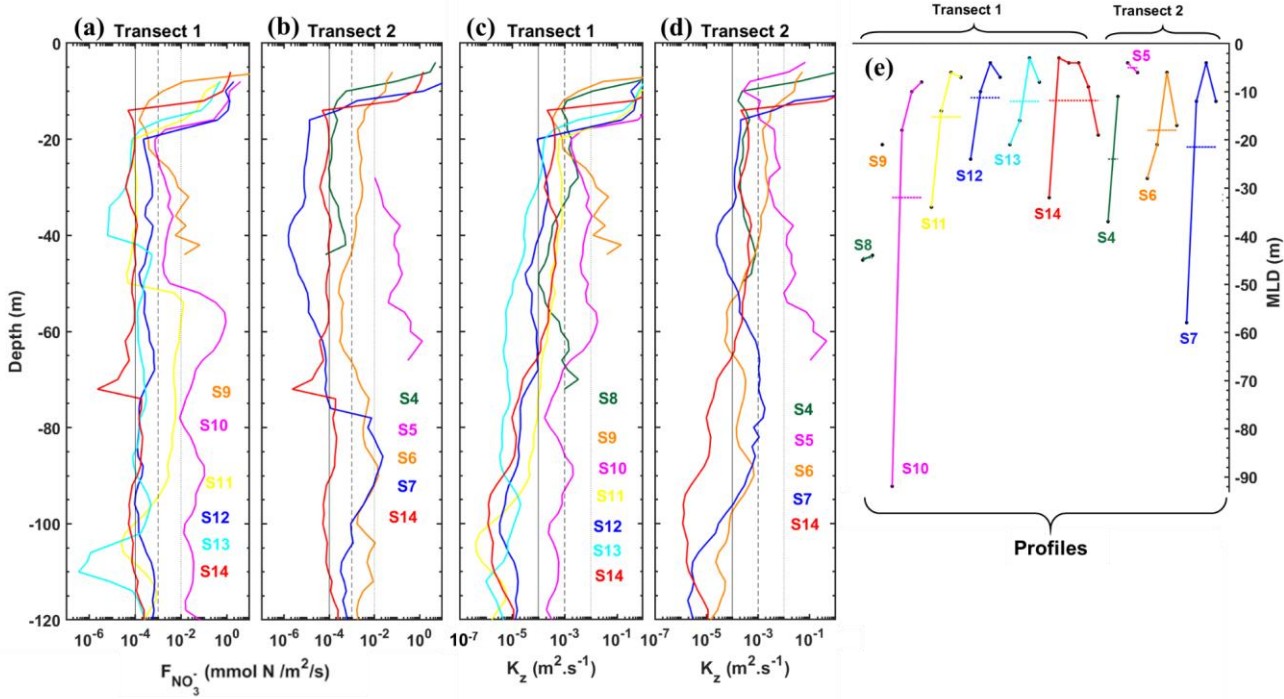

*Figure 6: (a)-(b) Vertical fluxes of nitrate (mmol N m$^{-2}$.s$^{-1}$) and (c)-(d) vertical diffusivity (in m$^2$.s$^{-1}$) at the base of MLD (defined at the MLD + 1m) during AMAZOMIX 2021 cruise for the transects/stations inside of the IT fields (a)-(c) T1 (S10-S14) and (b)-(d) T2 (S4-S7 and S14). Dashed and solid back lines are for comparison. (e) MLD (in m) for each CTD-O₂/LADCP (black dots), and average values (colored horizontal dashed lines) for each station. Colour is used to distinguish each station in each transect. Vertical flux of nitrate was null at S8 because of null nitrate concentration gradient.*




382

**4 Discussion and Conclusion**

AMAZOMIX 2021 cruise delivered for the first time direct measurements of turbulent dissipation using a velocity microstructure profiler VMP over several stations in and out of the IT influence that allow to study mixing in the Amazon Shelf break and open ocean facing it. To catch a tidal cycle, the measurements of turbulent dissipation rate, hydrography and currents, and also nutrient concentrations were collected alternately during 12h with 4 to 5 profiles for each station (see section 2). The position of 12h stations were chosen using modeling results that provide realistic maps of ITs generation and propagation (Fig. 1a; Tchilibou et al., 2022). Stations within IT influence were localized on the most energetics IT generation regions in sites Aa, Ab and D (S6, S10 and S3) as referenced by previous studies (Magalhaes et al., 2016; Tchilibou et al., 2022; Assene et al., 2024) and along the IT propagation paths (S8-S9 and S11-S14 for site Ab). Also, a less energetic ITs site was sampled, in site F (at S20 located in the generation and S21 in the offshore). In addition, other stations were localized on the specific points (S5, S9 for sites Aa and Ab and S19 for site F) where IT beams may also climb the shelf-break just after generation. Finally, as for stations on the shelf-break, AMAZOMIX cruise also sampled regions out of the IT fields (S24) and in open ocean (S25).

**Vertical Displacement, homogeneous layers**

The results showed that over a semi-diurnal tidal cycle, both large (up to 60 m length) isopycnal displacements and strong (up to 40 m thick) step-like structures were found along the transects T1 and T2, whereas they are smaller and thinner on T3, T4 than T5. This is probably related to the propagation of ITs that produces vertical displacements at tidal frequency and eventually mixing that creates homogeneous layers, identified on temperature and salinity structure as step-like features. We found stronger energy along T1 and T2, compared to T3 and T4, as previously described by modeling studies (Tchilibou et al., 2022; Assene et al., 2024). These step-like structures and isopycnal displacements in the pycnocline are also consistent with the observations over a tidal cycle in other IT regions (Stansfield et al., 2001; Simpson and Sharples, 2012; Bordois, 2015; Koch-Larrouy et al., 2015; Zhao et al., 2016; Bouruet-Aubertot et al., 2018; Xu et al., 2020).

**Direct measurements of dissipation rate**

Dissipation rates, obtained with the VMP, were stronger within $[10^{-6},10^{-4}]$ W.kg$^{-1}$ mainly on the generation sites Aa, Ab and D (at S6, S10 and S3) and were smaller but still large $[10^{-9},10^{-7}]$ W.kg$^{-1}$ a few kilometers (~40 km) from these sites (at S2, S11 and S7), along the IT path (at S11-S13 and S20) and far IT fields (e.g., at S24). They were still higher between $[10^{-8},10^{-6}]$ W.kg$^{-1}$ in the open ocean mainly ~225 km from generation site Ab (at S14), as resumed in Fig. 7.



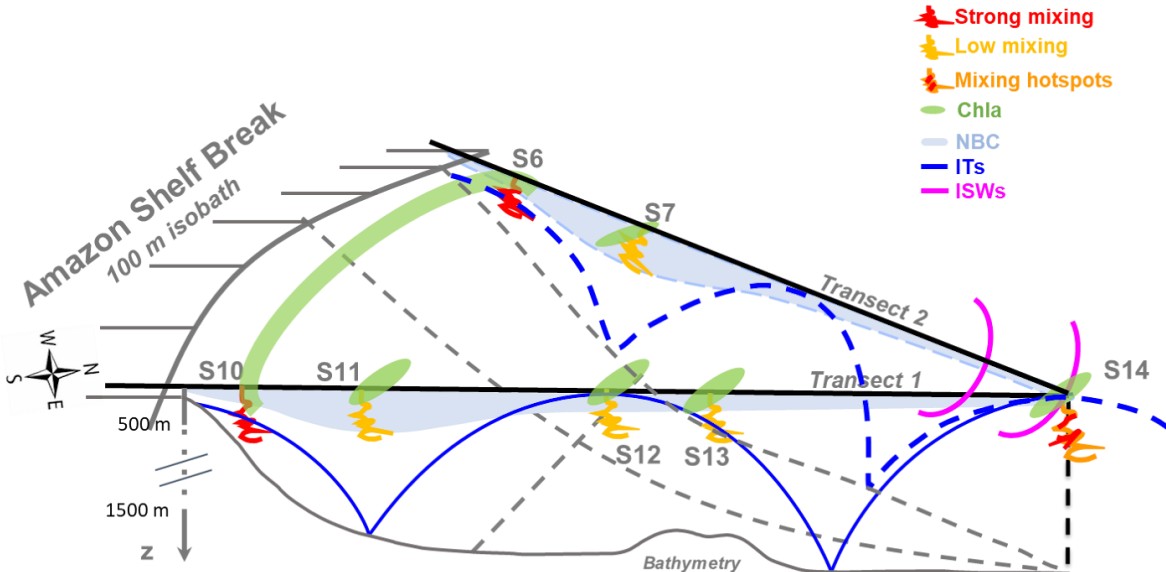


*Figure 7: Summary diagram showing the processes that lead to mixing along the AMAZOMIX transects (e.g., along T1 and*
*T2).*
In comparison, in other regions, dissipation rates measured by similar VMP instrument are found between $[10^{-7}, 10^{-5}]$ W.kg$^{-1}$
in the IT generation zone of Halmahera Sea, Indonesia (Koch-Larrouy et al., 2015; Bouruet-Aubertot et al., 2018), of Kaena
Ridge, Hawaii (Klymak et al., 2008) and off the Changjiang Estuary (Yang et al. 2020). Whereas it is $[10^{-10}, 10^{-8}]$ W.kg$^{-1}$ along
the IT path in the Southern Ocean (Gille et al., 2012) and in Halmahera Sea (Bouruet-Aubertot et al., 2018). Direct estimates
of dissipation far from IT influence are almost $[10^{-11}, 10^{-10}]$ W.kg$^{-1}$ (Koch-Larrouy et al., 2015; Bouruet-Aubertot et al., 2018)
or under the influence of geostrophic current (Takahashi and Hibiya, 2019).
Our study also found a very high dissipation rate for S3 and S5 of $[10^{-6}, 10^{-4}]$ W.kg$^{-1}$ on the Amazon shelf , increasing near the
bottom boundary layer. These findings compare well with values reaching $3\times10^{-9}$ W.kg$^{-1}$ within a kilometer of the seabed in
the Southern Ocean (Sheen et al., 2013) and up to $10^{-6}$ W.kg$^{-1}$ within a few meters from bottom topography off the Changjiang
Estuary (Yang et al. 2020). This may indicate the presence of an active bottom boundary layer. Thus, kinetic energy of bottom
flow was estimated using friction velocity, that was computed from total velocity averaged over the bottom-most 15 m for
shallow stations. It showed bottom fiction energy stronger 9-22 J.m$^{-2}$ at S3 and S5 mainly and lower (< 1 J.m$^{-2}$) in the other
stations (e.g., at S8) on shelf. These results are smaller but still important on the Amazon shelf and comparable to values (5-
17 kJ.m$^{-2}$) in the Drake Passage region (on the continental slope) of the Southern Ocean (Laurent et al., 2012). The bottom
mixing at S3 and S5 can indirectly exert a control on pycnocline mixing on the Amazon shelf (Inall et al., 2021).







**Contribution of Background circulation and ITs to mixing**

**Mean baroclinic current shear**
Another important issue raised in this study was the quantification of the contribution of each process that could explain this
heterogeneous mixing.
Firstly, we considered the mean baroclinic current (BC) as a proxy of the background circulation. BC was mainly structured
in northwestward surface and southeastward subsurface flow. The strong surface flow (e.g., at S11) toward northwest is
associated with NBC originating from the northeast coast of Brazil (e.g., Bourlès et al., 1999). The subsurface flow (e.g., at
S11) toward southeast could be associated with the instability of the NBC in depth (Dossa et al., 2024; in preparation). Both
baroclinic flows were unstable and produced vertical shear instabilities stronger $>10^{-4}$ s$^{-2}$ at 100-120 m for NBC and ~0.5 x
$10^{-4}$ s$^{-2}$ at 530-550 m for the subsurface flow at S7 and S11 mainly, while they were still larger ($>10^{-4}$ s$^{-2}$) above 80 m for NBC
at S6, S12-S14.
Other large shear instability of the mean current $> 2.5$ x $10^{-4}$ s$^{-2}$, located in NBC pattern at S20 could be associated with NBC
retroflection around 5-6°N during fall near 50°W (Didden and Schott, 1993). Finally, BC shear was ~0.8 x $10^{-4}$ s$^{-2}$ between
100-120 m at S21 and 40-60 m at S24 out the IT fields, and inside of the outer path of the Amazon plume. It is probably related
to the AWL from the continental inputs (Prestes et al., 2018) at S21 and S24, or due to the possible presence of subsurface
eddy at S21. This shows the potential of the mean flow to develop shear instability off the Amazon shelf.

**ITs shear**
Second, regarding IT, the baroclinic tidal currents, were separated from the total baroclinic currents, and showed significant
M2 semi-diurnal component signals, with high tidal modes 03-05 modes on the generations sites (S6 for Aa and S10 for Ab)
and on the IT path coming from sites Ab in S14), with strong vertical shear instabilities associated to it. These high modes
semi-diurnal baroclinic currents induced vertical tidal shears in the pycnocline layer (between 80-120 m). That was stronger
between 1.5-2 x $10^{-3}$ s$^{-2}$ in the generation areas (S6 and S10), still larger 1-1.5 x $10^{-3}$ s$^{-2}$ a few kilometers from generation sites
(S7 and S12) and smaller within 0.5-1 x $10^{-3}$ s$^{-2}$ in the open ocean (e.g., at S20). Finally, for the stations out of the IT fields,
IT shear was $<10^{-3}$ s$^{-2}$ at S24 in the water column, but higher than that close to the bottom topography probably due to the
active bottom boundary layer (Inall et al., 2021).

**IT/BC ratio**
Each of IT and BC shear contributes to mixing. The ratio changes from one site to another: in the vicinity of the ITs generation
sites on the shelf-break, ITs dominated this ratio with ~65%/~35% for IT/BC respectively (for S10). Also, it was about
~60%/~40% at S14 where IT beams from Aa and Ab encounter and may interfere.
Whereas along the A path, where NBC was strong at S11-S12, it was ~50%/~50% and ~55%/~45% at S13. This relative
diminution of IT influences compared to NBC may come from the fact that NBC was stronger at these stations.



Finally, for stations far from IT fields, BC dominated this ratio with ~49.6%/~50.4% at S24 due to both NBC and AWL.
These results are in good agreement with previous studies that showed strong tidal shear in the generation sites, such as in
Halmahera Sea (Bouruet-Aubertot et al., 2018), off the Changjiang Estuary (Yang et al. 2020), in the northwest European
continental shelf seas (Rippeth et al., 2005) or in the southern Yellow Sea (Xu et al., 2020).
The really new aspect raised in this study was the very large mixing found up to 225 km from generation sites, after the
reflection beams in S14 (Fig. 4), induced by strong tidal shears between $1$-$1.5 \times 10^{-3}$ s$^{-2}$ located above (60-80 m) and within
the pycnocline layer.

**Discussion on the strong mixing at S14**
Along IT paths, the higher remote dissipation rates (within $[10^{-7}, 10^{-6}]$ W.kg$^{-1}$) were found ~225 km from the shelf-break (S14).
Indeed, S14 experiences stronger mixing than S12 and S13. This region has been described by realistic models as a region of
strong dissipation of IT (Tchilibou et al., 2022; Assene et al. 2024). Also a recent satellite study, mapping ISWs generated
from ITs using MODIS images, showed that this region is also the region of higher occurrences of ISWs (de Macedo et al.,
2023). S14 is indeed localized where the ITs rays from Aa and Ab may intersect. It is also the region where the NBC vanishes.
Such change in the wave guide as well as the wave-wave interferences may generate the higher modes found. These higher
modes may in turn favor the generation of non-linear ISWs and higher dissipation rates found in this region. In addition, when
ITs disintegrate into a package of ISWs events (Jackson et al., 2012) they can lead to enhancing turbulent mixing (Xie et al

484 2013).

It should be noted that ITs interactions with baroclinic eddies could also lead to the turbulent dissipation (Booth and
Kamenkovich, 2008), in particular in this region of high eddy activities. However, no repeated AMAZOMIX stations
investigated here over a tidal period were observed to be enclosed by mesoscale eddy activity off the Amazon shelf, except
around S20-S21 where a possible subsurface eddy was detected.
Further study is necessary to better understand the complex interaction between all these processes, and AMAZOMIX data
will serve as a guide for future understanding and parameterization for modeling study.

**Role of mixing in nutrient fluxes off the Amazon shelf**
One last important aspect that we investigate in this study, is the impact of mixing on nutrient fluxes at the base of MLD.
Vertical fluxes of nitrate and phosphate were stronger around $[10^{-2}, 10^{-0}]$ mmol N m$^{-2}$.s$^{-1}$ and $[10^{-2}, 10^{-0}]$ mmol P m$^{-2}$.s$^{-1}$,
respectively, mainly on the generation sites Ab and D (e.g., at S10 and S3; Figure 6) where the vertical diffusivity was stronger
($[10^{-3}, 10^{-1}]$ m$^2$.s$^{-1}$). They were smaller but still large between $[10^{-3}, 10^{-2}]$ mmol N m$^{-2}$.s$^{-1}$ and $[10^{-4}, 10^{-3}]$ mmol P m$^{-2}$.s$^{-1}$ a few
kilometers (~40 km) from these sites (at S11 and S7). The nitrate and phosphate fluxes were still smaller within $[10^{-4}, 10^{-3}]$
mmol N m$^{-2}$.s$^{-1}$ and $<10^{-4}$ mmol P m$^{-2}$.s$^{-1}$ along the IT path (at S12-S14 and S20) and far IT fields (e.g., at S24). Finally, they
were even larger around $[10^{-3}, 10^{-0}]$ mmol N m$^{-2}$.s$^{-1}$ and $[10^{-3}, 10^{-0}]$ mmol P m$^{-2}$.s$^{-1}$ on the Amazon shelf (e.g., at S5) where the
vertical diffusivity was stronger ($[10^{-3}, 10^{-0}]$ m$^2$.s$^{-1}$).





Such diffusivity is consistent with the estimates of Koch-Larrouy et al. (2015) in the Indonesian seas, that also measured an
impact on nutrient fluxes and associated chlorophyll primary production (Zaron et al. 2023).
Tidal vertical diffusivity was also shown close enough to the surface to modify the heat content off the Amazon shelf (e.g.,
Assene et al. 2024). Due to the shallow euphotic zone in this region, the strong upward fluxes of nitrate and phosphate may
contribute in the surface layer to the local chlorophyll bloom off the Amazon shelf. The significant impact of the latter was
highlighted and shown: an increase in chlorophyll concentration (0.02-0.04 mg.m$^{-3}$) using simple calculation, from glider and
ocean color (M'Hamdi et al., 2024; in preparation), and from MODIS Terra sunglint images with and without ISWs events
(de Macedo et al., 2023) off the Amazon shelf. The increase (1-2 orders of magnitude) in nutrient fluxes confirms the theory
that mean supply of nutrients are often dominated by relevant IT mixing (Sharples and Zeldis, 2019; Kaneko et al., 2021; Yang
et al., 2020). This finding implies that, in the AMAZOMIX region, ITs act as an important supplier of nutrient into the euphotic
layer, able to maintain new primary production and to possibly impact the whole ecosystem.

**Appendices**

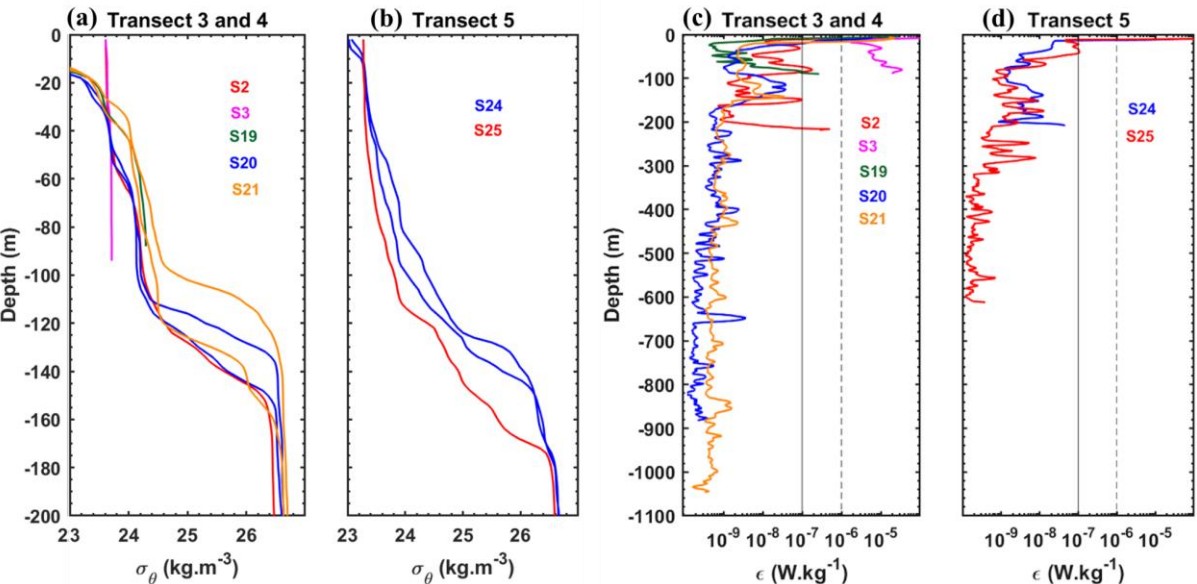


*Figure A1: (a)-(b) Potential density profiles (in kg.m$^{-3}$) from CTD-O$_2$ and (c)-(d) vertical dissipation rates ($\varepsilon$ in W.kg$^{-1}$, on log*
*scale) from VMP during the AMAZOMIX 2021 cruise for the transects/stations inside of the IT fields (a)-(c) T3 (S2 and S3)*
*and T4 (S19-S21), and far from IT fields (b)-(d) T5 (S24 and S25). For long stations (S20 and S21), two density profiles are*
*used to illustrate the isopycnal vertical displacements along the transects. The potential density at S3 varies between 23.6-*
*23.72 kg.$^{-3}$. Color of profiles is used to distinguish each station in each transect.*



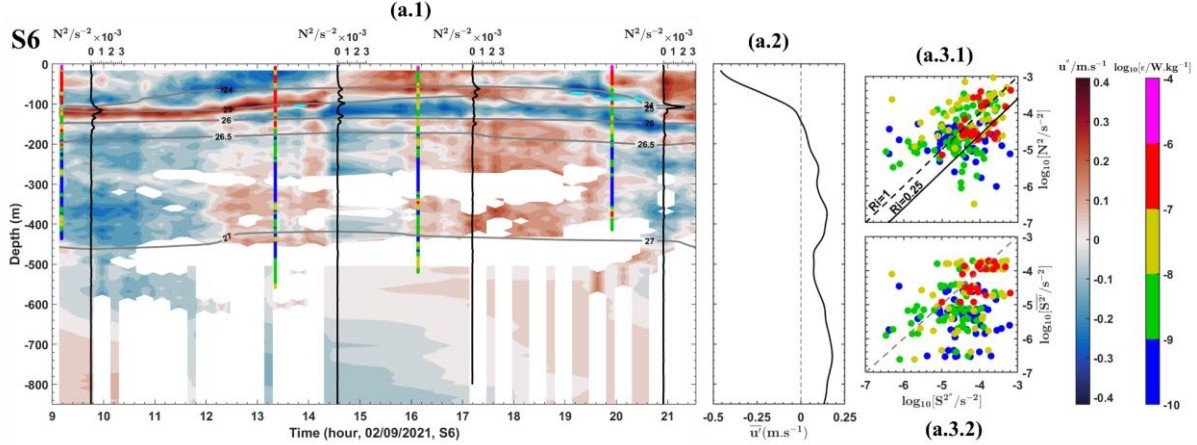


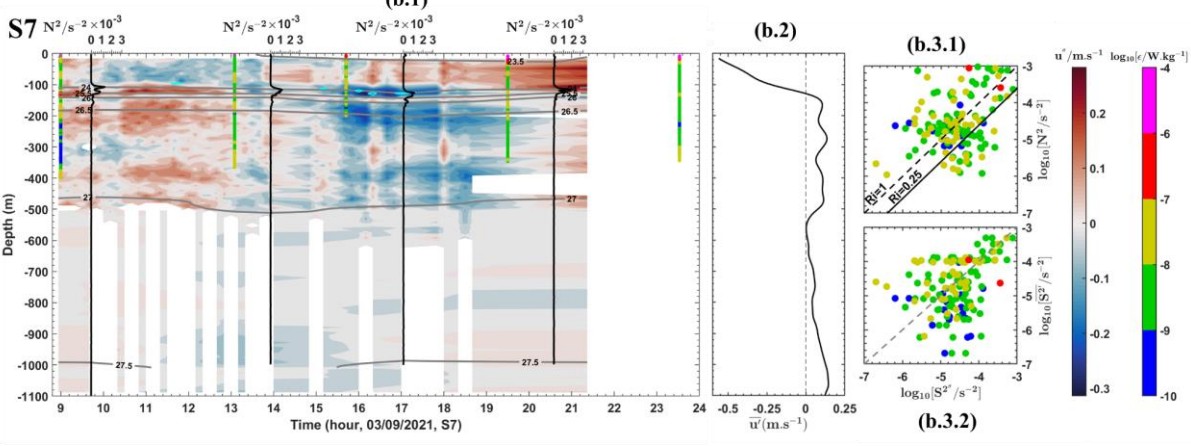


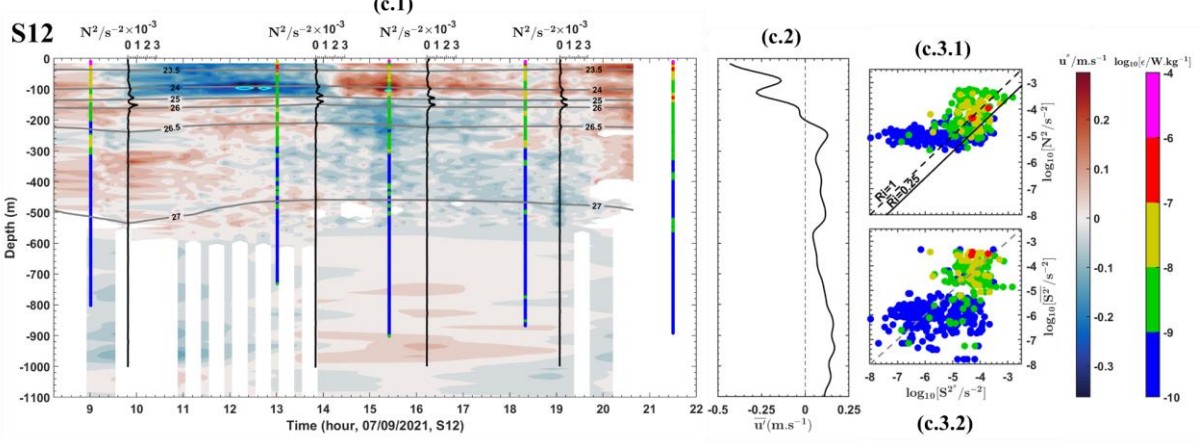




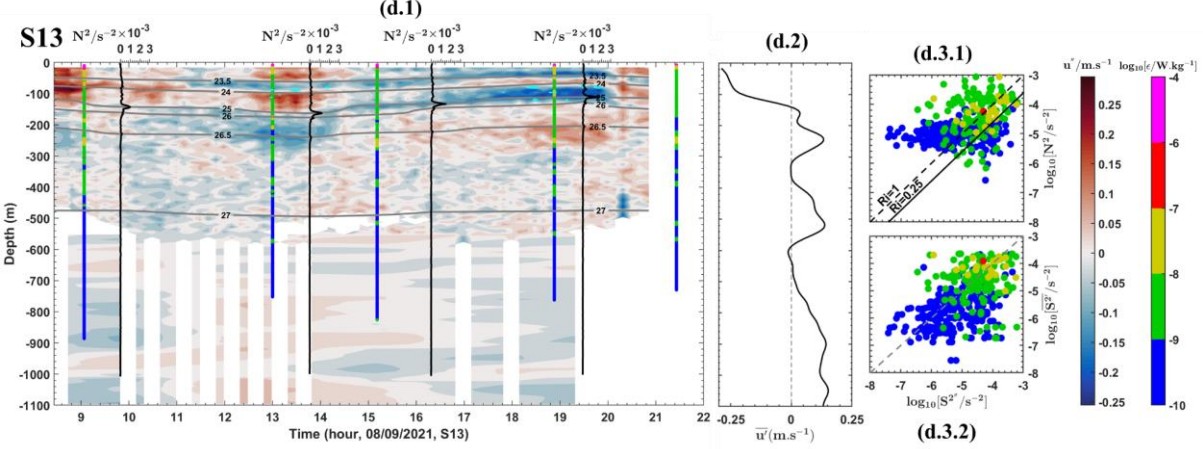


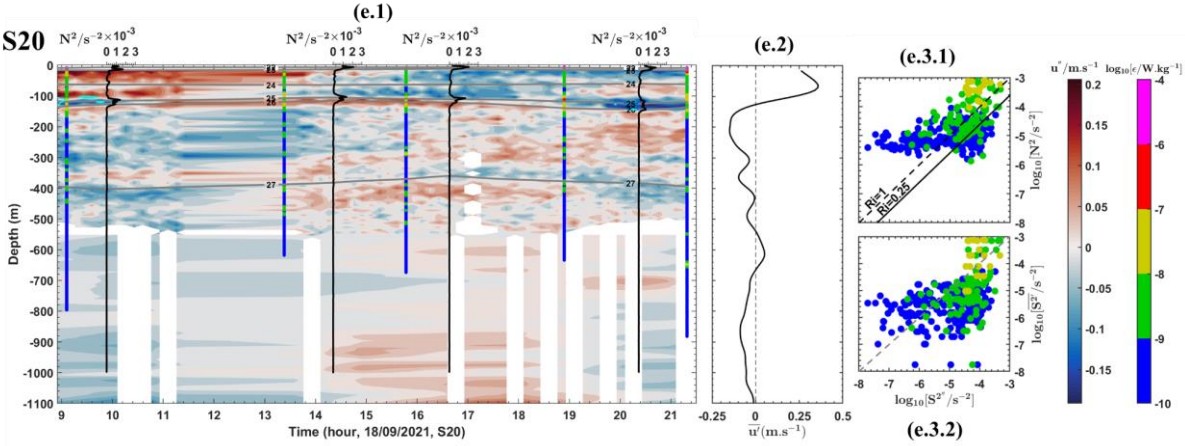


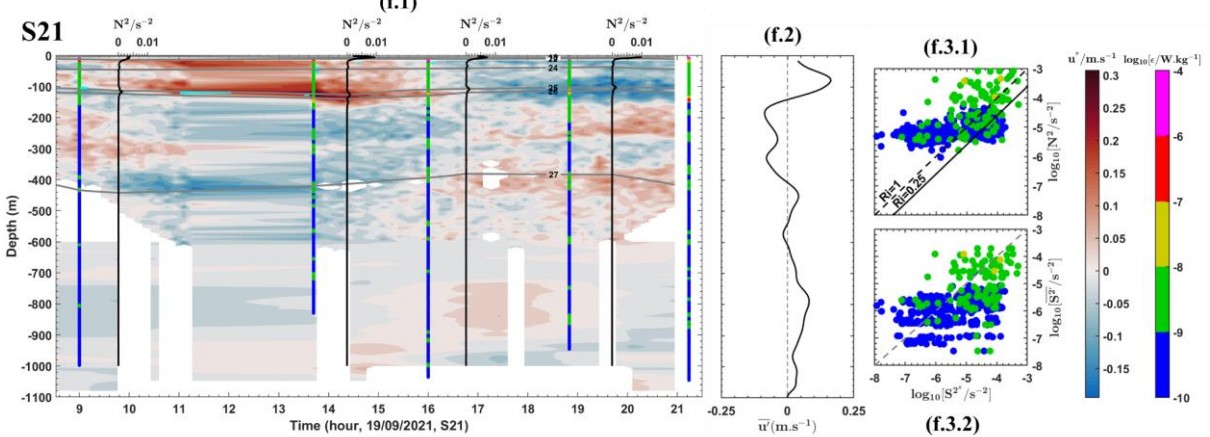


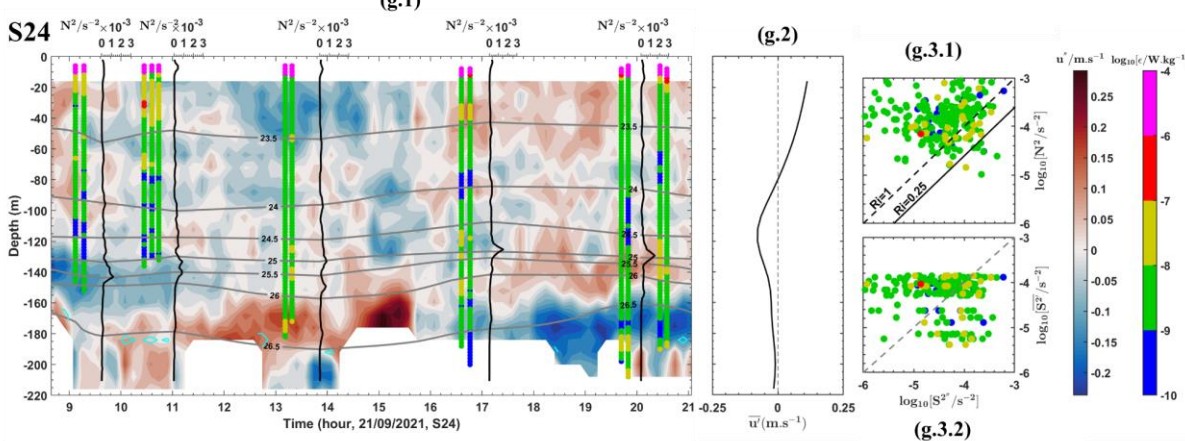

*Figure A2: (1st column: a.1 to g.1) Semi-diurnal ($u''$ in m.s$^{-1}$) baroclinic zonal currents from ADCP overlaid the semi-diurnal vertical shear squared ($S^{2''}$ in s$^{-2}$ x10$^{-3}$ with the following contours values: 1.4 at S6, 0.8 at S7, 1.1 at S12, 0.55 at S13, 0.5 at S20, 0.4 at S21 and 1.0 at S24, in cyan contours) from ADCP, the buoyancy frequency squared ($N^2$ in s$^{-2}$, in vertical black lines) and potential density (in grey contours background shift) from CTD-O$_2$ and dissipation rates ($\varepsilon$ in W.kg$^{-1}$ on log scale, in coloured bars) profiles from VMP. (2nd column: a.2 to g.2) Baroclinic mean (alongshore current) velocity ($\overline{u}'$ in m.s$^{-1}$) from ADCP. (3rd column: a.3.1 to g.3.1) $\varepsilon$ as a function of $S^{2''}$ and $N^2$ overlaid Richardson number (with critical value Ri = 0.25 in solid black line and Ri=1 in dashed black line). (3rd column: a.3.2 to g.3.2) $\varepsilon$ as function of mean vertical shear squared ($\overline{S^{2'}}$ in s$^{-2}$) and $S^{2''}$ overlaid dashed grey line for comparison. (first line, panels a) for S6, (second line, panels b) for S7, (third line, panels c) for S12, (fourth line, panels d) for S13, (fifth line, panels e) for S20, (sixth line, panels f) for S21 and (seventh line, panels g) for S24. $N^2$ was linearly interpolated into the depths of $S^{2''}$ to have same vertical scales.*



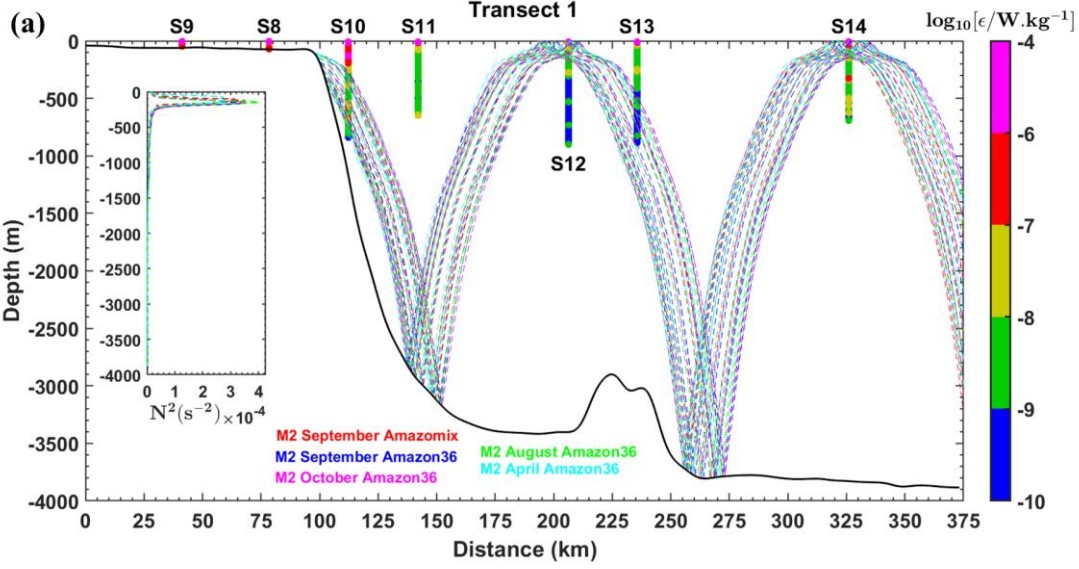


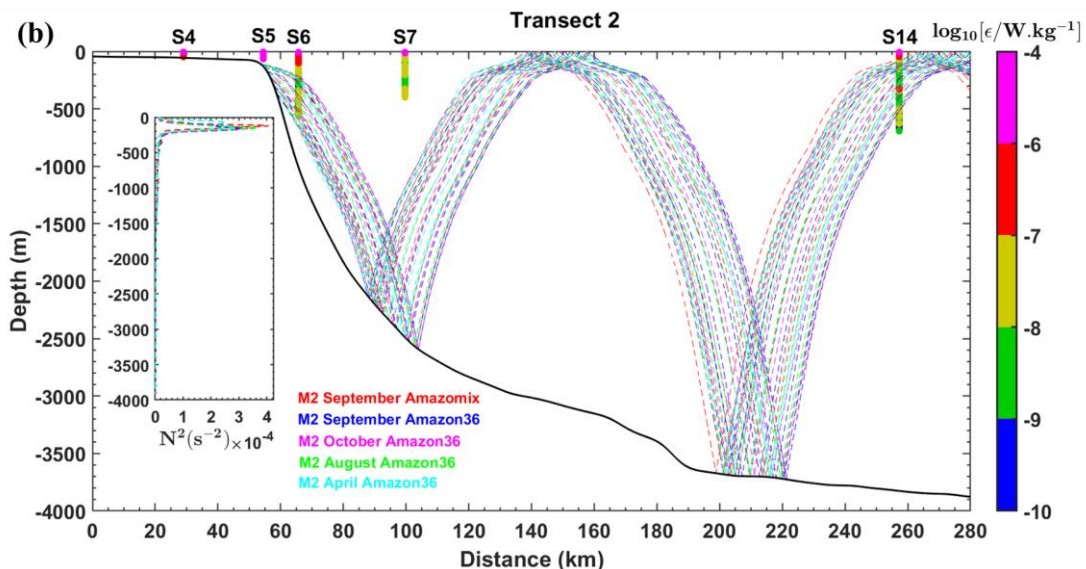




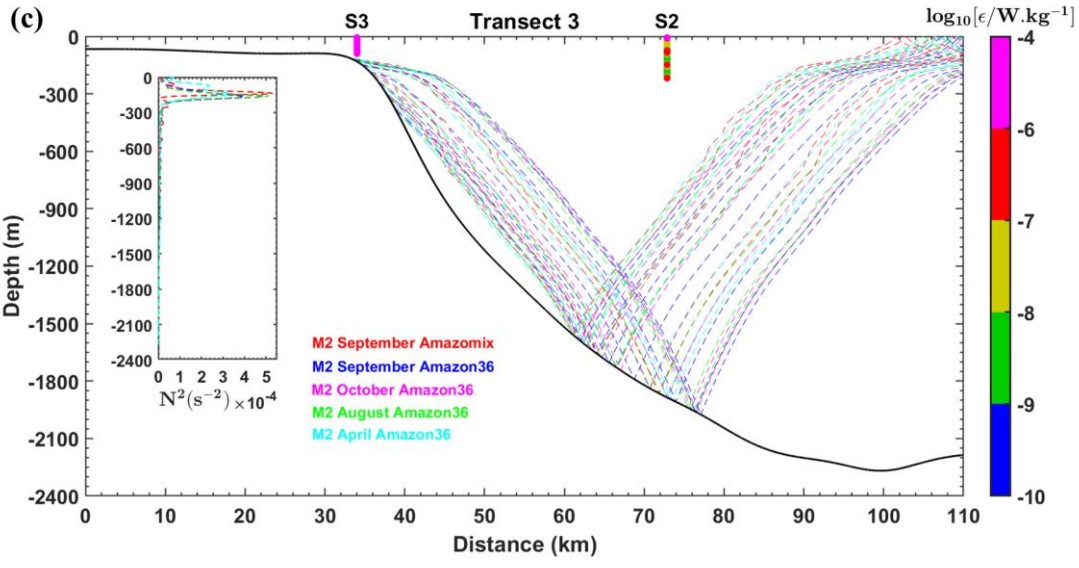


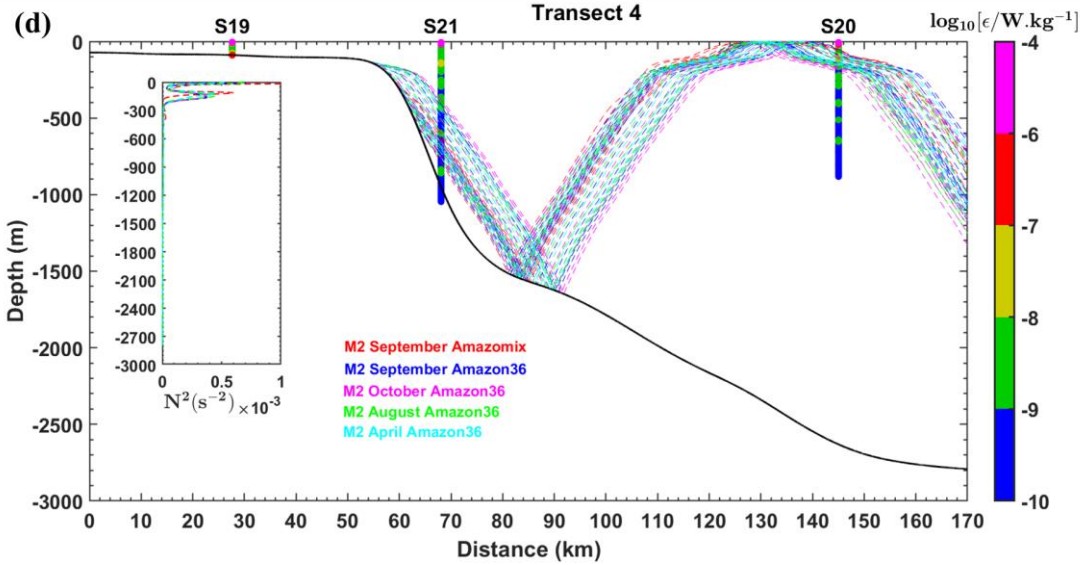




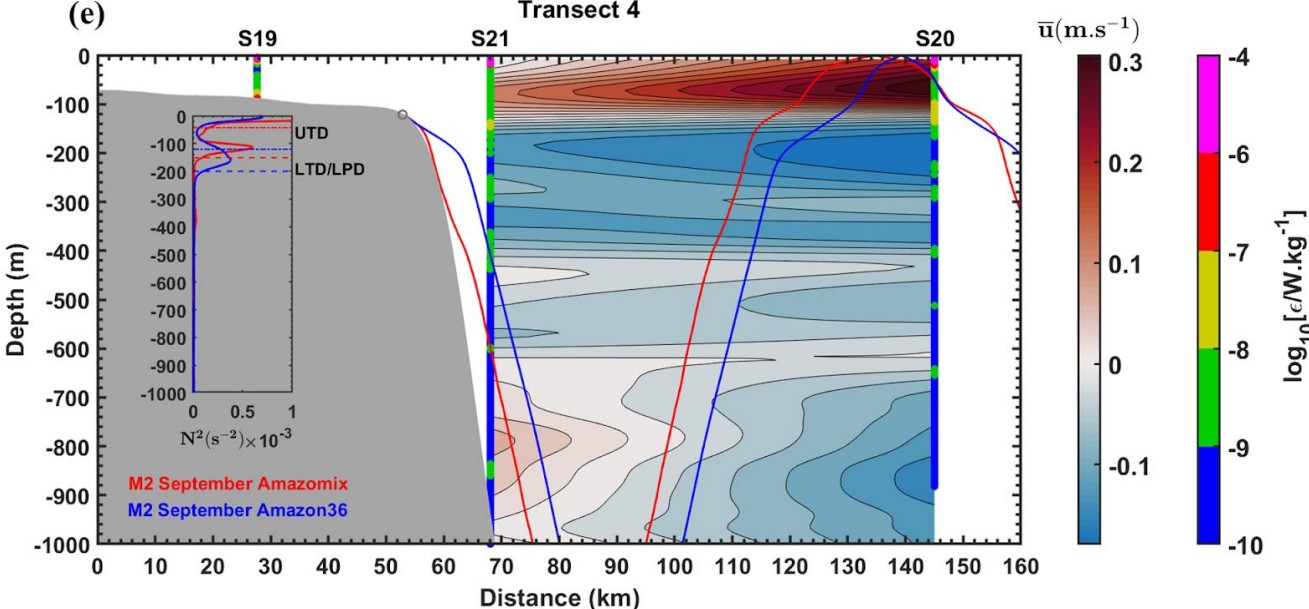

*Figure A3:* (a)-(d) Sensibility tests of IT ray tracing by varying the location of the critical topography in each month. *Ray tracing for the M2 tidal constituent assuming the mean stratification ($N^2$ in $s^{-2}$, in overlaid Figure in each panel) from AMAZOMIX CTD-O₂ (of September 2021) and NEMO model (Amazon36 of September, October, August and April, 2012-2016), along the transects (a) T1, (b) T2, (c) T3 and (d) T4. Overlaid on the panels the profiles of dissipation rates (ε in W.kg$^{-1}$ in log scale, in coloured bars) from VMP. Solid black lines represent local bathymetry. Dashed colour lines are used to distinguish IT beam months in each transect. (e) Zoom in the upper layer of 1000 m at T4 showing IT ray tracing for September and the profiles of dissipation rates overlaid profiles of mean total (alongshore) current ($\bar{u}$ in m.s$^{-1}$, in background) from ADCP. UTD (in dotted lines) and LTD/LPD (in dashed lines) correspond to Upper and Lower Thermocline/Pycnocline Depth, respectively.* In the panel (e), grey circles indicate the critical slope and grey areas represent local bathymetry.



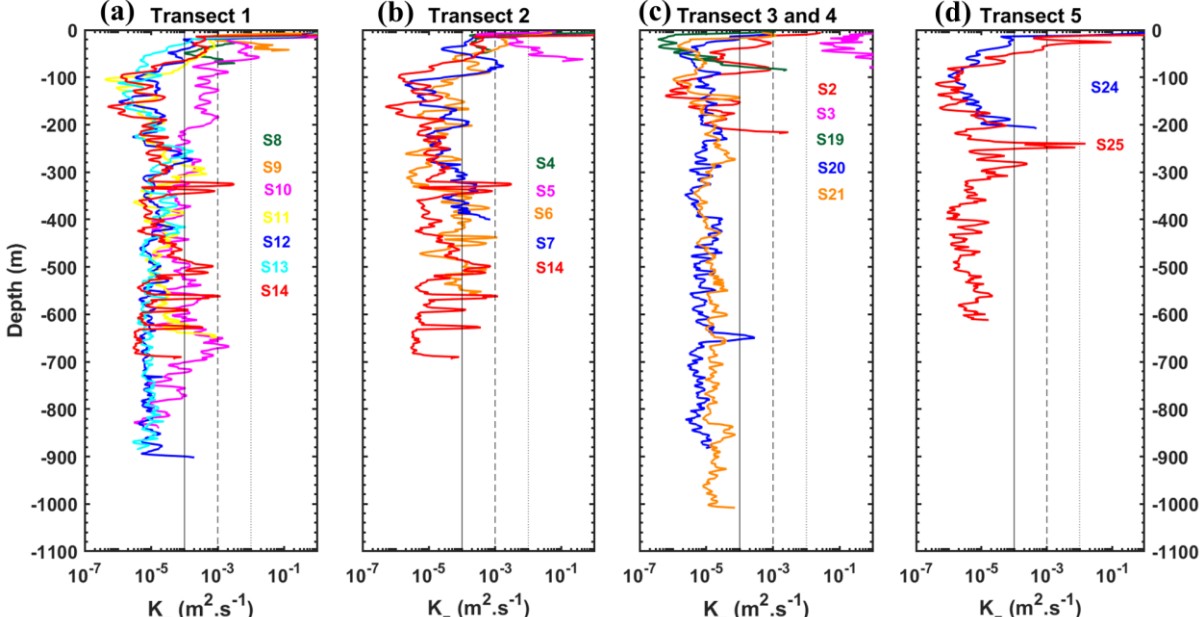

*Figure A4: Vertical diffusivity (in $m^2.s^{-1}$) during AMAZOMIX 2021 cruise for the transects/stations inside of the IT fields (a)*

*T1 (S8-S14), (b) T2 (S4-S7 and S14), and (c) T3 (S2 and S3) and T4 (S19-S21), and far from IT fields (d) T5 (S24 and S25).*

*Colour is used to distinguish each station in each transect. Dashed and solid back lines are for comparison.*

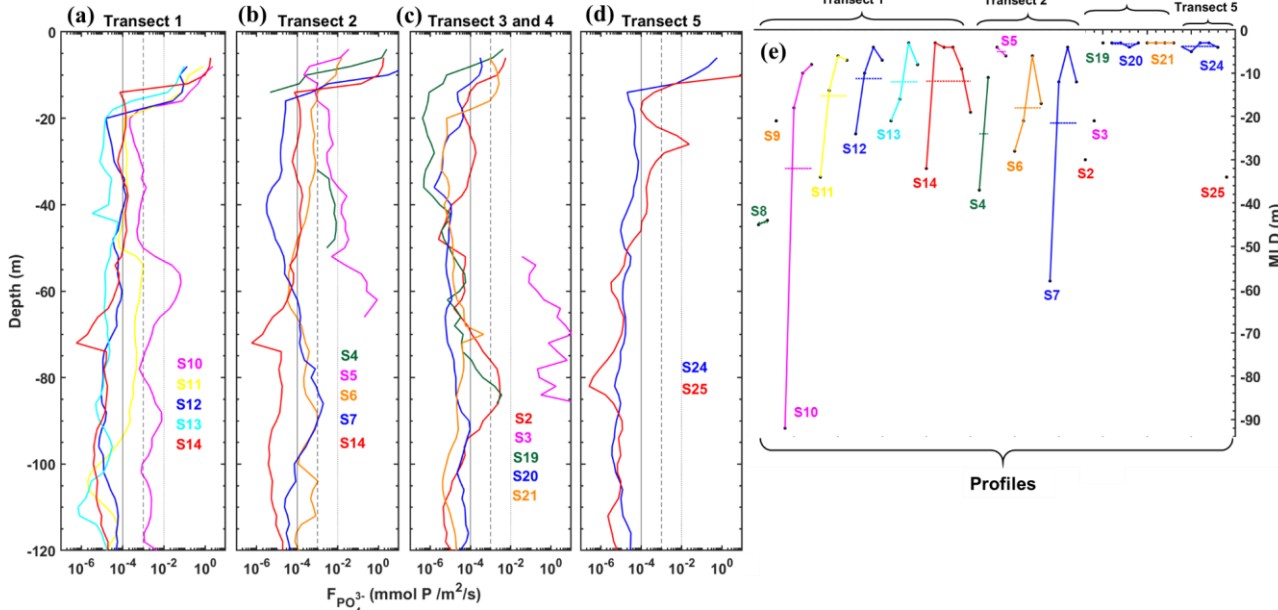

*Figure A5: Vertical fluxes of phosphate (in mmol P $m^{-2}.s^{-1}$) at the base of MLD (defined at the MLD + 1m) during AMAZOMIX*

*2021 cruise for the transects/stations inside of the IT fields (a) T1 (S10-S14), (b) T2 (S4-S7 and S14), and (c) T3 (S2 and S3)*



*and T4 (S19-S21), and far from IT fields (d) T5 (S24 and S25). Dashed and solid back lines are for comparison. (e) MLD (in m) for each CTD-O$_2$ profile (black dots) with average values (horizontal dashed lines) for each station. Colour is used to distinguish each station in each transect. S9 was missing from the phosphate data and the vertical flux of phosphate was null at S8 because of null phosphate concentration gradient in Fig. A5.a.*

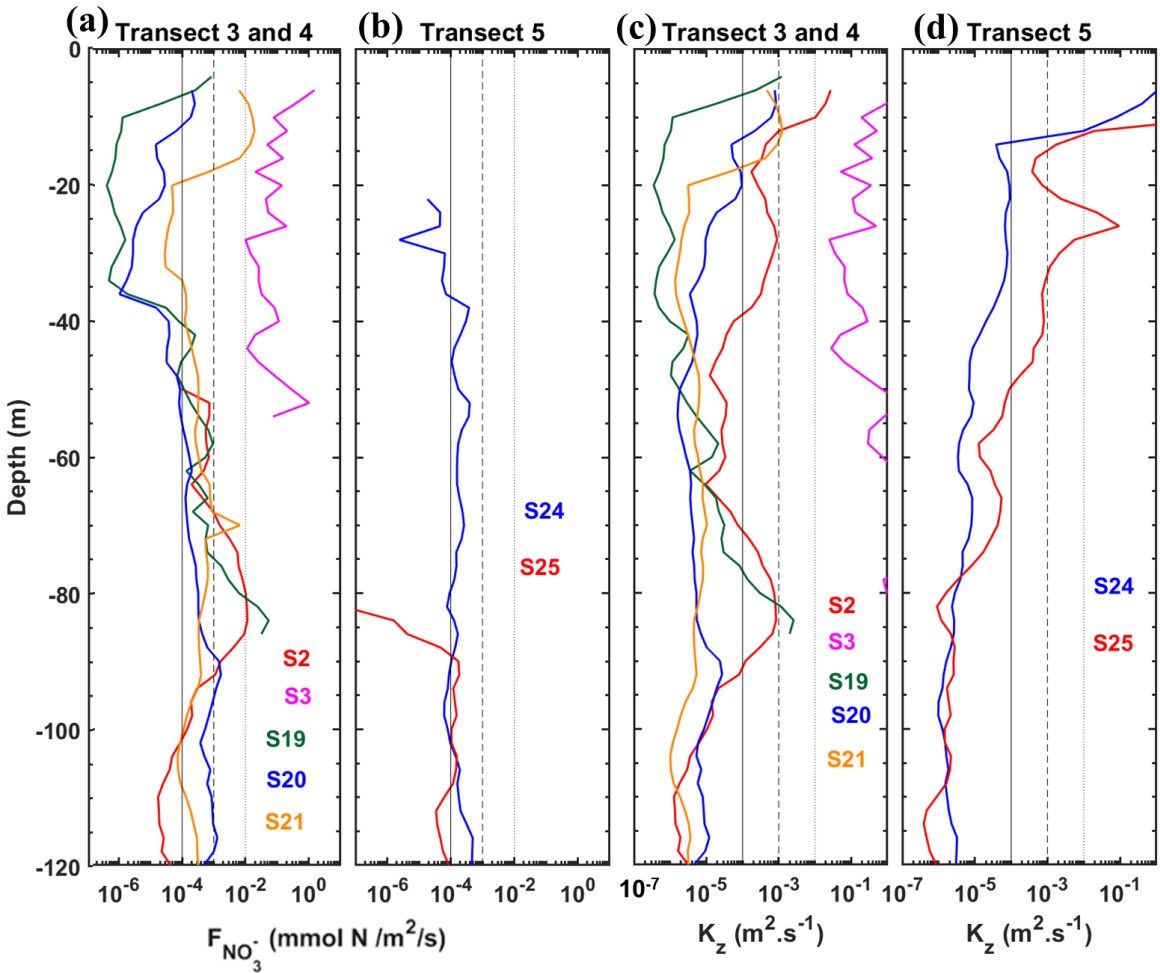

*Figure A6: (a)-(b) Vertical fluxes of nitrate (in mmol N m$^{-2}$.s$^{-1}$) and (c)-(d) vertical diffusivity (in m$^2$.s$^{-1}$) at the base of MLD (defined at the MLD + 1m, see Figure A5.e) during AMAZOMIX 2021 cruise for the transects/stations inside of the IT fields (a)-(c) T3 (S2 and S3) and T4 (S19-S21), and far from the IT fields (b)-(d) T5 (S24 and S25). Dashed and solid back lines are for comparison. Colour is used to distinguish each station in each transect.*



*Table A1:  miXing Layer Depth (XLD), Bottom layer-integrated flow Kinetic Energy (BKE), Contribution of the semi-diurnal*
*(CSBE) and mean baroclinic (CMBE) energy to total baroclinic energy at AMAZOMIX stations. BKE was absent at stations*
*with density stratification until the end of the profile.*

| N° Stations | XLD (m) | BKE (J.m$^{-2}$) | CSBE (%) | CMBE (%) |
|---|---|---|---|---|
| S2 | 16 | - | - | - |
| S3 | 16 | 22.94 | - | - |
| S4 | 12 | - | - | - |
| S5 | 8 | 9.52 | - | - |
| S6 | 18 | - | 61.2 | 38.8 |
| S7 | 44 | - | 48.38 | 51.62 |
| S8 | 22 | 0.45 | - | - |
| S9 | 14 | 1.32 | - | - |
| S10 | 22 | - | 65.94 | 34.06 |
| S11 | 18 | - | 48.07 | 51.93 |
| S12 | 18 | - | 51.85 | 48.15 |
| S13 | 18 | - | 56.03 | 43.97 |
| S14 | 14 | - | 58.94 | 41.06 |
| S19 | 24 | 0.56 | - | - |
| S20 | 36 | - | 52.64 | 47.36 |
| S21 | 36 | - | 59.13 | 40.87 |
| S24 | 14 | 0.18 | 49.59 | 50.41 |
| S25 | 12 | - | - | - |


**Data availability**

The AMAZOMIX data can be downloaded directly on the SEANOE site: https://www.seanoe.org/data/00860/97235/. The
NEMOv3.6 model outputs are available upon request by contacting the corresponding author.

**Authors contributions**

AKL: funding acquisition. FK and AKL, with assistance from JM: conceptualization and methodology. FK, with assistance
from PR, AB, EC and AKL: data pre-processing. Formal analysis: FK with interactions from all co-authors. Preparation of the
manuscript: FK with contributions from all co-authors. This work is a contribution to the LMI TAPIOCA (www.tapioca.ird.fr).

**Competing interests**



The authors declare that they have no conflict of interest.

**Acknowledgments**

The authors would like to thank the "Flotte Océanographique Française" and the officers and crew of the R/V Antea for their
contribution to the success of the operations aboard the R/V ANTEA, as well as, all the scientists involved in data and water
samples collection, for their valuable support during and after the AMAZOMIX cruise. We acknowledge the Brazilian
authorities for authorising the survey. The authors acknowledge Rockland company for their instrument and their support
during the cruise and during the analysis of the VMP data, the National french parc of instrument (DT-INSU) for their
instrument during the cruise and support in data analysis, as well as, the US-IMAGO from IRD for its help during the cruise
and for biogeochemical data analysis.

**Financial support**

This work is a part of the project "AMAZOMIX", funded multiple agencies : the "Flotte Océanographique Française" that
funded the 40 days at sea of the R/V Antea, the Institut de Recherche pour le Développement (IRD), via among other the LMI
TAPIOCA, the CNES, within the framework of the APR TOSCA MIAMAZ TOSCA project (PIs Ariane Koch-Larrouy,
Vincent Vantrepotte, and Isabelle Dadou), the LEGOS and the program international Franco-Brazileiro GUYAMAZON (call
N° 005/2017). It is also part of the PhD Thesis of Fabius Kouogang, funded by the Coordenação de Aperfeiçoamento de
Pessoal de Nível Superior (CAPES), under the co-advisement of Ariane Koch-Larrouy and Moacyr Araujo.

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
