# Peer review of "Turbulent dissipation along contrasting internal tide paths, off the"

_EGUsphere, 2024_

## Author Comment (AC1)

We thank this Reviewer for thoughtful and constructive comments on our manuscript. We appreciate the time s/he invested in the review. We believe that our revised manuscript addresses all the comments. In this regard, we have revised and rewritten a few sections such as Abstract, method, results, discussion and conclusion in the revised manuscript. Quality control and revision of dissipation estimates, as well as current calculations were also done to ensure the validity of our results. We thought it useful to point out its detailed revisions (lines and sections) in the reply to your comments. Below (highlighted in blue and magenta) is an itemized response to the different issues raised in the review.

**Review**

Turbulent dissipation from AMAZOMIX off the Amazon shelf along internal tides paths

Kouogang et al.

**Summary**

This manuscript describes measurements of currents, hydrography, and turbulence at numerous sites near the Amazon outflow, where internal tides are also generated at the slope. The dataset appears suited to study the questions posed by the authors. Do internal tides affect the mixing? Does the mixing affect nutrient fluxes?

However, I have some concerns about the analysis listed below. Since many of the details of the analysis are not presented, it is difficult for me to evaluate what has been done. Some of the more prominent ones include:

tidal decomposition of the currents,

use/decomposition of shear,

description/applicability of the finescale parameterizations used,

and shaded color plots of SADCP velocity that include transits and times on station.

There are also presentation/stylistic problems that make the manuscript very difficult to read. Many of them are major, but also getting down to such points as the labelling of figures and sections. Some examples are listed below.  It is well beyond the scope of a review to point all of these out. Furthermore, the cumulative difficulty of numerous minor things (let alone the more major ones) turns into a major distraction to the detriment of the hard work done and the science. The senior authors must make considerably more effort in this regard.

**Major comments**

1. SI units should be written W kg-1 not W.kg-1 and 90 km not 90km and so on.

 R: Thanks for your remarks.

We have revised the SI units throughout the revised manuscript.

an example, in lines 22-24 of the revised manuscript, is shown below:

"

Near generation sites, mixing rates were elevated, between $[10^{-6}, 10^{-5}]$ W kg$^{-1}$, with IT shear contributing ~65 %, compared to mean baroclinic current (BC) shear. Along IT pathways and in far-field IT regions, mixing decreased to $[10^{-8}, 10^{-7}]$ W kg$^{-1}$ but remained substantial, driven by nearly equal contributions from IT and BC shear.

"

2. 50.4% on line 29.  Please state the error and adjust the significant figures accordingly.

R: Thanks for your remarks.

We have revised as suggested

The error, associated with the contribution of tidal and mean shear, was estimated with the standard deviation. It is relatively small. you will find the error we have declared in table A1 of the revised manuscript.

3.  line 34 - A "guide for the mixing parameterization in future numerical simulation" is mentioned. What is the guidance?

R: Thanks for your remark.

Indeed, we have revised all the Abstract and sentence in line 29-30 of the revised manuscript:

"

These findings provide valuable insights for developing parameterizations of tidal and mean shear mixing for ocean general circulation models and coupled models, including biogeochemical and atmospheric systems.
"

4. Section 2.2- some of the microstructure processing steps should be shown in either the main text, appendix, or supplementary material. In particular, I would like to see some examples of the spectral fits.

R: Thanks for your suggestions.

We have incorporated additional microstructure processing steps and spectral fits into both the main text and the appendix. Revisions and checks of the VMP profile processing, and dissipation estimates have been carried out, along with quality assurance tests, in alignment with all reviewers' comments and ATOMIX's recommendations (Lueck et al., 2024).

a) First, the revisions were focused on the VMP profile processing, particularly on:

- Parameters controlling shear spectra estimation, such as record lengths (L), which are cosine-windowed and overlapped (O) by 50%.
- Parameter for extracting the section or profile (continuous part of the time series), including the minimum depth of extraction (Pmin).

For shallow stations, we used L = 5 s and O = 2.5 s, instead of the previous L = 4 s and O = 2 s. Parameters for deep stations remain unchanged (L = 8 s and O = 4 s).

Additionally, we adjusted the minimum depth of extraction to Pmin = 10 m for deep stations and Pmin = 3 m for shallow stations, compared to the previously used value of Pmin = 1 m for both station types.

b) Second, the revisions and checks focused on the quality assurance measures for dissipation estimates, with quality control checks and adjustments applied across all stations (e.g., S2, S7, S12, S14).

For instance, at station S14, previous dissipation estimates showed some peaks at various depths, particularly between 100–200 m, 300–400 m, and 500–700 m. While the fraction of shear data affected by despiking during processing was <0.05, the figure of merit (FM)—used to filter out poor-quality data—for shear probe 1 was >>1.4 at certain depths (e.g., around 327 m compared to 122 m, as shown in figure RC1.1 below). In contrast, the FM for shear probe 2 remained <1.4.

After checks, quality control of dissipation estimates have been revised for all stations (e.g., S6, S10, S14). We have retained only dissipation estimates from either one or both probes that met the quality assurance criteria (FM < 1.4 and fraction of despiked shear data <0.05, as recommended by ATOMIX), as shown at S6 and S10 for example (figure RC1.2 below).

The final dissipation estimate was computed as the average of the estimates from both shear probes, followed by the mean of the dissipation profiles for the station, as illustrated in figure RC1.3 below.

The revision of section "methods/TKE dissipation rates" can be found in lines 121-151 of the revised manuscript, as shown below:

"

**2.2 Methods**

**TKE dissipation rates**

The VMP data are processed using ODAS Matlab library (developed by Rockland Scientific International, Inc) to infer the TKE dissipation rate ($\varepsilon$). The processing methods for the VMP data are briefly described here and adhere to the recommendations of ATOMIX (Analyzing ocean turbulence observations to quantify mixing), as reported by Lueck et al. (2024), and have been validated against the benchmark estimates (presented in Fer et al., 2024).

First, the VMP data are converted into physical shear units, and the time series are prepared. Continuous sections of the time series are selected for dissipation estimation. Before spectral estimation, the aberrant shear signals caused by vessel wake contamination are removed. Collisions of the shear probe with plankton and other particles are removed using the de-spiking routine. The records from each section are then high-pass filtered (e.g., at station S6 and S10; Fig. 2a, and Fig. A1, Appendix).

Shear spectra are estimated using record lengths (L) and Fast Fourier Transform segments of 2 s, which are cosine windowed and overlapped by 50% (e.g., at station S6 and S10; Fig. 2b, and Fig. A1, Appendix). Additionally, vibration-coherent noise is removed. Different L and overlap (O) settings were selected and tested based on the environment (e.g., deep vs. shallow water), following Fer et al. (2024). For shallow stations, L (O) was shortened to 5 s (2.5 s), in contrast to the 8 s (4 s) used for deeper stations, due to evidence of overturns observed in AMAZOMIX acoustic measurements at deeper stations (Koch-Larrouy et al., 2024; in preparation). This adjustment helped to optimize the spatial resolution of dissipation estimates in shallow water stations.

Finally, $\varepsilon$ is determined using the spectral integration method and by comparison with the Nasmyth empirical spectrum (Nasmyth, 1970). Quality assurance tests are carried out in accordance with ATOMIX's recommendations (Lueck et al., 2024). A figure of merit < 1.4 is used to exclude bad data (e.g., at station S6 and S10; Fig. 2b, and Fig. A1, Appendix), and the fraction of data affected by de-spiking is < 0.05.

[Figure]

Figure 2: Example of wavenumber spectra from a dissipation structure segment used to determine the dissipation rate at station S6 at a pressure of 87.9 dBar. (a) Cleaned and high-pass filtered signals from shear probe 1 (blue) and shear probe 2 (red, offset by 5 s$^{-1}$). (b) Wavenumber spectra for shear probes 1 and 2. Thick lines (blue for probe 1, red for probe 2) show shear spectra with coherent noise correction, while thin lines (sky blue for probe 1, orange for probe 2) show spectra without correction. Triangles mark the maximum wavenumber used for dissipation rate estimation. Black lines represent Nasmyth reference spectra for estimated dissipation rate of 3.8 x 10$^{-8}$ W kg$^{-1}$ for both shear probes. Dissipation rate estimates for shear probe 1 and shear probe 2 at a pressure of 87.9 dBar yielded a figure of merit of 0.93 and 0.94, respectively.

"

[Figure]

Figure RC1.1: Example of wavenumber spectra from a dissipation structure segment used to determine the dissipation rate at station S14 at a pressure of 337.7 dBar. (a) Pressure record for the entire data file (blue) and the specific segment being analyzed (red). (b) Cleaned and high-pass filtered signals from shear probe 1 (blue) and shear probe 2 (red, offset by $5\,s^{-1}$). (c) Wavenumber spectra for shear probes 1 and 2. Thick lines (blue for probe 1, red for probe 2) show shear spectra with coherent noise correction, while thin lines (sky blue for probe 1, orange for probe 2) show spectra without correction. Triangles mark the maximum wavenumber used for dissipation rate estimation. Black lines represent Nasmyth reference spectra for estimated dissipation rate of 8.9 x $10^{-10}$ W $kg^{-1}$ and 1 x $10^{-6}$ W $kg^{-1}$ for shear probes 1 and 2, respectively. Dissipation rate estimates for shear probes 1 and shear probe 2 at a pressure of 337.7 dBar yielded a figure of merit of 1.3 and 9, respectively. Panel (d) is similar to panel (c) but:- with Nasmyth reference spectra for estimated dissipation rate of 3.9 x $10^{-8}$ W $kg^{-1}$ and 4.6 x $10^{-8}$ W $kg^{-1}$ for shear probes 1 and 2. -wth dissipation rate estimates for shear probes 1 and 2 at a pressure of 122.4 dBar yielding a figure of merit of 0.84 and 1, respectively.

[Figure]

Figure RC1.2: Similar to Fig. RC1.1. but for stations (a)-(b)-(c) S6 and (d)-(e)-(f) S10. For S6 (panels c), Black lines represent Nasmyth reference spectra for estimated dissipation rate of 3.8 x 10-8 W kg-1 for both shear probes, and dissipation rate estimates for shear probes 1 and shear probe 2 at a pressure of 337.7 dBar yielded a figure of merit of 0.93 and 0.94, respectively. For S10 (panel f), Black lines represent Nasmyth reference spectra for estimated dissipation rate of 1.6 x 10$^{-8}$ W kg$^{-1}$ and 1.5 x 10$^{-8}$ W kg$^{-1}$ for shear probes 1 and 2, respectively, and dissipation rate estimates for both shear probes at a pressure of 337.7 dBar yielded a figure of merit of 1.2.

[Figure]

figure RC1.3: (a) Horizontal maximum and (b)-(c)-(d)-(e) vertical dissipation rates (ε, in W kg-1, on a logarithmic scale) before revisions and checks processes for all stations along transects T1 to T2. (f) Horizontal maximum and (g)-(h)-(i)-(j) vertical dissipation rates (ε, in W kg-1, on a logarithmic scale) after revisions and checks processes for all stations along transects Aa, Ab, D, E, and G. Distinct colors are used to represent each station within each transect. Dashed and solid black lines in panels (b) to (e) are included for comparison purposes.

5.  Semidiurnal currents are obtained by removing the mean current over a tidal period from the baroclinic velocities.  This will include other frequencies.  In fact Figure 4 has everything: 6 hrs, 12 hrs, …  It would be better to make a least squares fit of sines and cosines to the various tidal periods. With a short time series of 1 day, the frequency resolution is 1 cycle/day.  So you could try fitting diurnal and semidiurnal, but recognizing that they are not formally distinct.  Also the inertial period here is at least 5 days and so cannot be determined in your dataset.  This is a key limitation that should be pointed out.  At the equator every frequency just about is in the internal wave band.

 How or even whether the currents should be separated in frequency seems to be a major question given the limitations of the data and the desire to examine dissipation.

R: Thanks for your comments and suggestions. Based on your suggestion we completely re-worked our methodology, it has gained in robustness.

Indeed, semidiurnal currents are obtained by removing the mean current over a tidal period from the baroclinic velocities. This could include other frequencies. We have tried to make a least squares fit of sines and cosines to the various tidal periods (diurnal and semidiurnal), as you recommended. This fitting only resolved for the S14 station, which is 42 hours long, compared to the other shorter stations of 17 hours. This may be due to the length of the time series.

Below we present the result at S14. Figure RC1.4 shows the time series of the data collected and Figure RC1.5 and RC1.6 shows the diurnal and semidiurnal fitting at S14.

Indeed, with a short time series of 1 day, the frequency resolution is 1 cycle/day. We recognize that diurnal and semidiurnal fitting are not formally distinct, and that the inertial period here is at least 5 days and so cannot be determined in our dataset. We have pointed out this key limitation, as you recommended, in the manuscript in the lines 191-197 of the revised manuscript.

We know that our time series data are shorter than 17 h hours at all long stations, except for S14 (long of 42 hours). Indeed, the diurnal and semidiurnal periods fitting are not formally distinct and the inertial period (at least 5 days) cannot be determined in our dataset, limiting the currents separation in frequency and our desire to examine dissipation associated.

We have added the Figure RC1.4 and Figure RC1.5 in the Appendix of the revised manuscript (Figs. A4 and A5).

Indeed, even if currents need to be separated in frequency, this seems to be a major issue given the limitations of the data and the desire to examine dissipation. We have also pointed out this limitation in the manuscript in the lines 191-197 of the revised manuscript, as shown below:

"

Note that continuously collected SADCP for some stations (e.g., S11) are not sufficiently resolved due to gaps filled by interpolating between time points. The similar processing are applied to the CTD-O$_2$ data collected alternately. SADCP time series data are less than 17 hours at all long stations, except for S14, which spans 42 hours. As a result, the diurnal and semidiurnal period fittings are not formally distinct (except at S14; Figs. A4 and A5, Appendix), and the inertial period (at least 5 days) cannot be resolved in our dataset. This limits our ability to separate currents by frequency and examine the associated dissipation.

The velocity profiles from LADCP are glued into our SADCP time series data below ~ 500 m depth at long stations..

"

[Figure]

Figure RC1.4: Time series of (a) zonal and (b) meridional current of the original SADCP data at a fixed position for station S14. Time is scaled to start at t=0.

[Figure]

Figure RC1.5: Least squares fit of sines and cosines to M2 period for (a) zonal and (b) meridional current at station S14. Time is scaled to start at t=0.

[Figure]

Figure RC1.6: Least squares fit of sines and cosines to semidiurnal (M2+S2+N2+K2) + M4 + diurnal (K1+O1+P1+Q1) periods for (a) zonal and (b) meridional current at station S14. Time is scaled to start at t=0.

The revision of section "methods/Baroclinic currents" can be found below in lines 178-212 of the revised manuscript:

"

**Baroclinic currents**

To analyze the processes explaining dissipation and mixing, particularly along internal tidal (IT) paths, we estimate shear instabilities associated with the semi-diurnal (M2) ITs and mean circulation, as well as their contributions to mixing.

The M2 tidal component of the tidal current is derived by calculating the baroclinic (semi-diurnal) tidal velocity $[u'', v'']$ (Fig. A3, Appendix), following these equations:

$$[u', v'] = [u, v] - [u_{bt}, v_{bt}],$$

(1)

$$[u_{bt}, v_{bt}] = \frac{1}{H} \int_{-H}^{0} [u, v] dz, \qquad (2)$$

$$[u'', v''] = [u', v'] - [\overline{u'}, \overline{v'}]. \qquad (3)$$

Here, $[u, v]$ represent total horizontal velocities (Fig. A3, Appendix) obtained from SADCP data. The components $[u', v']$ and $[u_{bt}, v_{bt}]$ represent baroclinic and barotropic components of horizontal velocities, respectively (Fig. A3, Appendix). H is water depth. The baroclinic mean velocities $[\overline{u'}, \overline{v'}]$ (Fig. A3, Appendix), calculated to estimate mean circulation along IT paths, are decomposed into along-shore $\overline{u'}_l$ and cross-shore $\overline{u'}_c$ velocities. The overbar denotes the average over a M2 tidal period.

Note that continuously collected SADCP for some stations (e.g., S11) are not sufficiently resolved due to gaps filled by interpolating between time points. The similar processing are applied to the CTD-$O_2$ data collected alternately. SADCP time series data are less than 17 hours at all long stations, except for S14, which spans 42 hours. As a result, the diurnal and semidiurnal period fittings are not formally distinct (except at S14; Figs. A4 and A5, Appendix), and the inertial period (at least 5 days) cannot be resolved in our dataset. This limits our ability to separate currents by frequency and examine the associated dissipation.

The velocity profiles from LADCP are glued into our SADCP time series data below ~ 500 m depth at long stations.

To evaluate shear instabilities associated with ITs and the mean background circulation, we compute the baroclinic tidal vertical shear squared ($S^{2''}$) and mean shear squared ($\overline{S^{2'}}$) (Fig. A3, Appendix), as follows:

$$S^{2''} = (\partial u''/\partial z)^2 + (\partial v''/\partial z)^2, \qquad (4)$$

$$\overline{S^{2'}} = (\partial \overline{u'}/\partial z)^2 + (\partial \overline{v'}/\partial z)^2. \qquad (5)$$

To evaluate the impact of bottom friction on mixing, we calculate kinetic energy $\varepsilon_f = \frac{1}{2}\rho_s(u_f^2)$ near the bottom boundary layer at shallow stations using friction velocity $u_f = u_b\sqrt{C_d}$, where $C_d$=2.5 x 10$^{-3}$ is a drag coefficient obtained from the NEMO model. Huang et al. (2019) showed that the bottom boundary layer thickness spatially varies between 15-123 m in the Atlantic Ocean, with a median of ~ 30-40 m in the North Atlantic. We define bottom layer thicknesses in our study area based on measured bathymetry from CTD-$O_2$ and near-bottom currents from ADCP. Here, $u_b$ is the total velocity averaged over a thickness of 20 m above the seabed for shallow stations and 40 m for deep stations.

The individual contributions of semi-diurnal ITs and mean circulation are then expressed as follows: $E''/(\overline{E'} + E'')$ for tidal contribution and $\overline{E'}/(\overline{E'} + E'')$ for mean circulation contribution. Here, $E = N*S$. N is the buoyancy frequency and $S$ is vertical shear. S can be substituted by $S^{2''}$ and $\overline{S^{2'}}$.

"

6. Shear. Based on the calculations for barotropic u, baroclinic u, and tidal u. I do not see why shear is different for baroclinic and tidal u.

Tidal u = baroclinic u - lowpassed u.

It seems like it should be almost the same as the shear in baroclinic u. I'm confused. It would be helpful to see more of the processing steps and their justifications for these steps.

R: Thanks for your comments.

We have explained with more care the methodology step by step. Please see the previous response to your comment n°5.

Perhaps there is a problem with the short time series or large tidal isopycnal displacements. I suggest plotting some profiles of u and shear for barotropic, baroclinic, and tidal u to make sure these calculations are correct.

R: Thanks for your comments.

Indeed, our time series from SADCP are shorter than 17 hours at all long stations, except for S14 (long of 42 hours). Our calculations, as shown below, seem correct to be used. Please see below the example for S14 over a (M2) tidal cycle, the plots of u and shear for total, barotropic, total baroclinic, and baroclinic tidal, and mean baroclinic in figures RC1.7. We have added this example in the Appendix of the revised manuscript (Fig. A3)

[Figure]

Figure RC1.7: For Station S14. zonal currents for (b) total, (a) barotropic, (c) total baroclinic, (d) semi-diurnal baroclinic tidal, and (e) mean baroclinic. Meridional currents for (g) total, (f) barotropic, (h) total baroclinic, (i) semi-diurnal baroclinic tidal, and (j) mean baroclinic. Vertical shear for (k) total, (l) total baroclinic, (m) semi-diurnal baroclinic tidal, and (n) mean baroclinic.

Also you could try doing these calculations on an isopycnal since it looks like the velocity structure is being heaved up and down by the internal waves. That could lead to artifacts in the calculations on depth surfaces. For example in Figure 4a1, on sigma = 26, current is negative.

R: Thanks for your comments.

We performed the calculations on the 26 kg/m³ isopycnal as suggested and presented the results in the figure RC1.8 below. But we think that these results are not so robust since CTD data were collected alternately and needs to be interpolated between time points (e.g., between 17:00 and 20:00 at station S10). Consequently, the density data are not sufficiently resolved to allow the currents to be projected on over the tidal period. Doing this would introduce potential artifacts in the calculations on an isopycnal. So we decided to stick with our first calculation, pointing out in lines 191-197 of the revised manuscript:

"

Note that continuously collected SADCP for some stations (e.g., S11) are not sufficiently resolved due to gaps filled by interpolating between time points. The similar processing are applied to the CTD-$O_2$ data collected alternately. SADCP time series data are less than 17 hours at all long stations, except for S14, which spans 42 hours. As a result, the diurnal and semidiurnal period fittings are not formally distinct (except at S14; Figs. A4 and A5, Appendix), and the inertial period (at least 5 days) cannot be resolved in our dataset. This limits our ability to separate currents by frequency and examine the associated dissipation.

The velocity profiles from LADCP are glued into our SADCP time series data below ~ 500 m depth at long stations.

"

[Figure]

Figure RC1.8: For Station S10, calculations on the 26 kg/m³ isopycnal. zonal currents for (a) total, (b) total baroclinic, and (c) semi-diurnal baroclinic tidal. Meridional currents for (d) total, (e) total baroclinic, and (f) semi-diurnal baroclinic tidal. Vertical shear for (g) total, (h) total baroclinic, and (i) semi-diurnal baroclinic tidal.

At 120 m, current switches sign multiple times.  As far as I can tell, there are no plots of shear.

Also I do not understand why only one component of the current is shown.  Using both components is needed to understand propagation of internal tides either vertically or horizontally (alongshelf vs cross-shelf).

Section 3.2.2 uses a lot of words that could be instead expressed concisely in a map with current vectors.  Different colours could indicate different depths.

R:  Thanks for your comments.

We have added in the revised manuscript plots of the shear of shear for tidal U and mean U at selected stations (e.g., S10, S11, and S14; in Fig. 6, and Figs. A15 to A18, Appendix) and also reported here in figure RC1.9 below

Indeed, at 120 m, the current switches direction multiple times. We have added in the revised manuscript a 2D map of mean current (vectors) and also reported here in figure RC1.12 below.

Indeed, both zonal (u) and meridional (v) components are essential for understanding the propagation of internal tides, whether vertically or horizontally (along-shelf vs. cross-shelf). As we more focused on zonal (u) and meridional (v) components of ITs, we have added these components in the revised manuscript at selected stations (e.g., S10, S11, and S14; in Fig. 5, and Figs. A9 to A15, Appendix) and also reported here in figure RC1.10 and RC1.11 below.

Additionally, we have visualized the mean baroclinic current using a map with current vectors (as reported here in RC1.12).

In response to all reviewer comments, we have revised several sections of the manuscript, including "Methods," "Results," and "Discussion and Conclusion," spanning lines 121–568 of in the revised manuscript.

[Figure]

Figure RC1.9: *Semi-diurnal baroclinic vertical shear squared ($S^{2"}$, in m s$^{-1}$, on a logarithmic scale) for stations (a) S10, (c) S11, and (e) S14. Panels (a), (c), and (e) also display the buoyancy frequency squared ($N^2$, in s$^{-2}$) represented by vertical black lines, potential density represented by grey contours, and dissipation rate profiles ($\varepsilon$, in W kg$^{-1}$, on a logarithmic scale) represented by vertical colored bars. Mean baroclinic vertical shear squared ($\overline{S^{2'}}$, in m s$^{-1}$) for stations (b) S10, (d) S11, and (f) S14.*

[Figure]

Figure RC1.10: *Semi-diurnal baroclinic zonal currents (u", in m s⁻¹) from the ADCP for stations (a) S10, (c) S11, and (e) S14. Panels (a), (c), and (e) also display the buoyancy frequency squared (N², in s⁻²) represented by vertical black lines, potential density represented by grey contours, and dissipation rate profiles (ε, in W kg⁻¹, on a logarithmic scale) represented by vertical colored bars. Along-shore mean baroclinic currents (u'ₗ, in m s⁻¹) from the ADCP for stations (b) S10, (d) S11, and (f) S14.*

[Figure]

figure RC1.11: *Semi-diurnal baroclinic meridional currents (u", in m s⁻¹) from the ADCP for stations (a) S10, (c) S11, and (e) S14. Panels (a), (c), and (e) also display the buoyancy frequency squared (N², in s⁻²) represented by vertical black lines, potential density represented by grey contours, and dissipation rate profiles (ε, in W kg⁻¹, on a logarithmic scale) represented by vertical colored bars. Cross-shore mean baroclinic currents (ū'ₗ, in m s⁻¹) from the ADCP for stations (b) S10, (d) S11, and (f) S14.*

[Figure]

Figure RC1.12: Map of mean baroclinic currents (vectors) at stations. Colored arrows are used to distinguish the currents at different depths: Black, blue, cyan, yellow, green, red and magenta arrows at depth equal to 8 m, 120 m, 200 m, 304 m, 400 m, 504 m and 600 m, respectively. The blue line indicates the 200 m isobath.7. Dissipation is estimated using parameterizations based on shear in the tidal current. Actual dissipation depends on the total shear or strain.

R: Thanks for your remarks.

Indeed, dissipation was estimated using parameterizations based on shear in the tidal current. MacKinnon-Gregg parameterization was applied as a proxy to evaluate the contributions of tidal and low-frequency shear, primarily for comparison purposes. However, no scientific results were derived from it.

Ultimately, thanks to all the reviewers comments, we decided to remove the section using this parameterization. Eventually, we will work on a separate paper to test the robustness of this parameterization against these observations.

In the present paper, we focused on vertical shear to separate the contributions of tidal and mean shear. Revisions can be found in the "Methods" section in the revised manuscript.

8. Some more explanation of the applicability of MacKinnon-Gregg parameterization vs those based on Gregg (1989) is needed. The former is related to the limited bandwidth of internal waves (in shallow water for example), while the latter is for a typical open ocean environment away from generation sites. How does this apply to your study sites?

R: Thanks for your comments.

Please see the previous response to your comment n°7.

9. Naming of the sites and transects. Maybe these are what are used on the cruise but they should be rearranged in a logical order to *help the reader*. It is completely confusing as it is. Consider another system, such as A B C D … for generation sites. If there's a transect near site A, it's transect A.

If there are 3 stations on transect A, they are sites A1 A2 A3 and counting higher offshore. Or labelled by isobath, A500, A1000, A2000… There may not be a generation site for each transect.

For example lines 265-267 read: "The highest baroclinic tidal current velocities were observed (between 25-48 cm.s-1) at sites Aa and Ab along T1-T2. Whereas lower tidal velocities (< 25 cm.s-1) are found in site F along T4 (e.g., at S20 and S21) compared to OUT-ITs stations (e.g., at S24)." So I have to use the map. This could read instead as: "The highest baroclinic tidal current velocities were observed (between 25-48 cm.s-1) at sites D and E along transects D and E. Whereas lower tidal velocities (< 25 cm.s-1) are found in site A along transect A (e.g., at A2 and A3) compared to OUT-ITs stations (e.g., at site B1)."

R: Thanks for your comments and suggestions.

We acknowledge that the nomenclature might be somewhat confusing, and your suggestions would indeed simplify understanding. However, we have chosen to retain the naming convention used in previous studies of this region. Specifically, the names A to F for the generation sites were already defined in earlier works (e.g., Magalhães et al., 2016; Tchilibou et al., 2022; Assene et al., 2024; de Macedo et al., 2023).

To maintain consistency with these studies, we have decided to adhere to the established naming convention. As a compromise between simplifying the nomenclature and preserving coherence with previous works, we have designated the transects with the same letters as their corresponding generation sites: Aa, Ab, D, E, G, as shown in figure 1 of the revised manuscript and reported here in figure RC1.13.

Revisions reflecting this choice can be found in the subsection "Baroclinic tidal current" (lines 298–320) of the revised manuscript.

As an example, the previous phrase "is now transform in :

"

The baroclinic tidal velocities reveal a superposition of 3-5 tidal modes at IN-ITs stations (Figs. 5a, 5c, and 5e, and Figs. A9 to A15, Appendix). A greater number of modes is observed near the shelf-break (e.g., 4 modes at S6 and 5 modes at S10), while fewer modes are detected far from (e.g., 3 modes at S7, S12 and S14). Higher tidal velocities ranging from 25-50 cm s$^{-1}$ are found between 80-350 m along transects Aa and Ab (e.g., at S6, and S10). In contrast, lower tidal velocities, typically below 25 cm s$^{-1}$, are more pronounced along transect E (e.g., at S20, and S21) to OUT-ITs stations along transect G (e.g., at S24).

Figure RC1.13:*a) Map of a part of the AMAZOMIX 2021 cruise off the Amazon shelf, showing bathymetric contours (100 m, 750 m, 2000 m, and 3000 m isobaths) in gray. Colored circles and stars indicate short and long CTD-O2/L-S-ADCP stations, respectively, with the corresponding sampling dates represented by the color bar. Solid black lines depict SADCP transects (for Aa, Ab, D, G, and E). Magenta arrows show the 25-hour mean depth-integrated baroclinic IT energy flux (September 2015, from the NEMO model) originating from IT generation sites (Aa, Ab, D, and E) along the shelf break. The solid brown line represents the NBC pathways illustrating background circulation. Shattered colored lines highlight ISW signatures.*

10. Mixing is invoked as the explanation for increased nutrients. How about coastal upwelling? Where is the euphotic zone?

Thank you for this question, which gives us the opportunity to emphasize in the discussion that mixing is not the only significant process influencing the nutrient structure in the region.

Indeed, the nutrient inputs are controlled by several physical processes such as : Riverine Input, Coastal Upwelling, Internal Tides (ITs), Stratification Changes, and Eddies and Currents Off the Amazon shelf (Williams and Follows, 2003; Santos et al., 2008).

There is a very high variability of the euphotic layer in the region. It is very sensitive and heterogeneous from shelf to open ocean. The average estimated by in-situ observations is 13 m (Santos et al., 2008).

In our study, we tried to quantify the impact of internal tide on nutrient structuration. IT is indeed probably of a secondary order of importance in instantaneous but by its repetitiveness (12h, 14 Days), the cumulative effect of the tide over a year could impact the nutrient content. In addition, in a context of climate change this effect may be increased. (Yang et al., 2024)

In our study, we focused on quantifying what was measurable: specifically, vertical mixing, out of which approximately 50% was attributed to tidal processes. And, we also suggested that tidal mixing could contribute to nutrient inputs through vertical diffusivity since it has a strong contribution at the base of the mixed layer.

In the discussion, we contextualized these findings by emphasizing that mixing is not the only process influencing nutrient dynamics, and that upwelling also plays a significant role in nutrient supply.

Ultimately, we decided to remove and reserve all sections on "Nutrients fluxes" for a separate paper in progress.

11. Were adjustments to the shear parameterization based on the Gregg (2003), where it was shown that internal wave interactions may lead to little mixing at the equator?

R: Thanks for your comments and suggestions.

Please see the previous response to your comment n°7.

**Minor comments**

1. Line 107- While the bin size of the LADCP may be 8 m, its resolution will be about 50 m. Examine where the vertical wavenumber spectra fall off

R: Thanks for your comments.

We have examined the vertical wavenumber spectra. We fitted a slope to the spectrum in the steep region to estimate the roll-off wavenumber. For example, the spectrum falls off as $k^{-2.31}$ and $k^{-2.11}$ for the (a) zonal and (b) meridional at S14, as reported below in figure RC1.14.

[Figure]

Figure RC1.14: Panels (a) and (b) show the vertical wavenumber spectra for perturbations in the total zonal and meridional currents, respectively, derived from LADCP data at station S14. Black curves represent the vertical wavenumber spectra for individual current profiles over time, while the blue curves indicate the spectra averaged across all time profiles. The red and green lines illustrate the linear fits to the spectra slopes for individual profiles and the mean spectrum, respectively. The fitted slopes of the mean perturbation spectra are indicated in green: (a) -2.31 for the zonal current and (b) -2.11 for the meridional current.

2. I would recommend a native english speaker help with some of the grammar, if possible.  It's ok as is but could be much better.  It will be less distracting for the reader. Then the reader will pay more attention to the science.

R: Thanks for your suggestions. We have revised all sections of the manuscript.

3.  In sections 2.1 and 2.2, there is some jumping around from instrument to instrument.  Please collect into sections for each instrument.

R: Thanks for your comments and suggestions.

We have revised sections 2.1 and 2.2 in lines 92-234 of the revised manuscript and reported here:

"

**2 Data and Methods**

**2.1 Data collection**

[revised manuscript text omitted]

**The vertical eddy diffusivity coefficient**

The efficiency of turbulence in redistributing energy is assessed through the calculation of the vertical eddy diffusivity coefficient ($K_z$). This coefficient is particularly significant in regions such as pycnoclines, where stratification suppresses mixing, making turbulence-driven mixing a key mechanism for vertical energy transport (Thorpe, 2007).

$K_z$ is calculated from ε following the formulation of Osborn (1980), given by $K_z = \varepsilon\,\Gamma\,N^{-2}$. Here, $N^2$ is the buoyancy frequency squared, which is calculated using the sorted potential density profiles ($\sigma_\theta$) obtained from CTD-O$_2$ data. It is given by $N^2 = -\,(g/\rho_0)\,(d\sigma_\theta/dz)$, where $\rho_0$ is a reference density (1025 kg m$^{-3}$) and g is the gravitational acceleration. $\Gamma$ is the mixing efficiency, defined as the ratio between the buoyancy flux and the energy dissipation, and is typically set to 0.2, which corresponds to the critical Richardson number Ri = 0.17 (Osborn, 1980). ε is linearly interpolated into the depths of $N^2$.

Turbulence within the pycnocline can reduce stratification and increase vertical eddy diffusivity below the mixing layer (Thorpe, 2007). Subsurface mixing, driven by the breaking of ITs and shear instabilities, plays a particularly important role below the mixed layer, especially in equatorial waters (Gregg et al., 2003).

There are several criteria for defining the Mixed Layer Depth (MLD). In this study, we use the commonly accepted density threshold criterion of 0.03 kg m$^{-3}$, as defined by de Boyer Montégut et al. (2004) and Sutherland et al. (2014), to estimate the MLD for each CTD-O$_2$ profile. Notably, comparisons with density thresholds of 0.01 and 0.02 kg m$^{-3}$ revealed no major differences in MLD across the AMAZOMIX stations and transects (Fig. A2, Appendix).

The miXing Layer Depth (XLD) is defined as the depth at which ε decreases to a background level (Sutherland et al., 2014). Previous studies have applied various thresholds for background dissipation levels, such as such as 10$^{-8}$ and 10$^{-9}$ W kg$^{-1}$ in higher latitudes based on in situ observations (Sutherland et al., 2014; Lozovatsky et al., 2006; Cisewski et al., 2008; Brainerd and Gregg, 1995) and 10$^{-5}$ m$^2$ s$^{-1}$ using an ocean general circulation model (Noh and Lee, 2008). In this study, XLD is specified as the depth where ε drops from its first minimum value. This aligns with previous dissipation thresholds and ensures that mixing is captured independently of surface influences. The Upper (UTD) and Lower (LTD/LPD) Thermocline/Pycnocline Depth are delimited as defined by Assunçao et al (2020). UTD corresponded to the depth where the vertical temperature gradient $\partial\theta/\partial z = 0.1$ ℃ m$^{-1}$, while LTD/LPD were the last depth below the UTD at which $N^2 \geq 10^{-4}$ s$^{-2}$.

**Baroclinic currents**

To analyze the processes explaining dissipation and mixing, particularly along internal tidal (IT) paths, we estimate shear instabilities associated with the semi-diurnal (M2) ITs and mean circulation, as well as their contributions to mixing.

The M2 tidal component of the tidal current is derived by calculating the baroclinic (semi-diurnal) tidal velocity [$u''$, $v''$] (Fig. A3, Appendix), following these equations:

$$[u', v'] = [u, v] - [u_{bt}, v_{bt}], \tag{1}$$

$$[u_{bt}, v_{bt}] = \frac{1}{H} \int_{-H}^{0} [u, v]\,dz, \tag{2}$$

$$[u'', v''] = [u', v'] - [\overline{u'}, \overline{v'}]. \tag{3}$$

Here, [$u$, $v$] represent total horizontal velocities (Fig. A3, Appendix) obtained from SADCP data. The components [$u'$, $v'$] and [$u_{bt}$, $v_{bt}$] represent baroclinic and barotropic components of horizontal velocities, respectively (Fig. A3, Appendix). H is water depth. The baroclinic mean velocities [$\overline{u'}$, $\overline{v'}$] (Fig. A3, Appendix),

calculated to estimate mean circulation along IT paths, are decomposed into along-shore $\overline{u'}_l$ and cross-shore $\overline{u'}_c$ velocities. The overbar denotes the average over a M2 tidal period.

Note that continuously collected SADCP for some stations (e.g., S11) are not sufficiently resolved due to gaps filled by interpolating between time points. The similar processing are applied to the CTD-O$_2$ data collected alternately. SADCP time series data are less than 17 hours at all long stations, except for S14, which spans 42 hours. As a result, the diurnal and semidiurnal period fittings are not formally distinct (except at S14; Figs. A4 and A5, Appendix), and the inertial period (at least 5 days) cannot be resolved in our dataset. This limits our ability to separate currents by frequency and examine the associated dissipation.

The velocity profiles from LADCP are glued into our SADCP time series data below ~ 500 m depth at long stations.

To evaluate shear instabilities associated with ITs and the mean background circulation, we compute the baroclinic tidal vertical shear squared ($S^{2''}$) and mean shear squared ($\overline{S^{2'}}$) (Fig. A3, Appendix), as follows:

$$S^{2''} = (\partial u''/\partial z)^2 + (\partial v''/\partial z)^2, \tag{4}$$

$$\overline{S^{2'}} = (\partial \overline{u'}/\partial z)^2 + (\partial \overline{v'}/\partial z)^2. \tag{5}$$

To evaluate the impact of bottom friction on mixing, we calculate kinetic energy $\varepsilon_f = \frac{1}{2}\rho_s(u_f^2)$ near the bottom boundary layer at shallow stations using friction velocity $u_f = u_b\sqrt{C_d}$, where $C_d$=2.5 x 10$^{-3}$ is a drag coefficient obtained from the NEMO model. Huang et al. (2019) showed that the bottom boundary layer thickness spatially varies between 15-123 m in the Atlantic Ocean, with a median of ~ 30-40 m in the North Atlantic. We define bottom layer thicknesses in our study area based on measured bathymetry from CTD-O$_2$ and near-bottom currents from ADCP. Here, $u_b$ is the total velocity averaged over a thickness of 20 m above the seabed for shallow stations and 40 m for deep stations.

The individual contributions of semi-diurnal ITs and mean circulation are then expressed as follows: $E''/(\overline{E'} + E'')$ for tidal contribution and $\overline{E'}/(\overline{E'} + E'')$ for mean circulation contribution. Here, $E = N*S$. N is the buoyancy frequency and $S$ is vertical shear. S can be substituted by $S^{2''}$ and $\overline{S^{2'}}$.

**Ray tracing calculation**

Analyzing both the mean currents and the spatial dimension along the IT pathways offers another insight into the mechanisms responsible for observed mixing (Rainville and Pinkel, 2006). IT energy rays are generated in regions with steep topography, such as the shelf break, where IT slope matches with the bottom slope (i.e., critical slopes) before propagating within the ocean interior.. These rays, moving both downward and upward, encounter the seasonal pycnocline, resulting in beam scattering and the formation of large IT oscillations. As these oscillations steepen, they disintegrate into nonlinear ISWs, a process known as "local generation" of ISWs (New and Pingree, 1992). To explore IT paths, ray-tracing techniques are employed, as previously used by New and Da Silva (2002) and Muacho et al. (2014), to investigate the effectiveness and expected pathways of the IT beams off the Amazon shelf. One main assumption in our linear-theory-based hypothesis is that stratification

remains horizontally uniform along the IT propagation path, although in reality, it may vary due to submesoscale and mesoscale variability. This limitation makes the ray tracing approach less realistic but still useful as a first-order estimate of energy distribution. The IT ray-tracing calculation assumes that in a continuously stratified fluid, ITs energy can be described by characteristic pathways of beams (or rays) with a slope c to the horizontal:

$$c = \pm\left(\frac{\sigma^2 - f^2}{N^2 - \sigma^2}\right)^{1/2},$$ (6)

where $\sigma$ is the M2 tidal frequency ($1.4052 \times 10^{-4}$ rad s$^{-1}$), and f is the Coriolis parameter. $N^2$ are obtained from time-averaged AMAZOMIX CTD-O$_2$, glued with monthly $N^2$ profiles from Amazon36 (NEMO model outputs, 2012-2016) below 1000 m depth. Amazon36 is a NEMO configuration, specifically designed to cover the western tropical Atlantic from the mouth of the Amazon River to the open sea (see Tchilibou et al., 2022; Assene et al., 2024; for configuration details and model description). IT ray-tracing diagrams are performed along the transects. Seasonal sensitivity tests of rays (August, September, October, and April) are conducted by varying the critical slope positions and $N^2$ to explore its influence and generate a set of ray paths consistent with characteristics of IT pathways (Figs. A6 and A7, Appendix).

"

4. MacKinnon and Gregg (2003) provided validation in their paper. There needs to be some explanation of the advantages/conditions for this parameterization over those following from Gregg 1989. Gregg (2003) describe how latitude affects these mixing parameterizations. I saw no mention of this.

R: Thanks for your comments.

Please see the previous response to your comment n°7.

5. It is unclear how mixing layer depth is chosen. There is a statement about choosing a minimum but the profiles show a lot of variance.

R: Thanks for your remarks.

Mixed layer definitions are very sensitive depending on the criteria we adopt and the region of interest (e.g., Thomson and Fine, 2003; de Boyer Montégut et al., 2004; Cisewski et al., 2008; Noh and Lee 2008). We have performed a lot of tests and present them in the revision figure RC1.15.

In our study, we decided, since we have access to the direct microstructure profiles to calculate the mixing layer depth (XLD) defined as the depth at which dissipation estimates decrease from its first minimum value, ensuring that mixing is captured below the influence of surface dynamics. This definition of XLD is consistent with previous studies (e.g., Lozovatsky et al., 2006; Cisewski et al., 2008; Noh and Lee, 2008). As you mentioned, the dissipation profiles show a lot of variance, as does the XLD and the classical MLD (figure RC1.15). In the final manuscript we revised more carefully the definition of the XLD and the MLD and provided a new figure of comparison to show the sensitivity of all these layers.

We have revised the manuscript to clarify in section "methods/The vertical eddy diffusivity coefficient", in lines 153–176 of the revised manuscript and reported here:

"

**The vertical eddy diffusivity coefficient**

The efficiency of turbulence in redistributing energy is assessed through the calculation of the vertical eddy diffusivity coefficient ($K_z$). This coefficient is particularly significant in regions such as pycnoclines, where stratification suppresses mixing, making turbulence-driven mixing a key mechanism for vertical energy transport (Thorpe, 2007).

$K_z$ is calculated from $\varepsilon$ following the formulation of Osborn (1980), given by $K_z = \varepsilon\ \Gamma\ N^{-2}$. Here, $N^2$ is the buoyancy frequency squared, which is calculated using the sorted potential density profiles ($\sigma_\theta$) obtained from CTD-O$_2$ data. It is given by $N^2 = -(g/\rho_0)\ (d\sigma_\theta/dz)$, where $\rho_0$ is a reference density (1025 kg m$^{-3}$) and g is the gravitational acceleration. $\Gamma$ is the mixing efficiency, defined as the ratio between the buoyancy flux and the energy dissipation, and is typically set to 0.2, which corresponds to the critical Richardson number Ri = 0.17 (Osborn, 1980). $\varepsilon$ is linearly interpolated into the depths of $N^2$.

Turbulence within the pycnocline can reduce stratification and increase vertical eddy diffusivity below the mixing layer (Thorpe, 2007). Subsurface mixing, driven by the breaking of ITs and shear instabilities, plays a particularly important role below the mixed layer, especially in equatorial waters (Gregg et al., 2003).

There are several criteria for defining the Mixed Layer Depth (MLD). In this study, we use the commonly accepted density threshold criterion of 0.03 kg m$^{-3}$, as defined by de Boyer Montégut et al. (2004) and Sutherland et al. (2014), to estimate the MLD for each CTD-O$_2$ profile. Notably, comparisons with density thresholds of 0.01 and 0.02 kg m$^{-3}$ revealed no major differences in MLD across the AMAZOMIX stations and transects (Fig. A2, Appendix).

The miXing Layer Depth (XLD) is defined as the depth at which $\varepsilon$ decreases to a background level (Sutherland et al., 2014). Previous studies have applied various thresholds for background dissipation levels, such as such as 10$^{-8}$ and 10$^{-9}$ W kg$^{-1}$ in higher latitudes based on in situ observations (Sutherland et al., 2014; Lozovatsky et al., 2006; Cisewski et al., 2008; Brainerd and Gregg, 1995) and 10$^{-5}$ m$^2$ s$^{-1}$ using an ocean general circulation model (Noh and Lee, 2008). In this study, XLD is specified as the depth where $\varepsilon$ drops from its first minimum value. This aligns with previous dissipation thresholds and ensures that mixing is captured independently of surface influences. The Upper (UTD) and Lower (LTD/LPD) Thermocline/Pycnocline Depth are delimited as defined by Assunçao et al (2020). UTD corresponded to the depth where the vertical temperature gradient $\partial\theta/\partial z = 0.1\ ^\circ$C m$^{-1}$, while LTD/LPD were the last depth below the UTD at which $N^2 \geq 10^{-4}$ s$^{-2}$.

"

6. It is stated the mixing layer depth is always less than the mixed layer depth. This seems like it cannot be true by definition- you need mixing to make a mixed layer. Maybe the mixed layer depth criterion is too small. How about 0.1 kg m-3?

R: Thanks for your comments.

Indeed, you need mixing to make a mixed layer. There are several criteria for defining the mixed layer depth (MLD). We have tested different density threshold criterion from 0.01 to 0.03 kg m−3 , which are the commonly accepted density threshold used in previous studies (e.g., Thomson and

Fine, 2003; de Boyer Montégut et al., 2004; Cisewski et al., 2008; Noh and Lee 2008) as well the less used threshold and from 0.1 to 0.3 kg m−3 (e.g., Brainerd and Gregg, 1995). We have provided on figure RC1.15 and RC1.16, a calculation of these layers for each station. We can see that the XLD is typically deeper than the MLD, which is calculated using a density threshold 0.01-0.03 kg m-3, except for stations 8,10 and 25. This is consistent with regions exhibiting strong subsurface shear, such as the equatorial ocean and western boundary current areas (Noh and Lee, 2008). In contrast, XLD can be lesser than MLD during early restratification and at high latitudes during convective cooling (Noh and Lee, 2008). In our case, the exception for stations 8,10 and 25 (MLD>XLD) might be attributed to shelf break dynamics, including internal tides.

We have added the figure RC1.15 in Appendix of manuscript and pointed out in lines 261-263 and 461-464 of the revised manuscript and reported here:

Lines 261-263:

"

It is important to note that the XLD is typically deeper than the MLD at all stations (except at S8, S10, and S25), which is calculated using a density threshold 0.01, 0.02 or 0.03 kg m⁻³ (Fig. A2, Appendix).

"

Lines 461-464:

"

The XLD was found to be considerably larger than the MLD at all stations, except at S8, S10, and S25. This is consistent with regions exhibiting strong subsurface shear, such as the equatorial ocean and western boundary current areas (Noh and Lee, 2008). The exception observed at other stations may reflect larger mixing events that were not captured by the VMP measurements.

"

[Figure]

Figure RC1.15: *Comparison of Mixing Layer Depths (XLD, blue line) with Mixed Layer Depths (MLD) defined using (a) larger and (b) smaller density thresholds ($\Delta\varrho$). In panel (a), dotted, dashed, and solid red lines represent MLDs defined by $\Delta\varrho = 0.01, 0.02, 0.03$ kg m$^{-3}$, respectively. In panel (b), dotted, dashed, and solid magenta lines represent MLDs defined by $\Delta\varrho = 0.1, 0.2, 0.3$ kg m$^{-3}$, respectively.*

[Figure]

Figure RC1.16: Density profiles at selected stations (S8 to S14) overlaid MLD (in colored solid lines) and XLD (colored dashed lines). Distinct colors are used to represent each station.

7. Figure caption does not correspond to what is plotted.  Shear is missing?  Figure labelling: how about (a), (b), …  instead 4.a.1?  There are also faint grey lines around the figures.  All the dots in the shear vs N figures are hard to see.  Consider binning the results.

R: Thanks for your comments and suggestions.

We have reorganized and revised all the figures and captions (e.g., Figures 5, 6, and 7) in the revised manuscript to make them clearer. We also added the shear (e.g., Figure 6 and 7) in the revised manuscript and reported here in figure RC1.17 and RC1.18 for selected stations (e.g., S10, S11, S14).

[Figure]

Figure RC1.17: Semi-diurnal baroclinic vertical shear squared (S2", in m s-1, on a logarithmic scale) for stations (a) S10, (c) S11, and (e) S14. Panels (a), (c), and (e) also display the buoyancy frequency squared (N2, in s-2) represented by vertical black lines, potential density represented by grey contours, and dissipation rate profiles (ε, in W kg-1, on a logarithmic scale) represented by vertical colored bars. Mean baroclinic vertical shear squared (S2', in m s-1) for stations (b) S10, (d) S11, and (f) S14.

[Figure]

Figure RC1.18: *Dissipation rates (ε, in W kg$^{-1}$, on a logarithmic scale) below the XLD as a function of the buoyancy frequency squared (N$^2$, in s$^{-2}$, on a logarithmic scale) and semi-diurnal baroclinic vertical shear squared (S$^{2''}$, in m s$^{-1}$, on a logarithmic scale) for stations (a) S10, (c) S11, and (e) S14. Dissipation rates (ε, in W kg$^{-1}$, on a logarithmic scale) below the XLD as a function of mean baroclinic vertical shear squared ($\overline{S^{2'}}$, in m s$^{-1}$, on a logarithmic scale) and semi-diurnal baroclinic vertical shear squared (S$^{2''}$, in m s$^{-1}$, on a logarithmic scale) for stations (b) S10, (d) S11, and (f) S14. N$^2$ was linearly interpolated into the depths of S$^{2''}$ to have same vertical scales. Panels (a), (c), and (e) also display two solid black lines corresponding to Richardson*

*number Ri = 0.25 and Ri = 1, respectively. Dashed grey lines in panels (b), (d), and (f) are included for comparison purposes.*

8.  There are lots of seemingly correct facts and figures in the text but I am unclear what is the point of them.  Take section 3.2.2, for example.  Currents are going this way and that at various depths and locations.  True.  I would suggest a format for each paragraph as follows. "In this paragraph, we examine the flow patterns over the slope to show something [topic sentence]. Here are the relevant facts and skip ones not immediately related to the topic sentence.  In summary, we have shown flow is in this direction here which is important for something. Something is described in the next paragraph."

R: Thanks for your suggestions. All sections of manuscript have been revised.

**Comments by line number**

19 - twelve hours is the semidiurnal period but not the M2 period = 12.42 hrs

R: Thanks for your remarks. We have corrected it in the revised manuscript in line 19 of the revised manuscript, as shown below:

"

   During the AMAZOMIX survey (2021), currents, hydrography, and turbulence were measured over M2 tidal

   period (12.42h) at numerous sites near the Amazon outflow, where ITs are also generated along the slope.
".

129 - fft_length, etc.  Perhaps you could just give these a symbol if they come up more than once, e.g., L or n.  Otherwise skip the variable names in a manuscript.

R: Thanks for your remarks. We have corrected it in the revised manuscript in lines 181-187.

"

   Shear spectra are estimated using record lengths (L) and Fast Fourier Transform segments of 2 s, which are

   cosine windowed and overlapped by 50% (e.g., at station S6 and S10; Fig. 2b, and Fig. A1, Appendix).

   Additionally, vibration-coherent noise is removed. Different L and overlap (O) settings were selected and tested

   based on the environment (e.g., deep vs. shallow water), following Fer et al. (2024). For shallow stations, L (O)

   was shortened to 5 s (2.5 s), in contrast to the 8 s (4 s) used for deeper stations, due to evidence of overturns

   observed in AMAZOMIX acoustic measurements at deeper stations (Koch-Larrouy et al., 2024; in preparation).

   This adjustment helped to optimize the spatial resolution of dissipation estimates in shallow water stations.

"

136 - what is the high pass? and the low pass?

R: The high-pass filter was applied to the shear probe data and the low-pass filter was applied to the temperature gradients. We have rewritten this subsection "**TKE dissipation rates**" in the revised manuscript.

150 - mixing layer depth

R: Thanks for your remarks. We have corrected it in the revised manuscript in line 170 of the revised manuscript.

170 - I don't understand.  Please rephrase

R: Thanks for your remarks. We have rewritten it in line 198 of the revised manuscript, as shown below:

"

The velocity profiles from LADCP are glued into our SADCP time series data below ~ 500 m depth at long stations.

"

172  - what is H?

R:  Thanks for your remarks.

H is the water depth. We have revised it in the revised manuscript in lines 188-189 of the revised manuscript.

193 - references? Also strong internal tides propagating upward and impinging on the thermocline make ISW.  No need for bottom reflections.

R:Thanks for your remarks.

Indeed, strong internal tides propagating upward and impinging on the thermocline generate ISWs. We have revised the subsection "Ray Tracing Calculation" to clarify this, in lines 219–221 of the revised manuscript, as reported here:

"

These rays, moving both downward and upward, encounter the seasonal pycnocline, resulting in beam scattering and the formation of large IT oscillations. As these oscillations steepen, they disintegrate into nonlinear ISWs…

"

200- You have cross-shore measurements of N.  Are those not enough to make a horizontally varying N?

R: Thanks for your comments.

Indeed, we have cross-shore measurements of N. We have varied N2 profiles from stations (e.g., S10, S12, and S14) along transect Aa. Please see the results in figure RC1.19 and RC1.20 below. The sensitivity tests (Figure RC1.19) showed that ray paths align within the packets of rays observed when using mean N2 profiles along transect Aa at different times (e.g., in September and October; Figure RC1.20).

[Figure]

Figure RC1.19: Example of sensitivity tests with different cross-sectional measurements of $N^2$ along the transect T1 $N^2$. colors are used to distinguish different cross-shore measurements of $N^2$ for corresponding stations on T1. Topography steepness (gamma = ray slope / topography slope) for T1 using measured $N^2$ of S10. Gamma is illustrated by the colored bar (named gamma S10).

[Figure]

Figure RC1.20: Sensitivity tests of M2 IT ray-tracing along the transects Aa, conducted by varying the location of the critical topography slope. The tests use mean buoyancy frequency squared ($N^2$, in $s^{-2}$) obtained from CTD-O$_2$ data (September 2021) and NEMO-Amazon36 model data (2012-2016). Dashed colored lines represent IT beams calculated for different seasons (April, August, October, and September) and for varying locations of the critical topography slope. Grey areas indicate local topography. Panel also includes dissipation rate profiles ($\epsilon$, in W kg$^{-1}$, shown as vertical colored bars on a logarithmic scale) from

the VMP measurements. Subpanels within each panel illustrate the $N^2$ profiles derived from AMAZOMIX and the NEMO-Amazon36 model, which were used in the ray-tracing calculations. For comparison, sensitivity tests using $N^2$ measurements from individual stations along the corresponding transect (e.g., at S10) revealed similar ray paths (not shown), consistent with the packet of rays obtained using the mean $N^2$.

207- there has to be some minimal description of amazon36 here even if it is referenced

R: Thanks for your remarks.

We have added a short description of amazon36 in the revised manuscript (in lines 231-233) and refer to previous studies. We have reported the revision here:

"

Amazon36 is a NEMO configuration, specifically designed to cover the western tropical Atlantic from the mouth of the Amazon River to the open sea (see Tchilibou et al., 2022; Assene et al., 2024; for configuration details and model description).

"

210- sensitivity not sensibility test, although the latter could also be used

R: Thanks for your remarks. We have revised it in line 233 of the revised manuscript.

263- 3-5 not 03-05

R: Thanks for your remarks. We have revised itn lines 307-308 of revised manuscript, as shown below:

"

The baroclinic tidal velocities reveal a superposition of 3-5 tidal modes at IN-ITs stations (Figs. 5a, 5c, and 5e, and Figs. A9 to A15, Appendix).

"

267-272 - Shear obviously varies with N because if shear were bigger than it would have lower Ri and be unstable. It would be more instructive to consider Ri or reduced shear = S2 - 4N2.

R: Thanks for your comments.

Indeed, shear varies noticeably with $N$ at certain stations farther from IT generation sites. However, closer to the generation sites, such as at S10, stronger tidal shears were observed in areas where Ri was lower (<0.25), coinciding with hotspots of mixing.

This is why we specifically considered shear and quantified the contribution of tidal shear to mixing along IT paths. Please refer to the results for reduced shear presented below in figure RC1.21.

[Figure]

Figure RC1.21: For station S10. *Dissipation rates ($\varepsilon$, in W kg$^{-1}$, on a logarithmic scale) below the XLD as a function of the buoyancy frequency squared ($N^2$, in s$^{-2}$, on a logarithmic scale) and semi-diurnal baroclinic vertical shear squared ($S^{2''}$, in m s$^{-1}$, on a logarithmic scale), when using (a)-(b) reduced shear. Dissipation rates ($\varepsilon$, in W kg$^{-1}$, on a logarithmic scale) below the XLD as a function of mean baroclinic vertical shear squared ($\overline{S^{2'}}$, in m s$^{-1}$, on a logarithmic scale) and semi-diurnal baroclinic vertical shear squared ($S^{2''}$, in m s$^{-1}$, on a logarithmic scale), when using (c)-(d) no reduced shear. $N^2$ was linearly interpolated into the depths of $S^{2''}$ to have same vertical scales. Panels (a), (c), and (e) also display two solid black lines corresponding to Richardson number Ri = 0.25 and Ri = 1, respectively. Dashed grey lines in panels (b), (d), and (f) are included for comparison purposes.*

You could use some mean N or interpolated N and see what it looks like.

R: Indeed, several tests were conducted using both mean and interpolated $N$. Ultimately, we opted to use the time-interpolated $N$ and shear values at all dissipation times (at VMP measurement hours) along IT paths.

If the text talks a lot about shear, it would be helpful to plot shear. In fact the baroclinic vs tidal shear is discussed but there are no plots.

R: Thanks for your remarks. We have added the plots of the shears both in text and appendix.

Fig 4- caption and figure all on same page please. Add tick marks to the x axis to show when your CTD/VMP casts took place. Also this figure seems to include transits and time on station.

In Fig 4b1 there are some strange looking changes in the current. Perhaps it would be better to do your analysis station by station. Or are there artifacts of the SADCP processing going from on station to transit?

R: Thanks for your comments.

We have reorganized the figures and added an x-axis to indicate when CTD/VMP casts were conducted. Please refer to the updated figure in the revised manuscript (figure 5) and reported below in figure RC1.22.

Regarding the updated figure of the revised manuscript (figure 5), we noticed some unusual changes in the current patterns, which have been rechecked. The continuously collected SADCP for some stations (e.g., S11) are not sufficiently resolved due to gaps filled by interpolating between time points (e.g., between 13:30 and 15:30 at S11). Doing this interpolation would introduce potential artifacts in the calculations. The interpolation allowed us to extract some current profiles at the dissipation times to ensure consistency in the analyses.

The analyses were performed separately by station and by transect. For example, Figure RC1.22 shows the semidiurnal current over time at a fixed position for station S11. In contrast, Figure RC1.23 displays the mean alongshore currents along transect Aa, including transit between stations.

We have pointed out in text in lines 192-193 of revised manuscript and reported here

"

Note that continuously collected SADCP for some stations (e.g., S11) are not sufficiently resolved due to gaps filled by interpolating between time points. The similar processing are applied to the CTD-O$_2$ data collected alternately.

"

[Figure]

Figure RC1.22: Semi-diurnal baroclinic zonal currents ($u$", in m s-1) from the ADCP for stations (a) S10, (c) S11, and (e) S14. Panels (a), (c), and (e) also display the buoyancy frequency squared (N2, in s-2) represented by vertical black lines, potential density represented by grey contours, and dissipation rate profiles ($\epsilon$, in W kg-1, on a logarithmic scale) represented by vertical colored bars. Along-shore mean baroclinic currents (u'l, in m s-1) from the ADCP for stations (b) S10, (d) S11, and (f) S14.

[Figure]

Figure RC1.23: IT ray-tracing diagrams for the M2 tidal constituent along transects Aa. The calculations were performed using the mean buoyancy frequency squared ($N^2$, in $s^{-2}$) obtained from CTD-$O_2$ data (ray in red) and NEMO-Amazon36 model data (ray in blue) for September. Grey areas represent local topography and black circles indicate the critical topography slope (ray generation sites). Panel also show along the transects Aa: along-shore mean total currents ($u_l$, in $m\ s^{-1}$) from ADCP (Dashed black lines), potential density from CTD-$O_2$ (grey contours), and dissipation rate profiles ($\varepsilon$, in $W\ kg^{-1}$, on a logarithmic scale) from the VMP (vertical colored bars). Subpanels within each panel illustrate the $N^2$ profiles from AMAZOMIX (red line) and the NEMO-Amazon36 model (blue line) used for ray-tracing calculations. Upper Thermocline Depth (UTD, dotted lines) and Lower Thermocline/Pycnocline Depth (LTD/LPD, dashed lines) are also indicated.

301- what sort of of mean? time, depth, etc

R: Thanks for your remarks. It is time-mean. We have revised it in line 362 of the revised manuscript

312-314- More explanation is needed here.

R: These have been removed during the revision of the manuscript.

Fig 5- the tidal rays look to have shallower slope than the topography. It would be helpful to plot ray slope vs topographic slope to verify. The dissipation seems unrelated to the ray paths as far as I can tell. Perhaps it is not 2D. There appear to be 2 sources at least, which can constructively interfere. Rays may propagate at an angle to the coast.

R: Thanks for your comments.

We attempted to verify our analysis by plotting the ray slope against the topographic slope. Using our bathymetry data from the NEMO-AMAZON36 model or GEBCO, the tidal rays did not appear to have a shallower slope (around 225 km) than the topography in any of the sensitivity tests, as illustrated in Figure RC1.24 (with topography steepness "gamma") below. These results may differ if using alternative bathymetric products.

Following the ray tracing, we observed that some dissipation hotspots are located along the ray paths, particularly above the generation sites and during ray propagation.

Interestingly, there seem to be at least two sources that could constructively interfere, potentially explaining the high dissipation observed at S14. This wave-wave interference is being explored further in a separate study to confirm these findings.

[Figure]

Figure RC1.24: Example of sensitivity tests with different cross-sectional measurements of $N^2$ along the transect T1 $N^2$. colors are used to distinguish different cross-shore measurements of $N^2$ for corresponding stations on T1. Topography steepness (gamma = ray slope / topography slope) for T1 using measured $N^2$ of S10. Gamma is illustrated by the colored bar (named gamma S10).

384 - "for the first time"  How sure are you of this?  It would be conservative to add: to the best of our knowledge

R: Thanks for your remarks.

We have corrected the sentence of the text in line 436 of the revised manuscript, as follows:

"

To the best of our knowledge, the AMAZOMIX 2021 cruise provided, for the first time, direct measurements of turbulent dissipation using a velocity microstructure profiler….

"

---

## Author Comment (AC2)

**We thank this Reviewer for thoughtful and constructive comments on our manuscript. We appreciate the time s/he invested in the review. We believe that our revised manuscript addresses all the comments. In this regard, we have revised and rewritten a few sections such as Abstract, method, results, discussion and conclusion in the revised manuscript. Quality control and revision of dissipation estimates, as well as current calculations were also done to ensure the validity of our results. We thought it useful to point out its detailed revisions (lines and sections) in the reply to your comments. Below (highlighted in blue and magenta) is an itemized response to the different issues raised in the review.**

Review of "Turbulent dissipation from AMAZOMIX off the Amazon shelf along internal tides paths", by Kouogang et al. Submitted to EGUsphere.

By Kouogang et al.

The paper reports oceanographic observations in the tropical Atlantic ocean within the framework of project AMAZOMIX, which aims to quantify mixing, identify the associated processes, and investigate their impact on nutrient fluxes off the Amazon shelf. The observations were collected during a field campaign in August-October 2021, on board the RV ANTEA with a dedicated satellite coverage (reported elsewhere). The region is well known for large amplitude internal tides and Internal Solitary Waves (ISWs), that were first measured and reported in Brandt et al., 2002, with wave amplitudes exceeding 100 m. Those large amplitude ISWs are clearly seen in Synthetic Aperture Radar (SAR) satellite images, as illustrated in the author's Figure 1. Clearly, the cruise measurements were carefully planned in advance based on the knowledge provided by the SAR image archive. The most relevant aspect reported in the paper is the relative increase of mixing found about 225 km away from two distinct generation sites of internal tidal waves at the shelf break. The site where the relative mixing increases coincides with the surfacing of Internal Tidal (IT) rays emanating from different sites at the shelf break. But it also coincides with the appearance of large amplitude ISWs seen in the satellite record. While the constructive interference of IT rays has been suggested, and indeed is consistent with ocean colour images, in the central region of the Bay of Biscay (Western Europe), this aspect has not been documented elsewhere until now. It is therefore remarkable the association made by the authors between enhanced mixing and IT interference.

Thanks for this very concise and precise summary. We took the liberty of using some of your wording to improve our abstract and our conclusion.

However, a more comprehensive comment should be made concerning the presence of ISWs near Station S14, and its possible relation with the measured increased mixing.

R: This is a very interesting question. We have a specific point on the discussion (in lines 551-569 of the revised manuscript) on that matter, and we agree that this funding is quite important. Please find the revised section "**Discussion on the strong mixing at S14**" below:

"

**Discussion on the strong mixing at S14**

Along the IT paths, elevated remote dissipation rates (within $[10^{-7}, 10^{-6}]$ W kg$^{-1}$) were identified $\sim 230$ km from the shelf-break at S14.

This region is well known for intense IT dissipation, as shown by a realistic model (Tchilibou et al., 2022; Assene et al., 2024), and for the highest occurrences of ISWs generated by ITs (Fig. 1a; de Macedo et al., 2023), with large-amplitude ISWs exceeding 100 m clearly visible in satellite records (Brandt et al., 2002).

At station S14, where relative mixing increases, IT rays surfacing from two distinct IT generation sites coincide with the appearance of ISWs and mark the location where the NBC vanishes.

This region of wave-wave interactions can lead to the constructive interference of IT rays, potentially facilitating the emergence of higher tidal modes (New & Pingree, 1992; Silva et al., 2015; Barbot et al., 2022; Solano et al., 2023). These higher modes, in turn, could enhance the generation of nonlinear ISWs (e.g., Jackson et al., 2012) and contribute to the elevated dissipation rates (Xie et al., 2013), as observed at this station.

Moreover, IT interactions with baroclinic eddies may also contribute to turbulent dissipation (Booth and Kamenkovich, 2008), particularly in this area of pronounced eddy activity. However, no repeated AMAZOMIX stations observed during a tidal period were enclosed by mesoscale eddy activity, except potentially around site E, where possible evidence of a subsurface eddy was detected at S21 (Dossa et al., 2024, in preparation).

Future studies are needed to unravel the intricate interplay among these processes. The data collected during the AMAZOMIX cruise will provide a guide for improving our understanding and advancing parameterizations for modeling studies.

"

This question is so interesting that we are dealing with an additional paper. Indeed, AMAZOMIX provides acoustic data that clearly shows ISW and overturn associated with it at station 14, and a paper is in construction on this matter. Furthermore, their signature at the surface was studied in Macedo et al., 2023 in the region. The mixing hotspots observed at this station could be related to their presence, but more work is needed to establish it with more confidence

The authors calculate the turbulent kinetic energy (TKE) dissipation rates and vertical diffusivities making use of microstructure and hydrography data collected at a series of stations and transects in a vast study region, being able to compare these values within internal tide pathways, and away from those paths. The observations also reveal a great deal of mixing at the shelf break, as expected. **Baroclinic shear currents were calculated from ADCP data collected between stations and transects. It was then possible to estimate the ratio contribution to mixing between IT and Baroclinic Currents (BC). Interestingly, ITs dominate the ratio for IT/BC near the generation sites of ITs at the shelf break (65%/35%) and also at Station S14, 225 km away from the shelf break, admittedly where the IT beams coming from different sites may interfere (60%/40%).** Although this fact is suggestive of the IT rays interference being capable of impacting mixing well away from the shelf break, the authors do not attempt comparison with the impact ISWs alone would have, on mixing.

R: Thank you for your comment.

Indeed, ISWs may influence mixing. While our study did not quantify the specific contribution of ISWs, we suggest that their presence, likely resulting from IT decay, could play a role in the mixing process at S14. In order to explore further the difference between ITs mixing compared to ISWs mixing a non hydrostatic model at very fine resolution should be runned such as in da Silva et al. (2015) and Solano et al. (2023). It would be very interesting to compare the AMAZOMIX data to such model.

The manuscript is reasonably written (with some minor pitfalls indicated below), and the data analysis certainly merits publication at EGUsphere. The hypotheses tested in the paper are well explained and presented, representing a milestone in the oceanography of the study region. The paper also includes an extensive appendix with auxiliary results. This reviewer recommends publication after some minor corrections (please see list below).

Minor comments:

Introduction:

Line 46 – 47: "They can dissipate where the energy beam reflects at the bottom, at the surface or at the thermocline levels…"; The authors start describing bottom reflection of IT beams etc. without first explaining how those beams form and propagate in the vertical. A reader may like to read first how these beams are formed and why they propagate following rays that are determined by the well known formula that gives the angle to the horizontal. There should be a more comprehensive explanation about tidal beams before this point in the text. It may be worth referring to the work of Theo Gerkema (Gerkema, 2001 and Gerkema and Zimmerman, 2008).

R: Thanks for your comments and suggestions. We have rewritten to make it clearer in the text as you suggested. The revisions can be found in lines 34-82, as shown below:

"

**1 Introduction**

Turbulent mixing in the ocean plays an important role in sustaining the thermohaline and meridional overturning circulation and in closing the global ocean energy budget (Kunze, 2017). These processes have strong implications for the climate, influencing heat and carbon transport, as well as nutrients supply for photosynthesis (Huthnance, 1995; Munk and Wunsch, 1998). Mixing processes can result from wind in the surface waters layer, internal waves and shear instability in the ocean interior, and bottom friction near the bottom layer (Miles, 1961; Thorpe, 2018; Ivey et al., 2020; Inall et al., 2021). Barotropic tides interacting with steep shelf-break topography trigger internal waves at tidal frequencies and harmonics, known as internal tides (ITs), which can propagate and produce mixing. These ITs can be expressed by large vertical displacements (up to tens of meters) of water masses (Garrett and Kunze, 2007). After their generation on the shelf-break, the (more unstable) higher modes of ITs may dissipate locally, while the lower modes can propagate far away (Zhao et al., 2016). IT beams (generated where the slope of the ITs and the topography match together on the shelf-break) can propagate vertically, resulting in reflection, scattering and dissipation of ITs at the bottom, surface waters, or thermocline

levels (New and Da Silva, 2002; Gerkema and Zimmerman, 2008; Bordois, 2015; Zhao et al. 2016). They can also dissipate when energy fluxes interfere (Zhao et al., 2012) or interact with strong baroclinic eddies or currents (Rainville and Pinkel, 2006; Whalen et al., 2012). Furthermore, ITs may disintegrate into packets of higher-mode nonlinear internal solitary waves (ISWs), which can propagate and dissipate offshore (Jackson et al., 2012).

Previous and recent studies have shown that ITs-induced turbulent mixing can affect the surface, such as sea surface temperature (Ray and Susanto, 2016; Nugroho et al., 2018; Assene et al., 2024), chlorophyll content (Muacho et al., 2014; M'Hamdi et al., 2024; in preparation), marine ecosystems (Wang et al., 2007; Zaron et al., 2023), and atmospheric convection and the rainfall structure (Koch-Larrouy et al., 2010, Sprintall et al. 2014).

In the western tropical Atlantic, the Amazon River-Ocean Continuum (AROC) constitutes a key region of the global oceanic and climate system (Araujo et al., 2017; Varona et al., 2018). This region (Fig. 1a) is characterized by a system of western boundary currents, including North Brazil Current (NBC). NBC, which flows northwestward, has its core velocities ($\sim 1.2$ m s$^{-1}$) that remain stable from the surface to a depth of 100 m (Johns et al., 1998; Bourlès et al., 1999; Barnier et al., 2001; Neto and Silva, 2014). This region also experiences highly variable dynamics due to the Amazon River Plume. During the rainy season (May-July), peak discharge can extend the plume over 1500 km offshore, northwest along the NBC. In the dry season (September-November), reduced discharge and stronger saline intrusion may confine the plume to less than 500 km offshore, near the Amazon Shelf, with some eastward dispersion (Coles et al., 2013). The Amazon plume can generate vertical shear in underlying currents, enhancing mixing. Additionally, a system of Amazonian Lenses of water (AWL), influenced by continental inputs, may affect both the boundary layer and mixed layer patterns (Silva et al., 2005; Prestes et al., 2018).

In the AROC region, the Amazon shelf-break is a hotspot for the generation, propagation and dissipation of ITs and ISWs as a result of non-linear processes (Geyer, 1995; Brandt et al., 2002; Magalhães et al., 2016; Ruault et al., 2020; Tchilibou et al., 2022; Fig 1). Previous studies using Synthetic Aperture Radar (SAR) satellite images (Magalhaes et al., 2016) identified ISWs along the path of ITs propagating from two sites (i.e., sites Aa and Ab; Fig. 1a). Conversely, other sites showed no ISWs propagation (i.e., sites E and D; Fig. 1a, 1b and 1c) (see Magalhaes et al., 2016 for definition). Using numerical modeling, Tchilibou et al. (2022) showed that about 30 % of the M2 (dominant tidal component; Le Bars et al. 2010) ITs energy is dissipated locally (for higher-modes ITs) at sites E, Aa, Ab and D (Fig. 1a), while the remaining lower-modes ITs energy can be dissipated remotely. Dissipation away from the generation sites (E, Aa, Ab and D; Fig. 1a) can result from the shear instabilities caused by ITs-ITs and/or ITseddy/current interactions. Despite the presence of ITs, no direct measurements of dissipation rates have been conducted to our knowledge.

The mixing induced by these internal waves in the region was observed during the AMAZOMIX cruise (Bertrand et al., 2021). The cruise was designed with stations/transects inside and outside ITs fields (Fig. 1a and 1c) to measure ITs dissipation and study their impact on the AROC ecosystem. Direct microstructure measurements of temperature, salinity and velocity were conducted at the different repeated stations/transects over a M2 tidal cycle ($\sim 12.42$ h). These cruise measurements offer an opportunity to explore whether ITs play a role in mixing within the AROC region. In this study, we will quantify mixing and identify the associated processes off the Amazon shelf. We will calculate turbulent kinetic energy (TKE) dissipation rates, vertical displacements of isopycnal surfaces and vertical eddy diffusivities using in situ microstructure and hydrography

data. Finally, the baroclinic shear of currents and their contributions to mixing will be calculated from current data collected between stations and transects.

[Figure]

Figure 1: a) Map of a part of the AMAZOMIX 2021 cruise off the Amazon shelf, showing bathymetric contours (100 m, 750 m, 2000 m, and 3000 m isobaths) in gray. Colored circles and stars indicate short and long CTD-O2/L-S-ADCP stations, respectively, with the corresponding sampling dates represented by the color bar. Solid black lines depict SADCP transects (for Aa, Ab, D, G, and E). Magenta arrows show the 25-hour mean

*depth-integrated baroclinic IT energy flux (September 2015, from the NEMO model) originating from IT generation sites (Aa, Ab, D, and E) along the shelf break. The solid brown line represents the NBC pathways illustrating background circulation. Shattered colored lines highlight ISW signatures. b) 1A Sentinel image acquired on 12th September 2021, showing ISW signatures. c) Tidal range at AMAZOMIX stations, with ISW signature dates marked by red bars.*

"

Line 53: The reference Muacho et al, 2014 should be referred earlier, together with M'Hamdi et al., 2024, in preparation)

R: Thanks for your suggestions. We have rewritten to make it clearer in the text as you suggested. The revisions can be found in lines 51-52 of the revised manuscript, as shown in your previous comment.

Line 62: Please include reference Brandt et al., 2002 in JPO about ISW in situ observations.

R: we could not find this reference in JPO. Could you please share the link to this article, if you have it ?.

Line 69: "…no direct measurements of dissipation rates have been conducted.". Do you mean before the present paper?

R: We have corrected the sentence of the text in lines 72-73 of the revised manuscript. I meant that:
"
Despite the presence of ITs, no direct measurements of dissipation rates have been conducted to our knowledge.
"

Methods:

Line 170: "…time series with LADCP profiles glued below ~500m…". later, in line 209 (page 8), the word "stitched" is used with a similar meaning? Could the authors use jargon more carefully, and consistently, please.

R: Thanks for your remarks. We have rewritten to make it clearer as you suggested in lines 122-235 of the revised manuscript. The revision of section "methods" can be found below:

"
**2.2 Methods**

**TKE dissipation rates**
The VMP data are processed using ODAS Matlab library (developed by Rockland Scientific International, Inc) to infer the TKE dissipation rate (ε). The processing methods for the VMP data are briefly described here and adhere to the recommendations of ATOMIX (Analyzing ocean turbulence observations to quantify mixing), as

reported by Lueck et al. (2024), and have been validated against the benchmark estimates (presented in Fer et al., 2024).

First, the VMP data are converted into physical shear units, and the time series are prepared. Continuous sections of the time series are selected for dissipation estimation. Before spectral estimation, the aberrant shear signals caused by vessel wake contamination are removed. Collisions of the shear probe with plankton and other particles are removed using the de-spiking routine. The records from each section are then high-pass filtered (e.g., at station S6 and S10; Fig. 2a, and Fig. A1, Appendix).

Shear spectra are estimated using record lengths (L) and Fast Fourier Transform segments of 2 s, which are cosine windowed and overlapped by 50% (e.g., at station S6 and S10; Fig. 2b, and Fig. A1, Appendix). Additionally, vibration-coherent noise is removed. Different L and overlap (O) settings were selected and tested based on the environment (e.g., deep vs. shallow water), following Fer et al. (2024). For shallow stations, L (O) was shortened to 5 s (2.5 s), in contrast to the 8 s (4 s) used for deeper stations, due to evidence of overturns observed in AMAZOMIX acoustic measurements at deeper stations (Koch-Larrouy et al., 2024; in preparation). This adjustment helped to optimize the spatial resolution of dissipation estimates in shallow water stations.

Finally, $\varepsilon$ is determined using the spectral integration method and by comparison with the Nasmyth empirical spectrum (Nasmyth, 1970). Quality assurance tests are carried out in accordance with ATOMIX's recommendations (Lueck et al., 2024). A figure of merit < 1.4 is used to exclude bad data (e.g., at station S6 and S10; Fig. 2b, and Fig. A1, Appendix), and the fraction of data affected by de-spiking is < 0.05.

[Figure]

*Figure 2: Example of wavenumber spectra from a dissipation structure segment used to determine the dissipation rate at station S6 at a pressure of 87.9 dBar. (a) Cleaned and high-pass filtered signals from shear probe 1 (blue) and shear probe 2 (red, offset by 5 s⁻¹). (b) Wavenumber spectra for shear probes 1 and 2. Thick lines (blue for probe 1, red for probe 2) show shear spectra with coherent noise correction, while thin lines (sky blue for probe 1, orange for probe 2) show spectra without correction. Triangles mark the maximum wavenumber used for dissipation rate estimation. Black lines represent Nasmyth reference spectra for estimated dissipation rate of 3.8 x 10⁻⁸ W kg⁻¹ for both shear probes. Dissipation rate estimates for shear probe 1 and shear probe 2 at a pressure of 87.9 dBar yielded a figure of merit of 0.93 and 0.94, respectively.*

**The vertical eddy diffusivity coefficient**

The efficiency of turbulence in redistributing energy is assessed through the calculation of the vertical eddy diffusivity coefficient ($K_z$). This coefficient is particularly significant in regions such as pycnoclines, where stratification suppresses mixing, making turbulence-driven mixing a key mechanism for vertical energy transport (Thorpe, 2007).

$K_z$ is calculated from ε following the formulation of Osborn (1980), given by $K_z = \varepsilon\, \Gamma\, N^{-2}$. Here, $N^2$ is the buoyancy frequency squared, which is calculated using the sorted potential density profiles ($\sigma_\theta$) obtained from CTD-O$_2$ data. It is given by $N^2 = -\,(g/\rho_0)\,(d\sigma_\theta/dz)$, where $\rho_0$ is a reference density (1025 kg m$^{-3}$) and g is the gravitational acceleration. $\Gamma$ is the mixing efficiency, defined as the ratio between the buoyancy flux and the energy dissipation, and is typically set to 0.2, which corresponds to the critical Richardson number Ri = 0.17 (Osborn, 1980). ε is linearly interpolated into the depths of $N^2$.

Turbulence within the pycnocline can reduce stratification and increase vertical eddy diffusivity below the mixing layer (Thorpe, 2007). Subsurface mixing, driven by the breaking of ITs and shear instabilities, plays a particularly important role below the mixed layer, especially in equatorial waters (Gregg et al., 2003).

There are several criteria for defining the Mixed Layer Depth (MLD). In this study, we use the commonly accepted density threshold criterion of 0.03 kg m$^{-3}$, as defined by de Boyer Montégut et al. (2004) and Sutherland et al. (2014), to estimate the MLD for each CTD-O$_2$ profile. Notably, comparisons with density thresholds of 0.01 and 0.02 kg m$^{-3}$ revealed no major differences in MLD across the AMAZOMIX stations and transects (Fig. A2, Appendix).

The miXing Layer Depth (XLD) is defined as the depth at which ε decreases to a background level (Sutherland et al., 2014). Previous studies have applied various thresholds for background dissipation levels, such as such as 10$^{-8}$ and 10$^{-9}$ W kg$^{-1}$ in higher latitudes based on in situ observations (Sutherland et al., 2014; Lozovatsky et al., 2006; Cisewski et al., 2008; Brainerd and Gregg, 1995) and 10$^{-5}$ m$^2$ s$^{-1}$ using an ocean general circulation model (Noh and Lee, 2008). In this study, XLD is specified as the depth where ε drops from its first minimum value. This aligns with previous dissipation thresholds and ensures that mixing is captured independently of surface influences. The Upper (UTD) and Lower (LTD/LPD) Thermocline/Pycnocline Depth are delimited as defined by Assunçao et al (2020). UTD corresponded to the depth where the vertical temperature gradient $\partial\theta/\partial z = 0.1$ ℃ m$^{-1}$, while LTD/LPD were the last depth below the UTD at which $N^2 \geq 10^{-4}$ s$^{-2}$.

**Baroclinic currents**

To analyze the processes explaining dissipation and mixing, particularly along internal tidal (IT) paths, we estimate shear instabilities associated with the semi-diurnal (M2) ITs and mean circulation, as well as their contributions to mixing.

The M2 tidal component of the tidal current is derived by calculating the baroclinic (semi-diurnal) tidal velocity [$u''$, $v''$] (Fig. A3, Appendix), following these equations:

$$[u',\, v'] = [u,\, v] - [u_{bt},\, v_{bt}], \tag{1}$$

$$[u_{bt},\, v_{bt}] = \frac{1}{H}\int_{-H}^{0}[u,\, v]\,dz, \tag{2}$$

$$[u'',\, v''] = [u',\, v'] - [\overline{u'},\, \overline{v'}]. \tag{3}$$

Here, [$u$, $v$] represent total horizontal velocities (Fig. A3, Appendix) obtained from SADCP data. The components [$u'$, $v'$] and [$u_{bt}$, $v_{bt}$] represent baroclinic and barotropic components of horizontal velocities, respectively (Fig. A3, Appendix). H is water depth. The baroclinic mean velocities [$\overline{u'}$, $\overline{v'}$] (Fig. A3, Appendix),

calculated to estimate mean circulation along IT paths, are decomposed into along-shore $\overline{u'}_l$ and cross-shore $\overline{u'}_c$ velocities. The overbar denotes the average over a M2 tidal period.

Note that continuously collected SADCP for some stations (e.g., S11) are not sufficiently resolved due to gaps filled by interpolating between time points. The similar processing are applied to the CTD-O$_2$ data collected alternately. SADCP time series data are less than 17 hours at all long stations, except for S14, which spans 42 hours. As a result, the diurnal and semidiurnal period fittings are not formally distinct (except at S14; Figs. A4 and A5, Appendix), and the inertial period (at least 5 days) cannot be resolved in our dataset. This limits our ability to separate currents by frequency and examine the associated dissipation.

The velocity profiles from LADCP are glued into our SADCP time series data below ~ 500 m depth at long stations.

To evaluate shear instabilities associated with ITs and the mean background circulation, we compute the baroclinic tidal vertical shear squared ($S^{2''}$) and mean shear squared ($\overline{S^{2'}}$) (Fig. A3, Appendix), as follows:

$$S^{2''} = (\partial u''/\partial z)^2 + (\partial v''/\partial z)^2, \tag{4}$$

$$\overline{S^{2'}} = (\partial \overline{u'}/\partial z)^2 + (\partial \overline{v'}/\partial z)^2. \tag{5}$$

To evaluate the impact of bottom friction on mixing, we calculate kinetic energy $\varepsilon_f = \frac{1}{2}\rho_s(u_f^2)$ near the bottom boundary layer at shallow stations using friction velocity $u_f = u_b\sqrt{C_d}$, where $C_d$=2.5 x 10$^{-3}$ is a drag coefficient obtained from the NEMO model. Huang et al. (2019) showed that the bottom boundary layer thickness spatially varies between 15-123 m in the Atlantic Ocean, with a median of ~ 30-40 m in the North Atlantic. We define bottom layer thicknesses in our study area based on measured bathymetry from CTD-O$_2$ and near-bottom currents from ADCP. Here, $u_b$ is the total velocity averaged over a thickness of 20 m above the seabed for shallow stations and 40 m for deep stations.

The individual contributions of semi-diurnal ITs and mean circulation are then expressed as follows: $E''/(\overline{E'} + E'')$ for tidal contribution and $\overline{E'}/(\overline{E'} + E'')$ for mean circulation contribution. Here, $E = N*S$. N is the buoyancy frequency and $S$ is vertical shear. S can be substituted by $S^{2''}$ and $\overline{S^{2'}}$.

**Ray tracing calculation**

Analyzing both the mean currents and the spatial dimension along the IT pathways offers another insight into the mechanisms responsible for observed mixing (Rainville and Pinkel, 2006). IT energy rays are generated in regions with steep topography, such as the shelf break, where IT slope matches with the bottom slope (i.e., critical slopes) before propagating within the ocean interior. These rays, moving both downward and upward, encounter the seasonal pycnocline, resulting in beam scattering and the formation of large IT oscillations. As these oscillations steepen, they disintegrate into nonlinear ISWs, a process known as "local generation" of ISWs (New and Pingree, 1992). To explore IT paths, ray-tracing techniques are employed, as previously used by New and Da Silva (2002) and Muacho et al. (2014), to investigate the effectiveness and expected pathways of the IT beams off the Amazon shelf. One main assumption in our linear-theory-based hypothesis is that stratification

remains horizontally uniform along the IT propagation path, although in reality, it may vary due to submesoscale and mesoscale variability. This limitation makes the ray tracing approach less realistic but still useful as a first-order estimate of energy distribution. The IT ray-tracing calculation assumes that in a continuously stratified fluid, ITs energy can be described by characteristic pathways of beams (or rays) with a slope c to the horizontal:

$$c = \pm\left(\frac{\sigma^2 - f^2}{N^2 - \sigma^2}\right)^{1/2}, \tag{6}$$

where $\sigma$ is the M2 tidal frequency ($1.4052 \times 10^{-4}$ rad s$^{-1}$), and f is the Coriolis parameter. $N^2$ are obtained from time-averaged AMAZOMIX CTD-O$_2$, glued with monthly $N^2$ profiles from Amazon36 (NEMO model outputs, 2012-2016) below 1000 m depth. Amazon36 is a NEMO configuration, specifically designed to cover the western tropical Atlantic from the mouth of the Amazon River to the open sea (see Tchilibou et al., 2022; Assene et al., 2024; for configuration details and model description). IT ray-tracing diagrams are performed along the transects. Seasonal sensitivity tests of rays (August, September, October, and April) are conducted by varying the critical slope positions and $N^2$ to explore its influence and generate a set of ray paths consistent with characteristics of IT pathways (Figs. A6 and A7, Appendix).
"

Line 174: "… with overbar the average…". A word seems to be missing between "overbar" and "the average".

R: Thanks for your remarks. We have rewritten it in line 190 of the revised manuscript, as shown in your previous comment.

Line 183: "… for M2 semi-diurnal baroclinic energy)…". The parenthesis after the word "energy" appears to be unnecessary!
R: Thanks for your remarks. We have rewritten the subsection "Baroclinic currents" in lines 179-213 of the revised manuscript, as shown in your previous comment.

Line 187: "Previous study of Huang et al. (2019) shown…". Please revise grammar.

R: Thanks for your remarks. We have revised grammar in lines 207-208 of the revised manuscript, as shown in your previous comment.

Line 188: "… between 15-123 m in Ocean Atlantic". Do you mean "Atlantic Ocean"?

R: Yes, thanks for your remarks. We have rewritten it in line 207 of the revised manuscript, as shown in your previous comment.

Line 209: " and stitched…". Please be consistent with jargon language.

R:  Thanks for your remarks. We have rewritten it in line 198 of the revised manuscript, as shown in previous replies.

Results:

Line 218: "… . They are thicker along the IN-ITs…". If you are referring to the "step-like" structures from the previous line, consider using another word for "thicker", e.g. "larger".

R: Thanks for your remarks. We have rewritten it in lines 239-252 of the revised manuscript. The revision of section "Results" can be found below:

"

**3.1.1 Thermohaline and IT features**

In this subsection, we analyze the density profiles to gain insight into mixing processes and/or wave propagation. Step-like features are observed in the density profiles (Figs. 3a and 3b). During the M2 tidal period, step-like structures ~20-40 m in length occur at depths ranging from 80 to 160 m at stations S10, S12, S13, and S14 (Fig. 3a). These features are more pronounced along the IN-ITs transect Aa and Ab compared to the other transects (e.g., E and G; Figs. A8.a and A8.b, Appendix).

In this layer (between 60 and 170 m depth), significant vertical displacements, ranging from 20 to 60 m, are detected along transects Aa, Ab, and E (e.g., 40 m at S10, 48 m at S6, 52 m at S13, and 32 m at S14; Figs. 3a and 3b). The smallest displacements (~8 m at S25) are observed along the OUT-ITs transect G (Fig. A8.b, Appendix). These vertical displacements are also evident in the variability of the mixed layer depth (MLD), which fluctuates between 18 and 84 m over a semi-diurnal cycle (figure not shown).

In conclusion, the presence of step-like structures and isopycnal displacements suggests strong mixing in the water column, and supports the hypothesis of ITs propagating, with stronger energy along transects Aa and Ab, weaker energy along D and E, and almost absent along G (Fig. 1a).

"

In Figure 3 the choice of colours for dissipation rates (epsilon) is somewhat difficult to grasp. The colours "red" and "magenta" are sometimes confusing, particularly in the vertical profiles in Figs. 3b to 3e. Could the "red" colour be changed to a "light Orange", or similar?

R: Thank you for your comments and suggestions. The light orange color was already used in Figure 3 (as shown below for example) and is sometimes referenced in subsequent figures for analysis. We have changed the magenta color to purple to enhance visual clarity and make it more noticeable.

[Figure]

Figure RC3.1: *(b) Density profiles ($\sigma_\theta$, kg m$^{-3}$) obtained from CTD-O$_2$ measurements during the AMAZOMIX 2021 cruise for stations S8 to S14 along transects Aa, located within IT fields. For long stations (S10-S14), two density profiles are shown to highlight step-like structures and isopycnal vertical displacements (illustrated by black arrows) along the transects. Distinct colors are used to represent each station within each transect. The density values for stations S8, and S9 range between 23.4 and 23.8 kg m$^{-3}$. (e) Vertical dissipation profiles for stations along transects Aa. Distinct colors are used to represent each station within each transect. Dashed and solid black lines in panel (e) are included for comparison purposes.*

Line 236: "Now, …."; Unnecessary use of word "Now".

R: Thanks for your remarks. We have rewritten it in line 261 of the revised manuscript.

Line 246: "….are found almost anywhere at S14…"; is there a better word than "anywhere" to be used in this sentence. Perhaps the words "at any depth" would be more appropriate?

R: Thanks for your remarks. We have rewritten it in lines 260-288 of the revised manuscript.

Line 247: "…shelf stations in the ITs regions, S3 and S5, …." ; the vertical profile of station S3 does not appear in Fig. 3. (it appears only in Fig. A1.c). Please refer to where it appears.

R: Thanks for your remarks. We have rewritten it in lines 275-277 of the revised manuscript, as shown below:

"

For shelf/shallow stations within the ITs regions (S3 and S5; Fig. 4c, and Fig. A8.c, Appendix), mixing is more pronounced, between $[10^{-6}, 10^{-5}]$ W kg$^{-1}$, near the bottom layer.

".

Line 271: "… dissipation rates (epsilon) already presented in section 3.1.2, IS…."; do you mean "ARE also reported in Fig. 3"?

R: Yes, thanks for your remarks. We have rewritten it in lines 316-317 of the revised manuscript, as reported here:
"

Dissipation rates, previously presented in subsection 3.1.2 and shown in Fig. 4, are found to be 2-3 orders of magnitude higher in the pycnocline compared at greater depths.

"

In Figure 4b, the baroclinic tidal currents at S11 are less clear. Do you have any reason to suggest for this fact?

R: Thanks for your remarks.
This issue could be related to the temporal interpolation of the SADCP data. Continuous SADCP measurements revealed missing values (e.g., between 13:30 and 15:30) at S11. Interpolation allowed us to estimate current data corresponding to the dissipation times.

Line 287: "Now, …"; Unnecessary use of word "Now".

R: Thanks for your remarks. We have rewritten it in line 336 of the revised manuscript.

Line 331: "IT ray paths were observed reflecting at the surface…."; The words "were observed" are not the best choice here, as you are talking about a model calculation; This reviewer suggests the use of the words "are expected to reflect" or "were predicted to reflect".

R: Thanks for your remarks. We have rewritten it in lines 408-409 of the revised manuscript, as shown below:
"

After bottom reflection and subsequent interaction with the pycnocline, the rays are expected to reflect seaward at the surface, typically at distances between 115-400 km.

".

Line 335: "flow becomes unstable beyond …."; use the word "below" or beneath" instead of beyond?

R: Thanks for your remarks. We have rewritten it in lines 412-416 of the revised manuscript.

Line 336: "Large epsilon are encountered where IT rays presumably interfere between them or…."

This reviewer does not see "IT ray path interference, since the different ray tracing computations (in red and blue colours) are a matter of different stratification choices, and hence you either have a "blue" ray or a "red" ray.

R: Thanks for your comments.
Our IT interference hypothesis is based on ray-tracing calculations using various stratification scenarios and sensitivity tests, which revealed a packet of rays (see Fig. A19 of paper, Appendix, for example) as shown below in figure RC3.2. To explain the persistent high mixing observed at S14 in the open ocean, we also proposed constructive interference of IT rays originating from two generation sites, Aa and Ab, as discussed in lines 551-569 of the revised manuscript. This hypothesis will be further investigated in a subsequent study.

[Figure]

Figure RC3.2: *Sensitivity tests of M2 IT ray-tracing along the transects (a) Aa and (b) Ab, conducted by varying the location of the critical topography slope. The tests use mean buoyancy frequency squared ($N^2$, in $s^{-2}$) obtained from CTD-$O_2$ data (September 2021) and NEMO-Amazon36 model data (2012-2016). Dashed colored lines represent IT beams calculated for different seasons (April, August, October, and September) and for varying locations of the critical topography slope. Grey areas indicate local topography. Panels (a) and (b) also include dissipation rate profiles ($\varepsilon$, in W kg$^{-1}$, shown as vertical colored bars on a logarithmic scale) from the VMP measurements. Subpanels within each panel illustrate the $N^2$ profiles derived from AMAZOMIX and the NEMO-Amazon36 model, which were used in the ray-tracing calculations. For comparison, sensitivity tests using $N^2$ measurements from individual stations along the corresponding transect (e.g., at S10) revealed similar ray paths (not shown), consistent with the packet of rays obtained using the mean $N^2$.*

Line 338-339: "Some large epsilon are observed where IT rays radiated at the surface (e.g. at S12, S14 and S20)....."; Large epsilon values are a common feature to all turbulence

profiles of dissipation rates near the surface. It looks more like the effect of wind mixing, rather than surfacing and reflection of IT rays.

R: Thanks for your comments.
Indeed, the mixing on the surface would also be linked to the effect of the wind.
Our study focuses more on the mixing observed outside of surface influence along IT paths, and the contribution of IT compared to other processes.

Line 353: "The next steps of our study IS…"; either use singular ("step") or use correct tense ("ARE").

R: Thanks for your remarks. We have rewritten it.
Ultimately, we decided to remove and reserve all sections on "Nutrients fluxes" for a separate paper in progress.

Discussion and Conclusion.

Line 398: "…., both large (up to 60 m length) isopycnal displacements…"; perhaps the wording "…, both large (up to 60 m) AMPLITUDE isopycnal displacements…" is more apropriate?

R: Yes. Thanks for your remarks. We have rewritten it in line 449 of the revised manuscript.

Revisions of section "Discussion and Conclusion" can be found below:
.
"

**4 Discussion and Conclusion**

The AMAZOMIX 2021 cruise provided, to the best of our knowledge, for the first time, direct measurements of turbulent dissipation using a velocity microstructure profiler (VMP) at multiple stations both inside and outside the influence of ITs. These measurements enabled the study of mixing processes at the Amazon Shelf break and the adjacent open ocean. To capture a full tidal cycle, data on turbulent dissipation rates, hydrography, and currents were collected alternately over 12 hours, with 4 to 5 profiles taken per station (see section 2). The locations of the 12-hour sampling stations were selected based on modeling results that provided realistic maps of IT generation and propagation (Fig. 1a; Tchilibou et al., 2022). Stations were located in the most energetic regions of IT, specifically at sites Aa, Ab, and D, covering stations S2 to S14, as identified in previous studies (Magalhaes et al., 2016; Tchilibou et al., 2022; Assene et al., 2024). Stations S19 to S21 were positioned in less energetic IT generation areas at site E, while stations S24 and S25 were located outside the influence of the IT fields (site G). Stations were distributed across different areas, including the shelf (e.g., S4, S9, and S19), the shelf-break (e.g., S3, S6, and S10), and the open ocean (e.g., S14, S24, and S20).

**Vertical Displacement, homogeneous layers**

The results revealed that, over a semi-diurnal tidal cycle, relevant amplitudes of vertical displacements (up to 60 m in length) and pronounced step-like structures (up to 40 m thick) were observed along transects Aa and Ab. In contrast, smaller and thinner structures were identified along other transects, such as E. These

differences are likely related to the propagation of ITs, which induce vertical displacements at tidal frequencies and promote mixing by creating homogeneous layers visible as step-like features in the density structure. The isopycnal displacements and step-like structures observed within the pycnocline are consistent with findings from other IT regions (e.g., Stansfield et al., 2001; Simpson and Sharples, 2012; Bordois, 2015; Koch-Larrouy et al., 2015; Zhao et al., 2016; Bouruet-Aubertot et al., 2018; Xu et al., 2020). Furthermore, IT propagation appears to have stronger energy along transects Aa and Ab compared to others, consistent with prior modeling studies (Tchilibou et al., 2022; Assene et al., 2024).

**Direct measurements of dissipation rate**

Dissipation rates measured with the VMP ranged from between $[10^{-10}, 10^{-5}]$ W kg$^{-1}$ below the XLD, spanning from the continental shelf to the open ocean. The XLD was found to be considerably larger than the MLD at all stations, except at S8, S10, and S25. This is consistent with regions exhibiting strong subsurface shear, such as the equatorial ocean and western boundary current areas (Noh and Lee, 2008). The exception observed at other stations may reflect larger mixing events that were not captured by the VMP measurements.

The highest dissipation rates, within $[10^{-6}, 10^{-5}]$ W kg$^{-1}$, were observed primarily at generation sites Aa, Ab, and D (e.g., at stations S6, S10, and S3). Slightly lower but still substantial dissipation rates, ranging from $10^{-8}$ to $10^{-7}$ W kg$^{-1}$, occurred a few kilometers (~40 km) from these generation sites (e.g., at S11 and S7), along IT pathways (e.g., at S12, S13, and S20), and even in regions farther from IT influence (e.g., at S24). Interestingly, dissipation rates were higher within $[10^{-7}, 10^{-6}]$ W kg$^{-1}$ in the open ocean, such as at station S14, located ~230 km from generation site Aa, as summarized in Fig. 9.

Similarly, the vertical eddy diffusivity coefficient, ranging from $10^{-3}$ to $10^{-1}$ m$^2$ s$^{-1}$, was highest at the shelf-break (at stations S3, S5, and S10), Away from the shelf-break, diffusivity values were lower but still substantial, within $[10^{-4}, 10^{-3}]$ m$^2$ s$^{-1}$ (e.g., at S2, S7, and S11).

[Figure]

*Figure 9: summary diagram illustrating the key processes driving mixing along the AMAZOMIX transects (e.g., Aa and Ab). At IT generation sites (e.g., S6 and S10), mixing rates are stronger, with ITs contributing around 65%, compared to mean circulation (NBC). Along IT pathways (e.g., S7 and S11), mixing decreases but remains notable, driven by nearly equal contributions from ITs and mean circulation. A key observation is the increased mixing ~ 230 km from two distinct IT generation sites at the shelf break. This hotspot at S14 coincides with the surfacing of IT rays from different sites and the presence of ISWs, suggesting that constructive interference of IT rays may generate ISWs, amplifying mixing at S14.*

In comparison, in other regions, dissipation rates measured by similar VMP instrument are found between $[10^{-7}, 10^{-5}]$ W kg$^{-1}$ in the IT generation zone Halmahera Sea, Indonesia (Koch-Larrouy et al., 2015; Bouruet-Aubertot et al., 2018), of Kaena Ridge, Hawaii (Klymak et al., 2008) and off the Changjiang Estuary (Yang et al. 2020). Whereas it is $[10^{-10}, 10^{-8}]$ W.kg$^{-1}$ along the IT path in the Southern Ocean (Gille et al., 2012) and in Halmahera Sea (Bouruet-Aubertot et al., 2018). Direct estimates of dissipation are almost $[10^{-11}, 10^{-10}]$ W kg$^{-1}$ far from IT influence (Koch-Larrouy et al., 2015; Bouruet-Aubertot et al., 2018) or under the influence of geostrophic current (Takahashi and Hibiya, 2019).

Our mixing coefficients are consistent with, the annual mean between $[10^{-4}, 10^{-3}]$ m$^2$ s$^{-1}$ of Ffield and Gordon (1992) or Koch-Larrouy et al. (2007), and aligned with others previous studies using the microstructure data (e.g. Tian et al., 2009; Koch-Larrouy et al., 2015; Bouruet-Aubertot et al., 2018; Xu et al., 2020), or modeling results (e.g. Koch-Larrouy et al., 2007).

This crucial vertical eddy diffusivity close enough to the surface along the IT paths may play a role in modulating heat (e.g., Assene et al., 2024) and chlorophyll content (de Macedo et al., 2023; M'Hamdi et al., 2024; in preparation) observed off the Amazon shelf.

Our study also found the highest dissipation rates at stations S3 and S5 of $[10^{-6}, 10^{-4}]$ W kg$^{-1}$ on the Amazon shelf , increasing near the bottom boundary layer. These findings compare well with values reaching up to $10^{-9}$ W kg$^{-1}$ within a kilometer of the seabed in the Southern Ocean (Sheen et al., 2013) and up to $10^{-6}$ W kg$^{-1}$ within a few meters from bottom topography off the Changjiang Estuary (Yang et al. 2020). This may indicate the presence of an active bottom boundary layer. Thus, kinetic energy of bottom flow was estimated using friction velocity, that was computed from total velocity averaged over the bottom-most 15 m for shallow stations. It showed bottom friction energy stronger between 16-35 J m$^{-2}$ at S3 and S5 mainly and lower (< 3 J m$^{-2}$) in the other stations on shelf (e.g., at S8). These results are smaller but still important on the Amazon shelf and comparable to values (517 kJ m$^{-2}$) in the Drake Passage region (on the continental slope) of the Southern Ocean (Laurent et al., 2012). The bottom mixing at S3 and S5 can indirectly exert a control on pycnocline mixing on the Amazon shelf (Inall et al., 2021).

**Contribution of Background circulation and ITs to mixing**
**Mean baroclinic current shear**
Another important aspect addressed in this study was quantifying the contributions of different processes to the observed heterogeneous mixing.

First, the mean baroclinic current (BC) was considered as a proxy for the background circulation. The BC was predominantly structured into a northwestward surface flow and a southeastward subsurface flow along the IT pathways. The strong surface flow toward the northwest is associated with the North Brazil Current (NBC), which originates from the northeastern coast of Brazil (e.g., Bourlès et al., 1999) and propagates along the Amazon shelf-break (e.g., at stations S7, S10, S11, S14, and S24). Conversely, the southeastward subsurface flow observed at stations such as S7 and S11 might result from NBC instability or the presence of a countercurrent at depth (Dossa et al., 2024, in preparation). At site E, the flow reversal observed at S21 - characterized by a southeastward surface flow and a northwestward subsurface flow - was located inside of the outer path of the Amazon plume. This reversal could be related with the influence of AWL formed by continental inputs (Prestes et al., 2018).

Both baroclinic flows demonstrated a significant potential for shear instability, with vertical shear ranging from $10^{-5}$ to $10^{-3}$ s$^{-2}$ off the Amazon shelf. The shear associated with the NBC was particularly pronounced around the pycnocline (between 40 and 200 m depth) at sites Aa, Ab, and G (e.g., at S6, S7, S10, S11, S14, and S24). At site E, the shear instability was stronger ($> 2.5 \times 10^{-4}$ s$^{-2}$) at the base of the pycnocline (e.g., at S20), potentially associated with NBC retroflection near [5–6°N, 50°W] during the fall season (Didden and Schott, 1993). The higher BC shear observed at S21, where flow direction reversals occurred, could be associated with the presence of a subsurface cyclonic eddy (Dossa et al., 2024, in preparation).

**ITs shear**

Second, the baroclinic tidal current was extracted from the total baroclinic current, revealing significant semi-diurnal (M2) component signals around the pycnocline. These signals, characterized by higher tidal modes (3-5), were more pronounced at generation and propagation sites Aa and Ab (e.g., at S6, S10, and S14) compared to other sites. The tidal shear within the pycnocline layer (80-120 m) is consistent with the observed IT signal patterns and large vertical displacements. It was stronger, reaching up to $10^{-3}$ s$^{-2}$, near the generation sites Aa and Ab (at S6 and S10) and in the open ocean at S14. Further from the generation sites (e.g., at S7, S11, and S20), the IT shear was smaller but still notable (reaching up to $10^{-4}$ s$^{-2}$). This highlights the significant role of ITs in driving mixing processes, particularly within the pycnocline, where strong vertical shears were observed near the shelf-break compared to regions far away. Outside the IT fields, such as at S24, the persistent high vertical shear near the bottom topography could be attributed to the active bottom boundary layer (Inall et al., 2021).

**IT/BC ratio**

Both IT and BC shear contribute to mixing, with their relative dominance varying across sites. Near the generation sites on the shelf-break, IT shear dominated the IT/BC shear ratio, such as at S6 (61.44/38.56 %), S10 (65.82/34.18 %), and S21 (58.55/41.45 %). Along the IT paths, the contributions were nearly equal (~50/50 %) at locations farther from the generation sites (e.g., at S20, S7, S11, and S13), except at S14 in the open ocean, where IT shear remained dominant (58.50/41.50 %). These findings align with the presence of ITs at generation sites Aa, Ab, and E (Tchilibou et al., 2022; Assene et al., 2024) and the stronger energy associated with NBC cores, particularly at S7 and S11.

These results are consistent with previous studies that identified strong tidal shear near IT generation sites, such as the Halmahera Sea (Bouruet-Aubertot et al., 2018), the Changjiang Estuary (Yang et al., 2020), the northwest European continental shelf seas (Rippeth et al., 2005), and the southern Yellow Sea (Xu et al., 2020).

The most relevant finding of this study was the relative increase in mixing within the pycnocline layer, observed at S14 in the open ocean, far from the IT generation sites.

**Discussion on the strong mixing at S14**

Along the IT paths, elevated remote dissipation rates (within $[10^{-7}, 10^{-6}]$ W kg$^{-1}$) were identified ~ 230 km from the shelf-break at S14.

This region is well known for intense IT dissipation, as shown by a realistic model (Tchilibou et al., 2022; Assene et al., 2024), and for the highest occurrences of ISWs generated by ITs (Fig. 1a; de Macedo et al., 2023), with large-amplitude ISWs exceeding 100 m clearly visible in satellite records (Brandt et al., 2002).

At station S14, where relative mixing increases, IT rays surfacing from two distinct IT generation sites coincide with the appearance of ISWs and mark the location where the NBC vanishes.

This region of wave-wave interactions can lead to the constructive interference of IT rays, potentially facilitating the emergence of higher tidal modes (New & Pingree, 1992; Silva et al., 2015; Barbot et al., 2022; Solano et al., 2023). These higher modes, in turn, could enhance the generation of nonlinear ISWs (e.g., Jackson et al., 2012) and contribute to the elevated dissipation rates (Xie et al., 2013), as observed at this station.

Moreover, IT interactions with baroclinic eddies may also contribute to turbulent dissipation (Booth and Kamenkovich, 2008), particularly in this area of pronounced eddy activity. However, no repeated AMAZOMIX stations observed during a tidal period were enclosed by mesoscale eddy activity, except potentially around site E, where possible evidence of a subsurface eddy was detected at S21 (Dossa et al., 2024, in preparation).

Future studies are needed to unravel the intricate interplay among these processes. The data collected during the AMAZOMIX cruise will provide a guide for improving our understanding and advancing parameterizations for modeling studies.

"

Line 426: "… It showed bottom FICTION"; I think you mean "bottom friction"?

R: Thanks for your remarks. We have rewritten it in line 500 of the revised manuscript, as shown in your previous comment.

In Figure 7 the legend for NBC is given, but it is not drawn in Figure 7.

R: Thanks for your comments. We have updated Figure 9 of revised manuscript as shown below in figure RC3.3.

[Figure]

Figure RC3.3: summary diagram illustrating the key processes driving mixing along the AMAZOMIX transects (e.g., Aa and Ab). At IT generation sites (e.g., S6 and S10), mixing rates are stronger, with ITs contributing around 65%, compared to mean circulation (NBC). Along IT pathways (e.g., S7 and S11), mixing decreases but remains notable, driven by nearly equal contributions from ITs and mean circulation. A key observation is the increased mixing ~ 230 km from two distinct IT generation sites at the shelf break. This hotspot at S14 coincides with the surfacing of IT rays from different sites and the presence of ISWs, suggesting that constructive interference of IT rays may generate ISWs, amplifying mixing at S14.

Line 446: "This relative diminution….".; can you use another word for "diminution" that is more common in literature? Perhaps the word "decrease" suits what the authors want to say?
R: Thanks for your remarks. We have rewritten it in lines 538-549 of the revised manuscript, as shown in your previous comment.

Line 471: "The really new aspect raised in this study…"; Please avoid writing "The really new". Something is either new or "not new".

R: Thanks for your remarks. We have rewritten it in lines 548-549 of the revised manuscript, as shown in your previous comment.

Line 483: "ITs disintegrate into a package of ISWs events…"; Please use the word "packet" for ISWs because this is what became acknowledged in the literatute for groups of internal wave trains.

R: Thanks for your remarks. We have rewritten it in lines 551-566 of the revised manuscript, as shown in your previous comment.

Line 510: "…, ITs act as an important supplier of nutrient into…."; please use plural for "nutrients".

R: Thanks for your remarks.

Ultimately, we decided to remove and reserve all sections on "Nutrients fluxes" for a separate paper in progress.

---

## Author Comment (AC3)

**Many thanks for the detailed correction of all typos and inconsistencies that were still present in the text. We have closely followed your recommendations to include the major and minor changes you have recommended and to restructure parts of the manuscript accordingly.**

**In this regard, we have revised and rewritten a few sections such as Abstract, method, results, discussion and conclusion in the revised manuscript. Quality control and revision of dissipation estimates, as well as current calculations were also done to ensure the validity of our results. We thought it useful to point out its detailed revisions (lines and sections) in the replies to your comments. Below (highlighted in blue and magenta) is an itemized response to the different issues raised in the review.**
**We believe that with all these additional changes and thanks to your valuable suggestions, we have achieved a marked improvement in the readability of the manuscript, as well as in the formal presentation and description of its main findings.**

This paper reports on microstructure measurements off the Amazon shelf. This is an interesting area with a combination of various dynamical processes, internal tides, low frequency circulation and amazonian water lenses. Consistently with this dynamics contrasted dissipation rates are observed with the highest values at generation sites, and along internal tide pathways and the lowest values in no-tidal areas. The relative contribution to dissipation of the mean baroclinic current (North Brazil current) compared to that of internal tide was estimated : the contribution of IT shear was found larger than BC shear near generation sites, equal along IT pathways. In addition turbulent diffusive nutrient fluxes were computed : large values were obtained.

I think there is interesting material for publication but that part of the analysis must be revisited to provide convincing results before it could be accepted for publication. Some of the figures are overloaded and not easily readable, having many figures in the appendix does not facilitate a fluent reading. Part of the sections would need a careful polishing.

The main result to highlight is the spatial contrast of dissipation rates and give insights on the origin of these variations.

I list in the following my main general comments:

-More information on the background state should be given: bathymetry, sea surface salinity (Amazone plume), as well as the mean current and for instance information on generation sites for internal tides (reorganization of Figure 1 which is not easy to read, subplot (b) does not seem necessary)

R:Thank you for your comment.

We have updated the information on the background state, including bathymetry, the Amazon plume, internal tide generation sites, and mean circulation, and reorganized Figure 1 of the manuscript accordingly.

Revisions can be found in the "Introduction" section, lines 34-82, and the updated Figure 1 of the revised manuscript is provided below. We retained Figure 1b of the manuscript as it highlights the signature of the ISWs observed in the region during the AMAZOMIX cruise.

The revisions are shown below:

"

**1 Introduction**

Turbulent mixing in the ocean plays an important role in sustaining the thermohaline and meridional overturning circulation and in closing the global ocean energy budget (Kunze, 2017). These processes have strong implications for the climate, influencing heat and carbon transport, as well as nutrients supply for photosynthesis (Huthnance, 1995; Munk and Wunsch, 1998). Mixing processes can result from wind in the surface waters layer, internal waves and shear instability in the ocean interior, and bottom friction near the bottom layer (Miles, 1961; Thorpe, 2018; Ivey et al., 2020; Inall et al., 2021). Barotropic tides interacting with steep shelf-break topography trigger internal waves at tidal frequencies and harmonics, known as internal tides (ITs), which can propagate and produce mixing. These ITs can be expressed by large vertical displacements (up to tens of meters) of water masses (Garrett and Kunze, 2007). After their generation on the shelf-break, the (more unstable) higher modes of ITs may dissipate locally, while the lower modes can propagate far away (Zhao et al., 2016). IT beams (generated where the slope of the ITs and the topography match together on the shelf-break) can propagate vertically, resulting in reflection, scattering and dissipation of ITs at the bottom, surface waters, or thermocline levels (New and Da Silva, 2002; Gerkema and Zimmerman, 2008; Bordois, 2015; Zhao et al. 2016). They can also dissipate when energy fluxes interfere (Zhao et al., 2012) or interact with strong baroclinic eddies or currents (Rainville and Pinkel, 2006; Whalen et al., 2012). Furthermore, ITs may disintegrate into packets of higher-mode nonlinear internal solitary waves (ISWs), which can propagate and dissipate offshore (Jackson et al., 2012).

Previous and recent studies have shown that ITs-induced turbulent mixing can affect the surface, such as sea surface temperature (Ray and Susanto, 2016; Nugroho et al., 2018; Assene et al., 2024), chlorophyll content (Muacho et al., 2014; M'Hamdi et al., 2024; in preparation), marine ecosystems (Wang et al., 2007; Zaron et al., 2023), and atmospheric convection and the rainfall structure (Koch-Larrouy et al., 2010, Sprintall et al. 2014).

In the western tropical Atlantic, the Amazon River-Ocean Continuum (AROC) constitutes a key region of the global oceanic and climate system (Araujo et al., 2017; Varona et al., 2018). This region (Fig. 1a) is characterized by a system of western boundary currents, including North Brazil Current (NBC). NBC, which flows northwestward, has its core velocities ($\sim 1.2$ m s$^{-1}$) that remain stable from the surface to a depth of 100 m (Johns et al., 1998; Bourlès et al., 1999; Barnier et al., 2001; Neto and Silva, 2014). This region also experiences highly variable dynamics due to the Amazon River Plume. During the rainy season (May-July), peak discharge can extend the plume over 1500 km offshore, northwest along the NBC. In the dry season

(September-November), reduced discharge and stronger saline intrusion may confine the plume to less than 500 km offshore, near the Amazon Shelf, with some eastward dispersion (Coles et al., 2013). The Amazon plume can generate vertical shear in underlying currents, enhancing mixing. Additionally, a system of Amazonian Lenses of water (AWL), influenced by continental inputs, may affect both the boundary layer and mixed layer patterns (Silva et al., 2005; Prestes et al., 2018).

In the AROC region, the Amazon shelf-break is a hotspot for the generation, propagation and dissipation of ITs and ISWs as a result of non-linear processes (Geyer, 1995; Brandt et al., 2002; Magalhães et al., 2016; Ruault et al., 2020; Tchilibou et al., 2022; Fig 1). Previous studies using Synthetic Aperture Radar (SAR) satellite images (Magalhaes et al., 2016) identified ISWs along the path of ITs propagating from two sites (i.e., sites Aa and Ab; Fig. 1a). Conversely, other sites showed no ISWs propagation (i.e., sites E and D; Fig. 1a, 1b and 1c) (see Magalhaes et al., 2016 for definition). Using numerical modeling, Tchilibou et al. (2022) showed that about 30 % of the M2 (dominant tidal component; Le Bars et al. 2010) ITs energy is dissipated locally (for higher-modes ITs) at sites E, Aa, Ab and D (Fig. 1a), while the remaining lower-modes ITs energy can be dissipated remotely. Dissipation away from the generation sites (E, Aa, Ab and D; Fig. 1a) can result from the shear instabilities caused by ITs-ITs and/or ITseddy/current interactions. Despite the presence of ITs, no direct measurements of dissipation rates have been conducted to our knowledge.

The mixing induced by these internal waves in the region was observed during the AMAZOMIX cruise (Bertrand et al., 2021). The cruise was designed with stations/transects inside and outside ITs fields (Fig. 1a and 1c) to measure ITs dissipation and study their impact on the AROC ecosystem. Direct microstructure measurements of temperature, salinity and velocity were conducted at the different repeated stations/transects over a M2 tidal cycle (~ 12.42 h). These cruise measurements offer an opportunity to explore whether ITs play a role in mixing within the AROC region. In this study, we will quantify mixing and identify the associated processes off the Amazon shelf. We will calculate turbulent kinetic energy (TKE) dissipation rates, vertical displacements of isopycnal surfaces and vertical eddy diffusivities using in situ microstructure and hydrography data. Finally, the baroclinic shear of currents and their contributions to mixing will be calculated from current data collected between stations and transects.

[Figure]

Figure 1: a) Map of a part of the AMAZOMIX 2021 cruise off the Amazon shelf, showing bathymetric contours (100 m, 750 m, 2000 m, and 3000 m isobaths) in gray. Colored circles and stars indicate short and long CTD-O2/L-S-ADCP stations, respectively, with the corresponding sampling dates represented by the color bar. Solid black lines depict SADCP transects (for Aa, Ab, D, G, and E). Magenta arrows show the 25-hour mean depth-integrated baroclinic IT energy flux (September 2015, from the NEMO model) originating from IT generation sites (Aa, Ab, D, and E) along the shelf break. The solid brown line represents the NBC pathways

*illustrating background circulation. Shattered colored lines highlight ISW signatures. b) 1A Sentinel image acquired on 12th September 2021, showing ISW signatures. c) Tidal range at AMAZOMIX stations, with ISW signature dates marked by red bars.*

"

-Baroclinic currents and energy : it is unclear why a parameterization of e is referred to as baroclinic total energy.  The MG  parameterization does not provide any relevant information (the correlation with epsilon is not clear and it is used as a proxy to evaluate the contribution of tidal shear and low frequency shear which I find questionable)

R: Thank you for your comment.

Indeed, the MacKinnon-Gregg parameterization was applied as a proxy to evaluate the contributions of tidal and low-frequency shear, primarily for comparison purposes. However, no scientific results were derived from it in this study.

Ultimately, we decided to remove the section using the parameterization and reserve the study and comparison of in-situ data to mixing parameterization for a separate paper in progress. In our paper, we used the vertical shear to assess the contributions of tidal and mean shear.

Revisions can be found in the "Methods" section, lines 122–235, as shown below:

"

**2.2 Methods**

**TKE dissipation rates**

The VMP data are processed using ODAS Matlab library (developed by Rockland Scientific International, Inc) to infer the TKE dissipation rate ($\varepsilon$). The processing methods for the VMP data are briefly described here and adhere to the recommendations of ATOMIX (Analyzing ocean turbulence observations to quantify mixing), as reported by Lueck et al. (2024), and have been validated against the benchmark estimates (presented in Fer et al., 2024).

First, the VMP data are converted into physical shear units, and the time series are prepared. Continuous sections of the time series are selected for dissipation estimation. Before spectral estimation, the aberrant shear signals caused by vessel wake contamination are removed. Collisions of the shear probe with plankton and other particles are removed using the de-spiking routine. The records from each section are then high-pass filtered (e.g., at station S6 and S10; Fig. 2a, and Fig. A1, Appendix).

Shear spectra are estimated using record lengths (L) and Fast Fourier Transform segments of 2 s, which are cosine windowed and overlapped by 50% (e.g., at station S6 and S10; Fig. 2b, and Fig. A1, Appendix). Additionally, vibration-coherent noise is removed. Different L and overlap (O) settings were selected and tested based on the environment (e.g., deep vs. shallow water), following Fer et al. (2024). For shallow stations, L (O) was shortened to 5 s (2.5 s), in contrast to the 8 s (4 s) used for deeper stations, due to evidence of overturns

observed in AMAZOMIX acoustic measurements at deeper stations (Koch-Larrouy et al., 2024; in preparation). This adjustment helped to optimize the spatial resolution of dissipation estimates in shallow water stations.

Finally, $\varepsilon$ is determined using the spectral integration method and by comparison with the Nasmyth empirical spectrum (Nasmyth, 1970). Quality assurance tests are carried out in accordance with ATOMIX's recommendations (Lueck et al., 2024). A figure of merit < 1.4 is used to exclude bad data (e.g., at station S6 and S10; Fig. 2b, and Fig. A1, Appendix), and the fraction of data affected by de-spiking is < 0.05.

[Figure]

*Figure 2: Example of wavenumber spectra from a dissipation structure segment used to determine the dissipation rate at station S6 at a pressure of 87.9 dBar. (a) Cleaned and high-pass filtered signals from shear probe 1 (blue) and shear probe 2 (red, offset by 5 s⁻¹). (b) Wavenumber spectra for shear probes 1 and 2. Thick lines (blue for probe 1, red for probe 2) show shear spectra with coherent noise correction, while thin lines (sky blue for probe 1, orange for probe 2) show spectra without correction. Triangles mark the maximum wavenumber used for dissipation rate estimation. Black lines represent Nasmyth reference spectra for estimated*

*dissipation rate of 3.8 x 10$^{-8}$ W kg$^{-1}$ for both shear probes. Dissipation rate estimates for shear probe 1 and shear probe 2 at a pressure of 87.9 dBar yielded a figure of merit of 0.93 and 0.94, respectively.*

**The vertical eddy diffusivity coefficient**

The efficiency of turbulence in redistributing energy is assessed through the calculation of the vertical eddy diffusivity coefficient ($K_z$). This coefficient is particularly significant in regions such as pycnoclines, where stratification suppresses mixing, making turbulence-driven mixing a key mechanism for vertical energy transport (Thorpe, 2007).

$K_z$ is calculated from ε following the formulation of Osborn (1980), given by $K_z = ε \Gamma N^{-2}$. Here, $N^2$ is the buoyancy frequency squared, which is calculated using the sorted potential density profiles ($σ_θ$) obtained from CTD-O$_2$ data. It is given by $N^2 = - (g/ρ_0) (dσ_θ/dz)$, where $ρ_0$ is a reference density (1025 kg m$^{-3}$) and g is the gravitational acceleration. $\Gamma$ is the mixing efficiency, defined as the ratio between the buoyancy flux and the energy dissipation, and is typically set to 0.2, which corresponds to the critical Richardson number Ri = 0.17 (Osborn, 1980). ε is linearly interpolated into the depths of $N^2$.

Turbulence within the pycnocline can reduce stratification and increase vertical eddy diffusivity below the mixing layer (Thorpe, 2007). Subsurface mixing, driven by the breaking of ITs and shear instabilities, plays a particularly important role below the mixed layer, especially in equatorial waters (Gregg et al., 2003).

There are several criteria for defining the Mixed Layer Depth (MLD). In this study, we use the commonly accepted density threshold criterion of 0.03 kg m$^{-3}$, as defined by de Boyer Montégut et al. (2004) and Sutherland et al. (2014), to estimate the MLD for each CTD-O$_2$ profile. Notably, comparisons with density thresholds of 0.01 and 0.02 kg m$^{-3}$ revealed no major differences in MLD across the AMAZOMIX stations and transects (Fig. A2, Appendix).

The miXing Layer Depth (XLD) is defined as the depth at which ε decreases to a background level (Sutherland et al., 2014). Previous studies have applied various thresholds for background dissipation levels, such as such as 10$^{-8}$ and 10$^{-9}$ W kg$^{-1}$ in higher latitudes based on in situ observations (Sutherland et al., 2014; Lozovatsky et al., 2006; Cisewski et al., 2008; Brainerd and Gregg, 1995) and 10$^{-5}$ m$^2$ s$^{-1}$ using an ocean general circulation model (Noh and Lee, 2008). In this study, XLD is specified as the depth where ε drops from its first minimum value. This aligns with previous dissipation thresholds and ensures that mixing is captured independently of surface influences. The Upper (UTD) and Lower (LTD/LPD) Thermocline/Pycnocline Depth are delimited as defined by Assunçao et al (2020). UTD corresponded to the depth where the vertical temperature gradient $∂θ/∂z$ = 0.1 ℃ m$^{-1}$, while LTD/LPD were the last depth below the UTD at which $N^2 ≥ 10^{-4}$ s$^{-2}$.

**Baroclinic currents**

To analyze the processes explaining dissipation and mixing, particularly along internal tidal (IT) paths, we estimate shear instabilities associated with the semi-diurnal (M2) ITs and mean circulation, as well as their contributions to mixing.

The M2 tidal component of the tidal current is derived by calculating the baroclinic (semi-diurnal) tidal velocity [$u''$, $v''$] (Fig. A3, Appendix), following these equations:

$$[u', v'] = [u, v] − [u_{bt}, v_{bt}], \tag{1}$$

$$[u_{bt}, \ v_{bt}] = \frac{1}{H} \int_{-H}^{0} [u, \ v]dz, \tag{2}$$

$$[u'', \ v''] = [u', \ v'] - [\overline{u'}, \ \overline{v'}]. \tag{3}$$

Here, $[u, \ v]$ represent total horizontal velocities (Fig. A3, Appendix) obtained from SADCP data. The components $[u', \ v']$ and $[u_{bt}, \ v_{bt}]$ represent baroclinic and barotropic components of horizontal velocities, respectively (Fig. A3, Appendix). H is water depth. The baroclinic mean velocities $[\overline{u'}, \ \overline{v'}]$ (Fig. A3, Appendix), calculated to estimate mean circulation along IT paths, are decomposed into along-shore $\overline{u'}_l$ and cross-shore $\overline{u'}_c$ velocities. The overbar denotes the average over a M2 tidal period.

Note that continuously collected SADCP for some stations (e.g., S11) are not sufficiently resolved due to gaps filled by interpolating between time points. The similar processing are applied to the CTD-$O_2$ data collected alternately. SADCP time series data are less than 17 hours at all long stations, except for S14, which spans 42 hours. As a result, the diurnal and semidiurnal period fittings are not formally distinct (except at S14; Figs. A4 and A5, Appendix), and the inertial period (at least 5 days) cannot be resolved in our dataset. This limits our ability to separate currents by frequency and examine the associated dissipation.

The velocity profiles from LADCP are glued into our SADCP time series data below ~ 500 m depth at long stations.

To evaluate shear instabilities associated with ITs and the mean background circulation, we compute the baroclinic tidal vertical shear squared ($S^{2''}$) and mean shear squared ($\overline{S^{2'}}$) (Fig. A3, Appendix), as follows:

$$S^{2''} = (\partial u''/\partial z)^2 + (\partial v''/\partial z)^2, \tag{4}$$

$$\overline{S^{2'}} = (\partial \overline{u'}/\partial z)^2 + (\partial \overline{v'}/\partial z)^2. \tag{5}$$

To evaluate the impact of bottom friction on mixing, we calculate kinetic energy $\varepsilon_f = \frac{1}{2}\rho_s(u_f^2)$ near the bottom boundary layer at shallow stations using friction velocity $u_f = u_b\sqrt{C_d}$, where $C_d$=2.5 x $10^{-3}$ is a drag coefficient obtained from the NEMO model. Huang et al. (2019) showed that the bottom boundary layer thickness spatially varies between 15-123 m in the Atlantic Ocean, with a median of ~ 30-40 m in the North Atlantic. We define bottom layer thicknesses in our study area based on measured bathymetry from CTD-$O_2$ and near-bottom currents from ADCP. Here, $u_b$ is the total velocity averaged over a thickness of 20 m above the seabed for shallow stations and 40 m for deep stations.

The individual contributions of semi-diurnal ITs and mean circulation are then expressed as follows: $E''/(\overline{E'} + E'')$ for tidal contribution and $\overline{E'}/(\overline{E'} + E'')$ for mean circulation contribution. Here, $E = N*S$. N is the buoyancy frequency and S is vertical shear. S can be substituted by $S^{2''}$ and $\overline{S^{2'}}$.

**Ray tracing calculation**

Analyzing both the mean currents and the spatial dimension along the IT pathways offers another insight into the mechanisms responsible for observed mixing (Rainville and Pinkel, 2006). IT energy rays are generated in

regions with steep topography, such as the shelf break, where IT slope matches with the bottom slope (i.e., critical slopes) before propagating within the ocean interior. These rays, moving both downward and upward, encounter the seasonal pycnocline, resulting in beam scattering and the formation of large IT oscillations. As these oscillations steepen, they disintegrate into nonlinear ISWs, a process known as "local generation" of ISWs (New and Pingree, 1992). To explore IT paths, ray-tracing techniques are employed, as previously used by New and Da Silva (2002) and Muacho et al. (2014), to investigate the effectiveness and expected pathways of the IT beams off the Amazon shelf. One main assumption in our linear-theory-based hypothesis is that stratification remains horizontally uniform along the IT propagation path, although in reality, it may vary due to submesoscale and mesoscale variability. This limitation makes the ray tracing approach less realistic but still useful as a first-order estimate of energy distribution. The IT ray-tracing calculation assumes that in a continuously stratified fluid, ITs energy can be described by characteristic pathways of beams (or rays) with a slope c to the horizontal:

$$c = \pm\left(\frac{\sigma^2 - f^2}{N^2 - \sigma^2}\right)^{1/2},$$  (6)

where $\sigma$ is the M2 tidal frequency (1.4052x10$^{-4}$ rad s$^{-1}$), and f is the Coriolis parameter. $N^2$ are obtained from time-averaged AMAZOMIX CTD-O$_2$, glued with monthly $N^2$ profiles from Amazon36 (NEMO model outputs, 2012-2016) below 1000 m depth. Amazon36 is a NEMO configuration, specifically designed to cover the western tropical Atlantic from the mouth of the Amazon River to the open sea (see Tchilibou et al., 2022; Assene et al., 2024; for configuration details and model description). IT ray-tracing diagrams are performed along the transects. Seasonal sensitivity tests of rays (August, September, October, and April) are conducted by varying the critical slope positions and $N^2$ to explore its influence and generate a set of ray paths consistent with characteristics of IT pathways (Figs. A6 and A7, Appendix).

"

-Ray tracing calculation is applied but horizontal density gradients are neglected : this assumption is surprising owing to the major influence of the mean baroclinic flow.

Also in Figure 5 it would be more clear regarding the IT ray paths to consider the full water column

R: Thank you for your comment.

Indeed, our ray-tracing calculation neglected horizontal density gradients and the mean baroclinic flow, which we acknowledge as a limitation. In our study, the ray-tracing calculation superimposed with mean current data is used as another tool to gain insights into the mechanisms driving the observed mixing along the IT path. Indeed, change in density along the propagation path will affect the wavelength and beams of the rays.

To try to assess the potential influence of horizontal density gradients, we have tested different N2 profiles at specific stations (e.g., S10, S12, and S14) along transect Aa. The sensitivity tests (Figure RC2.1) demonstrated that ray paths align within the packets of rays observed when using mean N2 profiles at different times (e.g., in September and October; Figure RC2.2). Similarly, the influence of mean circulation could be very important for the ray. This question is beyond the scope of this study, and is tackled in

another paper we are working on using model results. Both influences-stratification and background circulation—are discussed in the sections "Methods" (lines 233–235) and "Results" (lines 401–423) of the revised manuscript.

In Figure 5 of the revised manuscript, we considered the full water column for internal tide (IT) ray paths, as depicted in the figure RC2.2. However, the y-axis in the manuscript is limited to 1000 m depth to enhance the visibility of dissipation profiles, density/stratification.

The influence of stratification and mean current on mixing and IT ray paths using ray-tracing calculations will be explored in a separate modelling paper.

[Figure]

figure RC2.1: Example of sensitivity tests with different cross-sectional measurements of $N^2$ along the transect T1 $N^2$. colors are used to distinguish different cross-shore measurements of $N^2$ for corresponding stations on T1. Topography steepness (gamma = ray slope / topography slope) for T1 using measured $N^2$ of S10. Gamma is illustrated by the colored bar (named gamma S10).

[Figure]

figure RC2.2: Sensitivity tests of M2 IT ray-tracing along the transects Aa, conducted by varying the location of the critical topography slope. The tests use mean buoyancy frequency squared ($N^2$, in $s^{-2}$) obtained from CTD-$O_2$ data (September 2021) and NEMO-Amazon36 model data (2012-2016). Dashed colored lines represent IT beams calculated for different seasons (April, August, October, and September) and for varying locations of the critical topography slope. Grey areas indicate local topography. Panel also includes dissipation rate profiles ($\varepsilon$, in W $kg^{-1}$, shown as vertical colored bars on a logarithmic scale) from the VMP measurements. Subpanels within each panel illustrate the $N^2$ profiles derived from AMAZOMIX and the NEMO-Amazon36 model, which were used in the ray-tracing calculations. For comparison, sensitivity tests using $N^2$ measurements from individual stations along the corresponding transect (e.g., at S10) revealed similar ray paths (not shown), consistent with the packet of rays obtained using the mean $N^2$.

Revisions can be found in the "Methos/Ray tracing calculation" section, in lines 223–226 and 233-235 of revised manuscript, as shown below:

"

One main assumption in our linear-theory-based hypothesis is that stratification remains horizontally uniform along the IT propagation path, although in reality, it may vary due to submesoscale and mesoscale variability. This limitation makes the ray tracing approach less realistic but still useful as a first-order estimate of energy distribution..

"

"

Seasonal sensitivity tests of rays (August, September, October, and April) are conducted by varying the critical slope positions and $N^2$ to explore its influence and generate a set of ray paths consistent with characteristics of IT pathways (Figs. A6 and A7, Appendix)..

"

-Figure 3 : it is difficult to have a view on the evolution along the transects, why not show e profils along the transect with density superimposed, some large values at the end of the e profils would need to be checked (S12 and S14 in Fig. 3. and 3.e as well as S2 figA.1.c)

R: Thank you for your comment.

Indeed, the evolution of dissipation profiles, as well as, density along the transect were shown in Figure 8 of revised manuscript and figures in appendix. An example is shown in the figure RC2.4 below.

[Figure]

Figure RC2.4: IT ray-tracing diagrams for the M2 tidal constituent along transects Aa. The calculations were performed using the mean buoyancy frequency squared ($N^2$, in $s^{-2}$) obtained from CTD-$O_2$ data (ray in red) and NEMO-Amazon36 model data (ray in blue) for September. Grey areas represent local topography and black circles indicate the critical topography slope (ray generation sites). Panel also show along the transects Aa: along-shore mean total currents ($u_l$, in m $s^{-1}$) from ADCP (Dashed black lines), potential density from CTD-$O_2$ (grey contours), and dissipation rate profiles ($\varepsilon$, in W $kg^{-1}$, on a logarithmic scale) from the VMP (vertical colored bars). Subpanels within each panel illustrate the $N^2$ profiles from AMAZOMIX (red line) and the NEMO-Amazon36 model (blue line) used for ray-tracing calculations. Upper Thermocline Depth (UTD, dotted lines) and Lower Thermocline/Pycnocline Depth (LTD/LPD, dashed lines) are also indicated.

We have revised the important values at the end of the e profiles at these stations (S12, S14, and S2).

Revisions and checks of the VMP profile processing and dissipation estimates have been carried out, along with quality assurance tests, in alignment with all reviewers' comments and ATOMIX's recommendations (Lueck et al., 2024).

a) First, the revisions were focused on the VMP profile processing, particularly on:

- Parameters controlling shear spectra estimation, such as record lengths (L), which are cosine-windowed and overlapped (O) by 50%.
- Parameter for extracting the section or profile (continuous part of the time series), including the minimum depth of extraction (Pmin).

For shallow stations, we used L = 5 s and O = 2.5 s, instead of the previous L = 4 s and O = 2 s. Parameters for deep stations remain unchanged (L = 8 s and O = 4 s).

Additionally, we adjusted the minimum depth of extraction to Pmin = 10 m for deep stations and Pmin = 3 m for shallow stations, compared to the previously used value of Pmin = 1 m for both station types.

b) Second, the revisions and checks focused on the quality assurance measures for dissipation estimates, with quality control checks and adjustments applied across all stations (e.g., S2, S7, S12, S14).

For instance, at station S14, previous dissipation estimates showed some peaks at various depths, particularly between 100–200 m, 300–400 m, and 500–700 m. While the fraction of shear data affected by despiking during processing was <0.05, the figure of merit (FM)—used to filter out poor-quality data—for shear probe 1 was >>1.4 at certain depths (e.g., around 327 m compared to 122 m, as shown in figure RC2.5). In contrast, the FM for shear probe 2 remained <1.4.

After checks, quality control of dissipation estimates have been revised for all stations (e.g., S6, S10, S14). We have retained only dissipation estimates from either one or both probes that met the quality assurance criteria (FM < 1.4 and fraction of despiked shear data <0.05, as recommended by ATOMIX), as shown at S6 and S10 for example (figure RC2.6).

The final dissipation estimate was computed as the average of the estimates from both shear probes, followed by the mean of the dissipation profiles for the station, as illustrated in figure RC2.7.

The revision of section "methods/TKE dissipation rates" can be found in lines 169-194, in text, as shown below:

"

The VMP data are processed using ODAS Matlab library (developed by Rockland Scientific International, Inc) to infer the TKE dissipation rate ($\varepsilon$). The processing methods for the VMP data are briefly described here and adhere to the recommendations of ATOMIX (Analyzing ocean turbulence observations to quantify mixing), as reported by Lueck et al. (2024), and have been validated against the benchmark estimates (presented in Fer et al., 2024).

First, the VMP data are converted into physical shear units, and the time series are prepared. Continuous sections of the time series are selected for dissipation estimation. Before spectral estimation, the aberrant shear signals caused by vessel wake contamination are removed. Collisions of the shear probe with plankton and other particles are removed using the de-spiking routine. The records from each section are then high-pass filtered (e.g., at station S6 and S10; Fig. 2a, and Fig. A1, Appendix).

Shear spectra are estimated using record lengths (L) and Fast Fourier Transform segments of 2 s, which are cosine windowed and overlapped by 50% (e.g., at station S6; Fig. 2b, and Fig. A1, Appendix). Additionally, vibration- coherent noise is removed. Different L and overlap (O) settings were selected and tested based on the environment (e.g., deep vs. shallow water), following Fer et al. (2024). For shallow stations, L (O) was shortened to 54s (2.5s), in contrast to 8 s (4 s) used for deeper stations, due to evidence of overturns observed in AMAZOMIX acoustic measurements at deeper stations (Koch-Larrouy et al.,

2024; in preparation). This adjustment helped to optimize the spatial resolution of dissipation estimates in shallow water stations.

Finally, $\varepsilon$ is determined using the spectral integration method and by comparison with the Nasmyth empirical spectrum (Nasmyth, 1970). Quality assurance tests are carried out in accordance with ATOMIX's recommendations (Lueck et al., 2024). A figure of merit < 1.4 is used to exclude bad data (e.g., at station S6; Fig. 2b, and Fig. A1, Appendix), and the fraction of data affected by de-spiking is < 0.05.

"

[Figure]

Figure RC2.5: Example of wavenumber spectra from a dissipation structure segment used to determine the dissipation rate at station S14 at a pressure of 337.7 dBar. (a) Pressure record for the entire data file (blue) and the specific segment being analyzed (red). (b) Cleaned and high-pass filtered signals from shear probe 1 (blue) and shear probe 2 (red, offset by 5 s$^{-1}$). (c) Wavenumber spectra for shear probes 1 and 2. Thick lines (blue for probe 1, red for probe 2) show shear spectra with coherent noise correction, while thin lines (sky blue for probe 1, orange for probe 2) show spectra without correction. Triangles mark the maximum wavenumber used for dissipation rate estimation. Black lines represent Nasmyth reference spectra for estimated dissipation rate of 8.9 x 10$^{-10}$ W kg$^{-1}$ and 1 x 10$^{-6}$ W kg$^{-1}$ for shear probes 1 and 2, respectively. Dissipation rate estimates for shear probes 1 and shear probe 2 at a pressure of 337.7 dBar yielded a figure of merit of 1.3 and 9, respectively. Panel (d) is similar to panel (c) but:- with Nasmyth reference spectra for estimated dissipation rate of 3.9 x 10$^{-8}$ W kg$^{-1}$ and 4.6 x 10$^{-8}$ W kg$^{-1}$ for shear probes 1 and 2. -wth dissipation rate estimates for shear probes 1 and 2 at a pressure of 122.4 dBar yielding a figure of merit of 0.84 and 1, respectively.

[Figure]

Figure RC2.6: Similar to Fig. RC1.1. but for stations (a)-(b)-(c) S6 and (d)-(e)-(f) S10. For S6 (panels c), Black lines represent Nasmyth reference spectra for estimated dissipation rate of 3.8 x 10-8 W kg-1 for both shear probes, and dissipation rate estimates for shear probes 1 and shear probe 2 at a pressure of 337.7 dBar yielded a figure of merit of 0.93 and 0.94, respectively. For S10 (panel f), Black lines represent Nasmyth reference spectra for estimated dissipation rate of $1.6 \times 10^{-8}$ W kg$^{-1}$ and $1.5 \times 10^{-8}$ W kg$^{-1}$ for shear probes 1 and 2, respectively, and dissipation rate estimates for both shear probes at a pressure of 337.7 dBar yielded a figure of merit of 1.2.

[Figure]

figure RC2.7: (a) Horizontal maximum and (b)-(c)-(d)-(e) vertical dissipation rates (ε, in W kg-1, on a logarithmic scale) before revisions and checks processes for all stations along transects T1 to T2. (f) Horizontal maximum and (g)-(h)-(i)-(j) vertical dissipation rates (ε, in W kg-1, on a logarithmic scale) after revisions and checks processes for all stations along transects Aa, Ab, D, E, and G. Distinct colors are used to represent each station within each transect. Dashed and solid black lines in panels (b) to (e) are included for comparison purposes.

-the relationship between step-like structures and strong internal tides is not convincing as it is presented

R: We agree that without prior knowledge of step-like features, this action can be difficult to understand. Thank you for highlighting this. In response, we have added arrows in Figure 3 of manuscript to indicate the "step-like structures" and "vertical displacement," making it easier for readers to understand the "step-like features" we are referring to (see figure RC2.8 below).

[Figure]

figure RC2.8: *Density profiles ($\sigma_\theta$, kg m$^{-3}$) obtained from CTD-O$_2$ measurements during the AMAZOMIX 2021 cruise for stations S8 to S14 along transects Aa, located within IT fields. For long stations (S10-S14), two density profiles are shown to highlight step-like structures and isopycnal vertical displacements (illustrated by black arrows) along the transects. Distinct colors are used to represent each station within transect. The density values for stations S8, and S9 range between 23.4 and 23.8 kg m$^{-3}$*

-Figure 4 : Emphasis is made on shear instability, this is quite convincing at S10 but not at S14, this should be interesting to comment on.

R: Thank you for your comment.

Indeed, the focus is on the contribution of tidal and low-frequency shear instability in the mixing process. It was found that tidal shear and its influence are stronger at S10, located near the tidal generation site, compared to S14, which is farther from the generation site in the open ocean. This is discussed in the section "Discussion and Conclusion" (lines 435–569), with particular emphasis at S14 in lines 551–566 of the revised manuscript, as shown below at the end of this document.

-Competitive processes to generate mixing: the hypothesis of shear-driven dissipation is followed with the aim to discriminate between the low frequency shear contribution and the IT shear one. I find this subsection difficult to follow, and I don't understand why the MG parameterization is introduced to this aim.

R: Thank you for your comment.

Indeed, the MacKinnon-Gregg parameterization was applied as a proxy to evaluate the contributions of tidal and low-frequency shear, primarily for comparison purposes. However, no scientific results were derived from it in this study.

Ultimately, we decided to remove and reserve the mixing parameterization for a separate paper in progress. Instead, we revised the subsection and focused on vertical shear to assess the contributions of tidal and mean shear.

-Figure 5 : the ray tracing approach should be revisited with taking into account the low frequency current, mean along shore current displays some spatial variations that may modify the ray structure ;

R: Thanks for your comment.

Indeed, our ray-tracing calculation neglected horizontal density gradients and the mean baroclinic flow, which we acknowledge as a limitation. In our study, the ray-tracing calculation superimposed with mean current data is used as another tool to gain insights into the mechanisms driving the observed mixing along the IT path.

To try to assess the potential influence of horizontal density gradients, we tested N2 profiles at specific stations (e.g., S10, S12, and S14) along transect Aa. The sensitivity tests (Figure RC2.9) showed that ray paths align within the packets of rays observed when using mean N2 profiles at different times (e.g., in September and October; Figure RC2.10). Similarly, the influence of mean circulation was evaluated by superimposing on the figure RC2.10 the mean total current, as shown in the corresponding figure 8 of manuscript revised.

Both influences—stratification and background circulation—are discussed in the sections "Methods" (lines 233–235) and "Results" (lines 401–423) of the revised manuscript.

The influence of stratification and mean current on mixing and IT ray paths using ray-tracing calculations will be explored more in a separate paper.

[Figure]

figure RC2.9: Example of sensitivity tests with different cross-sectional measurements of N²
along the transect T1 N². colors are used to distinguish different cross-shore measurements of N²

[Figure]

figure 10: IT ray-tracing diagrams for the M2 tidal constituent along transects Aa. The calculations were performed using the mean buoyancy frequency squared ($N^2$, in $s^{-2}$) obtained from CTD-$O_2$ data (ray in red) and NEMO-Amazon36 model data (ray in blue) for September. Grey areas represent local topography and black circles indicate the critical topography slope (ray generation sites). Panel also show along the transects Aa: along-shore mean total currents ($u_l$, in m $s^{-1}$) from ADCP (Dashed black lines), potential density from CTD-$O_2$ (grey contours), and dissipation rate profiles ($\varepsilon$, in W $kg^{-1}$, on a logarithmic scale) from the VMP (vertical colored bars). Subpanels within each panel illustrate the $N^2$ profiles from AMAZOMIX (red line) and the NEMO-Amazon36 model (blue line) used for ray-tracing calculations. Upper Thermocline Depth (UTD, dotted lines) and Lower Thermocline/Pycnocline Depth (LTD/LPD, dashed lines) are also indicated.

-Nutrients fluxes : profiles of Kz and nutrients fluxes are displayed. I don't see the point in giving values of nutrients fluxes without showing the concentration profiles and introduce the motivations and the issues.

R: Thanks for your comment.

Indeed, nutrient concentration profiles were analyzed prior to calculating nutrient fluxes.

Ultimately, we decided to remove and reserve all sections on "Nutrients fluxes" for a separate paper in progress.

-Discussion and conclusion : needs to be re-written and with convincing results for most part of it, one example : l442 « shear instabilities stronger >10-4s-2 », IT shear : high tidal modes are referred to but not shown etc

R: Thanks for your comment.

We have revised and rewritten the "Discussion and Conclusion" section, with updates found between lines 551–569 of the revised manuscript.
The tidal modes identified in the baroclinic tidal current time series (see figure RC2.11 below for an example) are presented in the section "Results" of the revised manuscript.

[Figure]

figure 11: Semi-diurnal baroclinic zonal currents ($u$", in m s-1) from the ADCP for stations (a) S10. Panel (a) also displays the buoyancy frequency squared (N2, in s-2) represented by vertical black lines, potential density represented by grey contours, and dissipation rate profiles ($\varepsilon$, in W kg-1, on a logarithmic scale) represented by vertical colored bars.

Revisions of section "Discussion and Conclusion" can be found below:
.
"

**4 Discussion and Conclusion**

The AMAZOMIX 2021 cruise provided, to the best of our knowledge, for the first time, direct measurements of turbulent dissipation using a velocity microstructure profiler (VMP) at multiple stations both inside and outside the influence of ITs. These measurements enabled the study of mixing processes at the Amazon Shelf break and the adjacent open ocean. To capture a full tidal cycle, data on turbulent dissipation rates, hydrography, and currents were collected alternately over 12 hours, with 4 to 5 profiles taken per station (see section 2). The locations of the 12-hour sampling stations were selected based on modeling results that provided realistic maps of IT generation and propagation (Fig. 1a; Tchilibou et al., 2022). Stations were located in the most energetic regions of IT, specifically at sites Aa, Ab, and D, covering stations S2 to S14, as identified in previous studies (Magalhaes et al., 2016; Tchilibou et al., 2022; Assene et al., 2024). Stations S19 to S21 were positioned in less energetic IT generation areas at site E, while stations S24 and S25 were located outside the influence of the IT fields (site G). Stations were distributed across different areas, including the shelf (e.g., S4, S9, and S19), the shelf-break (e.g., S3, S6, and S10), and the open ocean (e.g., S14, S24, and S20).

**Vertical Displacement, homogeneous layers**

The results revealed that, over a semi-diurnal tidal cycle, relevant amplitudes of vertical displacements (up to 60 m in length) and pronounced step-like structures (up to 40 m thick) were observed along transects Aa and Ab. In contrast, smaller and thinner structures were identified along other transects, such as E. These differences are likely related to the propagation of ITs, which induce vertical displacements at tidal frequencies and promote mixing by creating homogeneous layers visible as step-like features in the density structure. The isopycnal displacements and step-like structures observed within the pycnocline are consistent with findings from other IT regions (e.g., Stansfield et al., 2001; Simpson and Sharples, 2012; Bordois, 2015; Koch-Larrouy et al., 2015; Zhao et al., 2016; Bouruet-Aubertot et al., 2018; Xu et al., 2020). Furthermore, IT propagation appears to have stronger energy along transects Aa and Ab compared to others, consistent with prior modeling studies (Tchilibou et al., 2022; Assene et al., 2024).

**Direct measurements of dissipation rate**

Dissipation rates measured with the VMP ranged from between $[10^{-10}, 10^{-5}]$ W kg$^{-1}$ below the XLD, spanning from the continental shelf to the open ocean. The XLD was found to be considerably larger than the MLD at all stations, except at S8, S10, and S25. This is consistent with regions exhibiting strong subsurface shear, such as the equatorial ocean and western boundary current areas (Noh and Lee, 2008). The exception observed at other stations may reflect larger mixing events that were not captured by the VMP measurements.

The highest dissipation rates, within $[10^{-6}, 10^{-5}]$ W kg$^{-1}$, were observed primarily at generation sites Aa, Ab, and D (e.g., at stations S6, S10, and S3). Slightly lower but still substantial dissipation rates, ranging from $10^{-8}$ to $10^{-7}$ W kg$^{-1}$, occurred a few kilometers (~40 km) from these generation sites (e.g., at S11 and S7), along IT pathways (e.g., at S12, S13, and S20), and even in regions farther from IT influence (e.g., at S24). Interestingly, dissipation rates were higher within $[10^{-7}, 10^{-6}]$ W kg$^{-1}$ in the open ocean, such as at station S14, located ~230 km from generation site Aa, as summarized in Fig. 9.

Similarly, the vertical eddy diffusivity coefficient, ranging from $10^{-3}$ to $10^{-1}$ m$^2$ s$^{-1}$, was highest at the shelf-break (at stations S3, S5, and S10), Away from the shelf-break, diffusivity values were lower but still substantial, within $[10^{-4}, 10^{-3}]$ m$^2$ s$^{-1}$ (e.g., at S2, S7, and S11).

[Figure]

*Figure 9: summary diagram illustrating the key processes driving mixing along the AMAZOMIX transects (e.g., Aa and Ab). At IT generation sites (e.g., S6 and S10), mixing rates are stronger, with ITs contributing around 65%, compared to mean circulation (NBC). Along IT pathways (e.g., S7 and S11), mixing decreases but remains notable, driven by nearly equal contributions from ITs and mean circulation. A key observation is the increased mixing ~ 230 km from two distinct IT generation sites at the shelf break. This hotspot at S14 coincides with the surfacing of IT rays from different sites and the presence of ISWs, suggesting that constructive interference of IT rays may generate ISWs, amplifying mixing at S14.*

In comparison, in other regions, dissipation rates measured by similar VMP instrument are found between $[10^{-7}, 10^{-5}]$ W kg$^{-1}$ in the IT generation zone of Halmahera Sea, Indonesia (Koch-Larrouy et al., 2015; Bouruet-Aubertot et al., 2018), of Kaena Ridge, Hawaii (Klymak et al., 2008) and off the Changjiang Estuary (Yang et al. 2020). Whereas it is $[10^{-10},10^{-8}]$ W.kg$^{-1}$ along the IT path in the Southern Ocean (Gille et al., 2012) and in Halmahera Sea (Bouruet-Aubertot et al., 2018). Direct estimates of dissipation are almost $[10^{-11}, 10^{-10}]$ W kg$^{-1}$ far from IT influence (Koch-Larrouy et al., 2015; Bouruet-Aubertot et al., 2018) or under the influence of geostrophic current (Takahashi and Hibiya, 2019).

Our mixing coefficients are consistent with, the annual mean between $[10^{-4}, 10^{-3}]$ m$^2$ s$^{-1}$ of Ffield and Gordon (1992) or Koch-Larrouy et al. (2007), and aligned with others previous studies using the microstructure data (e.g. Tian et al., 2009; Koch-Larrouy et al., 2015; Bouruet-Aubertot et al., 2018; Xu et al., 2020), or modeling results (e.g. Koch-Larrouy et al., 2007).

This crucial vertical eddy diffusivity close enough to the surface along the IT paths may play a role in modulating heat (e.g., Assene et al., 2024) and chlorophyll content (de Macedo et al., 2023; M'Hamdi et al., 2024; in preparation) observed off the Amazon shelf.

Our study also found the highest dissipation rates at stations S3 and S5 of $[10^{-6},10^{-4}]$ W kg$^{-1}$ on the Amazon shelf , increasing near the bottom boundary layer. These findings compare well with values reaching up to $10^{-9}$ W kg$^{-1}$ within a kilometer of the seabed in the Southern Ocean (Sheen et al., 2013) and up to $10^{-6}$ W kg$^{-1}$ within a few meters from bottom topography off the Changjiang Estuary (Yang et al. 2020). This may indicate the

presence of an active bottom boundary layer. Thus, kinetic energy of bottom flow was estimated using friction velocity, that was computed from total velocity averaged over the bottom-most 15 m for shallow stations. It showed bottom friction energy stronger between 16-35 J m$^{-2}$ at S3 and S5 mainly and lower (< 3 J m$^{-2}$) in the other stations on shelf (e.g., at S8). These results are smaller but still important on the Amazon shelf and comparable to values (517 kJ m$^{-2}$) in the Drake Passage region (on the continental slope) of the Southern Ocean (Laurent et al., 2012). The bottom mixing at S3 and S5 can indirectly exert a control on pycnocline mixing on the Amazon shelf (Inall et al., 2021).

**Contribution of Background circulation and ITs to mixing**
**Mean baroclinic current shear**
Another important aspect addressed in this study was quantifying the contributions of different processes to the observed heterogeneous mixing.

First, the mean baroclinic current (BC) was considered as a proxy for the background circulation. The BC was predominantly structured into a northwestward surface flow and a southeastward subsurface flow along the IT pathways. The strong surface flow toward the northwest is associated with the North Brazil Current (NBC), which originates from the northeastern coast of Brazil (e.g., Bourlès et al., 1999) and propagates along the Amazon shelf-break (e.g., at stations S7, S10, S11, S14, and S24). Conversely, the southeastward subsurface flow observed at stations such as S7 and S11 might result from NBC instability or the presence of a countercurrent at depth (Dossa et al., 2024, in preparation). At site E, the flow reversal observed at S21 - characterized by a southeastward surface flow and a northwestward subsurface flow - was located inside of the outer path of the Amazon plume. This reversal could be related with the influence of AWL formed by continental inputs (Prestes et al., 2018).

Both baroclinic flows demonstrated a significant potential for shear instability, with vertical shear ranging from 10$^{-5}$ to 10$^{-3}$ s$^{-2}$ off the Amazon shelf. The shear associated with the NBC was particularly pronounced around the pycnocline (between 40 and 200 m depth) at sites Aa, Ab, and G (e.g., at S6, S7, S10, S11, S14, and S24). At site E, the shear instability was stronger (> 2.5 x 10$^{-4}$ s$^{-2}$) at the base of the pycnocline (e.g., at S20), potentially associated with NBC retroflection near [5–6°N, 50°W] during the fall season (Didden and Schott, 1993). The higher BC shear observed at S21, where flow direction reversals occurred, could be associated with the presence of a subsurface cyclonic eddy (Dossa et al., 2024, in preparation).

**ITs shear**
Second, the baroclinic tidal current was extracted from the total baroclinic current, revealing significant semi-diurnal (M2) component signals around the pycnocline. These signals, characterized by higher tidal modes (3-5), were more pronounced at generation and propagation sites Aa and Ab (e.g., at S6, S10, and S14) compared to other sites. The tidal shear within the pycnocline layer (80-120 m) is consistent with the observed IT signal patterns and large vertical displacements. It was stronger, reaching up to 10$^{-3}$ s$^{-2}$, near the generation sites Aa and Ab (at S6 and S10) and in the open ocean at S14. Further from the generation sites (e.g., at S7, S11, and S20), the IT shear was smaller but still notable (reaching up to 10$^{-4}$ s$^{-2}$). This highlights the significant role of ITs in driving mixing processes, particularly within the pycnocline, where strong vertical shears were

observed near the shelf-break compared to regions far away. Outside the IT fields, such as at S24, the persistent high vertical shear near the bottom topography could be attributed to the active bottom boundary layer (Inall et al., 2021).

**IT/BC ratio**

Both IT and BC shear contribute to mixing, with their relative dominance varying across sites. Near the generation sites on the shelf-break, IT shear dominated the IT/BC shear ratio, such as at S6 (61.44/38.56 %), S10 (65.82/34.18 %), and S21 (58.55/41.45 %). Along the IT paths, the contributions were nearly equal (~50/50 %) at locations farther from the generation sites (e.g., at S20, S7, S11, and S13), except at S14 in the open ocean, where IT shear remained dominant (58.50/41.50 %). These findings align with the presence of ITs at generation sites Aa, Ab, and E (Tchilibou et al., 2022; Assene et al., 2024) and the stronger energy associated with NBC cores, particularly at S7 and S11.

These results are consistent with previous studies that identified strong tidal shear near IT generation sites, such as the Halmahera Sea (Bouruet-Aubertot et al., 2018), the Changjiang Estuary (Yang et al., 2020), the northwest European continental shelf seas (Rippeth et al., 2005), and the southern Yellow Sea (Xu et al., 2020).

The most relevant finding of this study was the relative increase in mixing within the pycnocline layer, observed at S14 in the open ocean, far from the IT generation sites.

**Discussion on the strong mixing at S14**

Along the IT paths, elevated remote dissipation rates (within $[10^{-7}, 10^{-6}]$ W kg$^{-1}$) were identified ~ 230 km from the shelf-break at S14.

This region is well known for intense IT dissipation, as shown by a realistic model (Tchilibou et al., 2022; Assene et al., 2024), and for the highest occurrences of ISWs generated by ITs (Fig. 1a; de Macedo et al., 2023), with large-amplitude ISWs exceeding 100 m clearly visible in satellite records (Brandt et al., 2002).

At station S14, where relative mixing increases, IT rays surfacing from two distinct IT generation sites coincide with the appearance of ISWs and mark the location where the NBC vanishes.

This region of wave-wave interactions can lead to the constructive interference of IT rays, potentially facilitating the emergence of higher tidal modes (New & Pingree, 1992; Silva et al., 2015; Barbot et al., 2022; Solano et al., 2023). These higher modes, in turn, could enhance the generation of nonlinear ISWs (e.g., Jackson et al., 2012) and contribute to the elevated dissipation rates (Xie et al., 2013), as observed at this station.

Moreover, IT interactions with baroclinic eddies may also contribute to turbulent dissipation (Booth and Kamenkovich, 2008), particularly in this area of pronounced eddy activity. However, no repeated AMAZOMIX stations observed during a tidal period were enclosed by mesoscale eddy activity, except potentially around site E, where possible evidence of a subsurface eddy was detected at S21 (Dossa et al., 2024, in preparation).

Future studies are needed to unravel the intricate interplay among these processes. The data collected during the AMAZOMIX cruise will provide a guide for improving our understanding and advancing parameterizations for modeling studies.

"

As a conclusion, I think this manuscript is no ready for publication and needs significant work to produce convincing results. I think the most efficient way to proceed is a new submission.

---

## Author Response (AR2)

**For Anonymous referee #1**

**We thank this Reviewer for thoughtful and constructive comments on our manuscript. We appreciate the time s/he invested in the review. We believe that our revised manuscript addresses all the comments. In this regard, we have revised and rewritten a few sections such as Abstract, method, results, discussion and conclusion in the revised manuscript. We thought it useful to point out its detailed revisions (lines and sections) in the reply to your comments. Below (highlighted in blue and magenta) is an itemized response to the different issues raised in the review.**

Turbulent dissipation from AMAZOMIX off the Amazon shelf along internal tides paths

Fabius Kouogang et al.

The measurements include single microstructure profiles at a variety of stations inside and outside of a modelled tidal beam on the slope near the mouth of the Amazon River.

There are also 25 CTD stations and ship-based ADCP. The authors have tried to examine the difference in turbulence levels in- and outside of this tidal beam. They have mainly examined their ship-based sections with a few stations per section. In my view, the sparse data makes their task very difficult. Many statements in the manuscript are unsupported by the figures: *e.g., mixing is higher in the tidal beams, mixing is higher where tidal energy is higher, mixing is higher with low Ri, mixing is higher where internal tide rays cross, and so on.* While it is possible these are true statements, the chosen approach has not delivered a clear result.

Perhaps a way forward is to do some more averaging and establish 5 averaging areas. These are in the tidal beam: (1) slope, (2) offshore, and (3) internal solitary wave station. And outside the tidal beam: (4) slope and (5) offshore. So rather than make sections, the authors should make scatter plots like Fig 7 but using data from these groups and outside the surface and bottom boundary layers (SBL and BBL) and see if they can find something that is actually related to the dissipation. For example, find all the stations in the beam and calculate mean epsilon. Is this higher than stations out of the beam based on the model? Maybe compare (1) and (4) and also compare (2) and (5). Compare (1, 2, 3) vs (4-5). And so on. Scatter plots could be based on observed tidal amplitudes, current speed, lateral gradients of density or velocity.

R: Thank you for your feedback.
We have implemented your suggested approach, which yielded clear and useful results. While we selected particularly informative stations, we found the "IN" and "OUT" tidal beam classification unsuitable because the so-called "OUT" stations are actually located within a channel, and their limited number prevented meaningful analysis.
Instead, we focused the analysis on 3 distinct paths based on modeled M2 baroclinic tidal flux: 2 high tidal energy (HTE) and 1 low tidal energy (LTE) paths. We established five averaging regions along these paths:

HTE paths: (1) slope, (2) offshore, and (3) internal solitary wave station

We have subsequently regenerated the scatter plots using midwater data of these groups (below the maximum of [XLD, MLD] and above the bottom boundary layer). These analyses are presented and explained in Section 3.2.3 (lines 355-396) and Figure 10 and 11 of the revised manuscript, with the results shown below:

"

**3.2.3 Competitive processes to generate mixing**

Our aim in this subsection is to associate midwater mixing events with either baroclinic tidal currents or time-averaged (mean) currents. To achieve this, we map depth-integrated and maximum values of station-averaged $\varepsilon$ and plot all $\varepsilon$ values on a (time-mean shear , tidal shear ) diagram across five regions ($A_s$, $A_o$, $A_{isw}$, $E_s$, and $E_o$; Figs. 10 and 11). These regions are selected to contrast slope and open-ocean dynamics, with data included from the HTE and LTE transects. All data are collected from below the wind-influenced surface layer (defined as the maximum of XLD or MLD; see subsection 2.2.1) and above the friction-dominated bottom boundary layer ($H_{BB}L$; defined in subsection 2.2.1).

Mixing hotspots ($\varepsilon = [10^{-6}, 10^{-7}]$ W kg$^{-1}$; magenta and red circles in Fig. 11 and Fig. 10) are observed under strong vertical baroclinic shear ($[10^{-4}, 10^{-5}]$ s$^{-2}$), driven by either tidal or time-mean currents.

On the slope ($A^s$ and $E^s$), high $\varepsilon$ values in $A_s$ are associated with stronger  than  (magenta, red, and grey stars in Fig. 11a correspond to  $\approx 10\text{-}4 > $  $\approx 10^{-5}$), indicating that tidal shear explains ~60% of high $\varepsilon$ values (Table A3, Appendix C). Similarly, in $E_s$, moderate $\varepsilon$ values (yellow and grey stars in Fig. 11d) are primarily driven by tidal shear, which accounts for ~60% of the observed mixing (Table A3, Appendix C).

In the open ocean ($A_o$, $E_o$, and $A_{isw}$), moderate $\varepsilon$ values in $A_o$ and $E_o$ are found when  is nearly equal to  (yellow, red, and grey stars in Fig. 11b and 11e correspond to  $\approx$  $\approx 10^{-4}$ s$^{-2}$), suggesting tidal and time-mean shear each contribute ~50% to mixing (Table A3, Appendix C). An exception is observed in $A_{isw}$, where high $\varepsilon$ values coincide with slightly stronger tidal shear (red and grey stars in Fig. 11c correspond to  $\approx 2$ x  $\approx 2$ x $10^{-4}$ s$^{-2}$), suggesting that tidal shear explains ~60 % of mixing hotspots (Table A3, Appendix C).

These results suggest that mixing on the slope is slightly dominated by ITs, while offshore mixing is equally balanced by mean circulation and ITs. However, exceptions exist in the open ocean, particularly at stations $A_{isw}$ and , where tidal shear contributes ~60% and ~30% to mixing, respectively. The mixing at  is attributed to NBC. A key question remains: why does $A_{isw}$ exhibit strong IT-driven mixing ~230 km from IT generation sites, with mixing hotspots observed at various depths throughout the water column? To address this, we employ ray-tracing techniques to investigate potential IT propagation paths.

[Figure]

*Figure 10: Depth-integrated (in mW kg⁻¹, logarithmic scale) and maximum values (in W kg⁻¹, logarithmic scale) of station-averaged dissipation rates (ε) from VMP measurements during the AMAZOMIX 2021 cruise. Solid black lines depict transects (, , and E) along high tidal energy (HTE) and low tidal energy (LTE) paths. Data are from below the wind-influenced surface layer and above the friction-dominated bottom boundary layer. Colored circles and stars represent short and long stations, respectively. Small and large colored circles indicate depth-integrated and maximum values of ε, respectively, with ranges shown by the color bar. Similarly, small and large colored stars indicate depth-integrated and maximum values of ε, respectively, with ranges shown by the color bar. Stations are grouped into five areas: $A_s$ ( and ), $A_o$ (, , , and ), $A_{isw}$ ($A_{isw}$), $E_s$ ($E_s$), and $E_o$ ($E_o$). The five blue boxes indicate these defined areas. Subscripts denote locations: "s" for slope ($A_s$), "o" for offshore ($A_o$ and $E_o$), and "isw" for ISW regions ($A_{isw}$).*

[Figure]

*Figure 11: Dissipation rates (ε, in W kg⁻¹, logarithmic scale), measured below the wind-influenced surface layer (max [XLD, MLD]) and above the friction-dominated BBL (H$_{BBL}$), plotted as a function of the mean baroclinic vertical shear squared (, in s⁻², logarithmic scale) and semi-diurnal baroclinic vertical shear squared (, in s-2, logarithmic scale). Data are from defined areas: (a) A$_s$ ( and ), (b) A$_o$ (, , , and ), (c) A$_{isw}$ (A$_{isw}$), (d) E$_s$ (E$_s$), and (e) E$_o$ (E$_o$). ε are represented by colored circles, with their ranges indicated on the color bar. Each panel also includes vertical shear averages for specific ε ranges ([10⁻⁶], [10⁻⁷], [10⁻⁸], [10⁻⁹], and [10⁻¹⁰] W kg⁻¹), depicted as colored stars with black edges, grey stars with black edges represents the vertical shear averaged across all ε values. Dashed grey lines are included for comparison.*

*"*

This is a complicated region with a strong mean flow, fronts, eddies, strong river outflow, and strong tides. So it will be a difficult task to come up with a simple explanation based on 25 stations over a wide area.
 In other words, summary Fig 9 is not well supported but could be true. So there could be 5 groups of stations with sufficient averaging, arranged in a logical manner as opposed to 25 stations somewhat randomly named on 5 transects again without fully logical naming with limited statistics. I have suggested one approach above but the authors could come up with something else.

R: Thanks for your comments.
This region exhibits complex dynamics involving multiple interacting processes. Rather than analyzing stations individually, we have opted to separate the contributions of mean currents and tidal currents at each station.

Additionally, to better support our analyses and results, we have revised the summary figure (see below Figure RC1.1).

The sites/transects and stations names were systematically re-named and re-organized by location. Each site received a unique identifier based on its position along the HTE and LTE paths. Stations were categorized by site and region: superscripts 'a' and 'b' denoted stations at sites $A^a$ and $A^b$, respectively, while subscripts indicated location—'sh' for shelf, 's' for slope, 'o' for offshore/open ocean, and 'isw' for ISW regions. This structured naming system ensured clear identification and logical grouping of stations for consistent data analysis. This naming system is corrected through the revised manuscript and reported below:

"

**Appendix A**

The AMAZOMIX measurement sites and stations were systematically named and organized by location. Each site received a unique identifier based on its position along the HTE and LTE paths. Stations were categorized by site and region: superscripts "a" and "b" denoted stations at sites  and , respectively, while subscripts indicated location–"sh" for shelf, "s" for slope, "o" for offshore/open ocean, and "isw" for ISW regions (Table A1). This structured naming system ensured clear identification and logical grouping of stations for consistent data analysis.

**Table A1:** The naming system of the AMAZOMIX cruise measurement sites and stations.

| Paths / Transects | Sites | Stations | | | | | | | |
|---|---|---|---|---|---|---|---|---|---|
| | | Shelf | | Slope | | Offshore/Open ocean | | | ISWs |
| **High Tidal Energy (HTE) paths** | $A^a$ | $A^a_{sh_1}$ | $A^a_{sh_2}$ | $A^a_s$ | | $A^a_{o_1}$ | $A^a_{o_2}$ | $A^a_{o_3}$ | $A_{isw}$ |
| | $A^b$ | $A^b_{sh}$ | | $A^b_{s_1}$ | $A^b_{s_2}$ | $A^b_o$ | | | |
| **Low Tidal Energy (LTE) path** | $E$ | $E_{sh}$ | | $E_s$ | | $E_o$ | | | - |

"

[Figure]

Figure RC1.1: Summary diagram illustrating the key processes driving mixing across the HTE paths ($A^a$ and $A^b$) and LTE path (E) off the Amazon shelf . At IT generation sites (stations $A^a_s$, $A^b_{s_2}$, and $E_s$), mixing is generally stronger (red zigzags), except at $E_s$, where it is moderated (yellow zigzags). At these generation sites, ITs contribute ~60% of the mixing, exceeding the contribution of the mean circulation (NBC). Away from generation sites in the open ocean (e.g., $A^b_o$ , $A^a_{o_1}$, and $E_o$; yellow zigzags), mixing decreases but remains substantial, driven by nearly equal contributions from ITs and mean circulation. A key observation is the increased mixing ~230 km from the generation sites, forming a hotspot at $A_{isw}$ (orange zigzags). This coincides with: the surfacing of IT rays (blue lines) from two distinct generation sites on the HTE paths, the vanishing of the NBC (sky-blue shaded areas), and the presence of ISWs (magenta lines). These observations suggest that constructive interference of IT rays may generate ISWs, amplifying mixing at $A_{isw}$.

"

There is a lot of extraneous material in this manuscript. Basically everything in the paper that does not support Fig 9 should be eliminated from the paper. Fig 9 should be presented much earlier. Perhaps as Fig 1b. The current Fig 1b-c could be moved later.

R: Thanks for your remarks.
While the dataset is extensive, this paper serves as the foundational study for the Amazomix database and will provide a reference framework for future research. To this end, we have allocated a dedicated portion of the database for subsequent studies.
In line with this objective, we have reorganized the summary figure as mentioned earlier. Additionally, we have split Figure 1 (of revised manuscript) into two separate figures (shown below; Figure RC1.2-3) to improve clarity and presentation.

[Figure]

**Figure RC1.2**: Map of a part of the AMAZOMIX 2021 cruise off the Amazon shelf, showing bathymetric contours (100 m, 750 m, 2000 m, and 3000 m isobaths) in gray. Magenta arrows show the 25-hour mean depth-integrated baroclinic IT energy flux (September 2015, from the NEMO model) originating from IT generation sites ($A^a$, $A^b$, and E) along the shelf break. Solid black lines depict transects ($A^a$, $A^b$, and E) defined on the high tidal energy (HTE) and low tidal energy (LTE) paths. The solid brown line represents the NBC pathways, illustrating background circulation. Shattered colored lines highlight ISW signatures. Colored circles and stars indicate short and long CTD-O$_2$/L-S-ADCP stations, respectively, with the corresponding sampling dates represented by the color bar. The superscripts 'a' and 'b' on station names correspond to sites $A^a$ and $A^b$, respectively. The subscripts 'sh', 's', 'o', and 'isw' indicate station locations: shelf ($A^a_{sh_1}$, $A^a_{sh_2}$, $A^b_{sh}$, and E$_{sh}$), slope ($A^a_s$, $A^b_{s_1}$, $A^b_{s_2}$, and E$_s$), open ocean ($A^a_{o_1}$, $A^a_{o_2}$, $A^a_{o_3}$, $A^b_o$, and E$_o$), and ISW regions (A$_{isw}$) for sites $A^a$, $A^b$ and E, respectively.

[Figure]

Figure RC1.3 : a) 1A Sentinel image acquired on 12th September 2021, showing ISW signatures. b) Tidal (M2 and S2) amplitude of the currents (at -45.5°W, 1°N) derived from the FES2014 model (Lyard et al., 2014). ISW signature dates are marked by red bars.

"

If the manuscript emphasizes internal tide-related mixing, then the focus should likely be on mid water. BBL and SBL mixing could be related to other processes and will cloud relationships between internal tide shear/strain and dissipation. Strain = d(displacement)/dz. Shear and strain are related to turbulence as shown by Gregg (1989) and papers that follow.

R: Thanks for your comments.
We have focused our analyses on midwater mixing (i.e., below maximum of [XLD, MLD] and above the bottom boundary layer). This approach is detailed in previous responses and further clarified in the revised manuscript (Subsection 2.2.1, lines 155-165; Subsection 3.2.3, lines 355-396;)

In this revision, the microstructure methods have been updated nicely and closely follow community standards.

R: Thanks

The summary compares the observed turbulence to a wide variety of measurements around the world. I would be interested in a more narrow view. There are other papers related to this AmazonMix project. Do the results in this manuscript help us to understand any of the previous finding from previous AmazonMix papers?

R: Thanks for your remarks
A direct comparison with previous studies from the AMAZOMIX project was not possible, as this represents the first published work in the series. However, several companion papers are currently in preparation or under submission, some of which have been cited in this study (e.g., Mhamdi et al., in preparation; Dossa et al., to be submitted this year).

The appendix is 23 figures and zero text. This is one relatively major example, which leads me to believe that the first author has not received appropriate guidance from the senior authors of this manuscript. There are further examples. The senior authors should see such obvious organizational issues and get them fixed before submitting a revised manuscript to a journal. I raised a similar point in the previous review with seemingly limited success.

R: Thanks for your comments
We acknowledge the extensive supplementary materials. To facilitate future research, we have allocated a dedicated portion of this database for subsequent publications.
We have added the additional supporting information into appendices in lines 549-588 of revised manuscript and reported below.
The manuscript is currently under collaborative review by senior scientists together with the first author.
"

**Appendix A**

The AMAZOMIX measurement sites and stations were systematically named and organized by location. Each site received a unique identifier based on its position along the HTE and LTE paths. Stations were categorized by site and region: superscripts "a" and "b" denoted stations at sites  and , respectively, while subscripts indicated location–"sh" for shelf, "s" for slope, "o" for offshore/open ocean, and "isw" for ISW regions (Table A1). This structured naming system ensured clear identification and logical grouping of stations for consistent data analysis.

*Table A1: The naming system of the AMAZOMIX cruise measurement sites and stations.*

| Paths / Transects | Sites | Stations | | | | | | | |
|---|---|---|---|---|---|---|---|---|---|
| | | Shelf | | Slope | | Offshore/Open ocean | | | ISWs |
| **High Tidal Energy (HTE) paths** | $A^a$ | $A^a_{sh_1}$ | $A^a_{sh_2}$ | $A^a_s$ | | $A^a_{o_1}$ | $A^a_{o_2}$ | $A^a_{o_3}$ | $A_{isw}$ |
| | $A^b$ | $A^b_{sh}$ | | $A^b_{s_1}$ | $A^b_{s_2}$ | $A^b_o$ | | | |
| **Low Tidal Energy** | $E$ | $E_{sh}$ | | $E_s$ | | $E_o$ | | | - |

| (LTE) path | | | | | |
|---|---|---|---|---|---|
| | | | | | |

**Appendix B**

To relate each mixing event with either tidal or mean (time-averaged) currents along the HTE transects ( and ) and the LTE transect (E), we quantify the relative contributions of mean and semi-diurnal baroclinic vertical shear squared at transect stations (see Table A2).

*Table A2: miXing Layer Depth (XLD), Mixed Layer Depth (MLD), Contribution (mean and standard deviation) of the Semi-diurnal (CSBS), and Mean Baroclinic (CMBS) Shear to total baroclinic shear.*

| Stations | XLD (m) | MLD (m) | CSBS (mean ± SD) (%) | CMBS (mean ± SD) (%) |
|---|---|---|---|---|
| | 27 | 25.0 | - | - |
| | 20 | 5.0 | - | - |
| | 57 | 17.8 | 66.2 ± 0.3 | 33.8 ± 0.3 |
| | 46 | 22.5 | 36.7 ± 3.7 | 63.3 ± 3.7 |
| | 23 | 44.5 | - | - |
| | 29 | 21.0 | - | - |
| | 26 | 32.5 | 60.0 ± 4.0 | 40.0 ± 4.0 |
| | 104 | 15.5 | 47.6 ± 4.9 | 52.4 ± 4.9 |
| | 75 | 11.3 | 56.6 ± 3.3 | 43.4 ± 3.3 |
| | 82 | 12.3 | 59.1 ± 3.4 | 40.9 ± 3.4 |
| $A_{isw}$ | 97 | 12.3 | 63.6 ± 4.8 | 36.4 ± 4.8 |
| $E_{sh}$ | 45 | 1.0 | - | - |
| $E_o$ | 73 | 1.8 | 56.6 ± 3.9 | 43.4 ± 3.9 |
| $E_s$ | 53 | 1.0 | 60.2 ± 2.8 | 39.8 ± 2.8 |

SD = Standard Deviation

**Appendix C**

Following subsection 2.2.2, we examined the cross-shelf component of baroclinic tidal currents to investigate IT amplitude (current strength) and shear instability around the pycnocline (70–180 m depth; see Table A3). Additionally, the along-shelf component of mean baroclinic currents (MBC) was analyzed to evaluate the strength of the mean flow and its associated shear instability in the upper 200 m (see Table A3).

*Table A3:* *Strength of baroclinic tidal and mean baroclinic currents.*

| Stations | IT amplitudes (cm s-1; maximum) | Estimated number of IT eigenmodes | IT vertical shears (s-2 x 10-4; maximum) | MBC velocities (cm s-1; maximum) | MBC vertical shear (s-2 x 10⁻⁴; maximum) |
|---|---|---|---|---|---|
| | 35 | 6-7 | 5.5 | 90 | 1.2 |
| | 15 | 4-5 | 2.5 | 98 | 1.7 |
| | 45 | 6-7 | 7.7 | 30 | 0.7 |
| | 25 | 4 | 2.0 | 90 | 1.2 |
| | 40 | 3 | 7.6 | 67 | 1.2 |
| | 25 | 3 | 3.3 | 69 | 1.3 |
| Aisw | 40 | 4.5 | 5.0 | 71 | 1.1 |
| Eo | 15 | 4 | 3.5 | 43 | 2.7 |
| Es | 20 | 4 | 1.2 | 28 | 0.8 |

"

Figures and captions on the same page are usual.

R: Thanks for your remark.
We have reorganized the figures to appear on the same page as their corresponding captions for improved readability and clarity.

The first lines in paragraphs on adjacent lines have an indent. Or skip a line and paragraphs can be left justified.
R: Thanks for your remark.
We have reformatted the opening lines of each paragraph to improve document structure and readability.

Maybe try running the text through some AI grammar checker. It's ok as is, but could be better.

R: Thanks for your comment.
We have reviewed the manuscript using AI-powered grammar checking tools to ensure linguistic accuracy and clarity.

--

Comment by line

Fig 1c - tidal height of what and where?

R: Thanks for your comments.
We have revised the caption in Figure 2 of the revised manuscript, as presented below:
"

 *b) Tidal (M2 and S2) amplitude of the currents (at -45.5°W, 1°N) derived from the FES2014 model (Lyard et al 2014).*

"

16- reserve "significant" for statistically significant and just drop significant and your sentence is still ok

R: Thanks for your comments.
Throughout the revised manuscript, we have systematically replaced non-technical uses of "significant" with alternative terms, reserving "significant" exclusively for cases of statistical significance.

30 - there is not further mention of parameterizations in this manuscript. Delete this sentence.
R: Thanks for your comments.
We have removed the sentence from the revised manuscript.

68 - units: 30% not 30[space]%. Still some funny units here and there- e.g., m.s-1 Check each figure.

R: Thanks for your comments.
We have revised units throughout the revised manuscript.

73 - The introduction is still a bit diffuse. There's a lot of info which seems good to know but it's unclear where the manuscript is going with all this info. I would favor a more narrow focus on this region and previous AmzonMix papers. Some more specific information on what was found in Bertrand (2021), what gaps that left, and how this manuscript fills those gaps. This can be short but it seems to be missing or maybe not clearly stated. Maybe it's something like: "The Amazon plume is highly variable with boundary currents, eddies, and internal tides. Bertrand showed something about mixing. However, he didn't really talk about internal tides. We show internal tides and nonlinear solitary waves produceda lot of mixing. So there are implications for some other things." There are other AmazonMix papers. How does this paper expand on/explain gaps in thoise papers?

R: Thanks for your comments.
This study represents the first peer-reviewed analysis of AMAZOMIX data, with particular emphasis on mixing induced by internal tidal waves. The work of Bertrand et al. (2021) just provides a valuable mission report containing raw VMP measurements. Their work: (1) has not been peer-reviewed, (2) presents uncalibrated data, and (3) lacks complete station coverage in its preliminary plots.

We have accordingly revised the Introduction section (lines 33-69), as presented below:

"

[revised manuscript text omitted]

75 - "Direct microstructure measurements of temperature, salinity and velocity were conducted..." Does this mean the instrument was equipped with fast thermistors, microconductivity probes, and shear probes? And are you going to talk about all these?

R: Thanks for your comments.
The measurement system incorporated three key sensors: fast-response thermistors, micro conductivity probes, and shear probes. We have simply rewritten this sentence (lines 65-66) in revised manuscript, as shown below:
"
Microstructure and hydrographic measurements were collected at repeated stations over an M2 tidal cycle (~12.42 hrs), providing dissipation estimates and insights into associated processes.
"

Fig 1- this is really about the worst naming of CTD stations I have ever seen in decades as a scientist. It is slightly better than completely random. Some stations increase onshore and some offshore. Sometimes they do both on the same transect. The letters do not all increase northward.

If you feel the need to retain this because you are attached to the cruise naming convention and previous papers, maybe you could add a supplement with this station naming and provide something logical so that a reasonable reader can follow along without having to look at Fig 1a literally all the time.

R: Thanks for your comments and suggestions.
We have implemented a standardized naming system for all stations, as detailed in previous responses. This structured approach enables clear identification and logical grouping of stations, ensuring consistency throughout our data analysis. The updated station names are presented below.

"

**Appendix A**

The AMAZOMIX measurement sites and stations were systematically named and organized by location. Each site received a unique identifier based on its position along the HTE and LTE paths. Stations were categorized by site and region: superscripts "a" and "b" denoted stations at sites and , respectively, while subscripts indicated location–"sh" for shelf, "s" for slope, "o" for offshore/open ocean, and "isw" for ISW regions (Table A1). This structured naming system ensured clear identification and logical grouping of stations for consistent data analysis.

*Table A1: The naming system of the AMAZOMIX cruise measurement sites and stations.*

| Paths / Transects | Sites | Stations | | | |
|---|---|---|---|---|---|
| | | Shelf | Slope | Offshore/Open ocean | ISWs |

| High Tidal Energy (HTE) paths | $A^a$ | $A^a_{sh_1}$ | $A^a_{sh_2}$ | $A^a_s$ | | $A^a_{o_1}$ | $A^a_{o_2}$ | $A^a_{o_3}$ | $A_{isw}$ |
|---|---|---|---|---|---|---|---|---|---|
| | $A^b$ | $A^b_{sh}$ | | $A^b_{s_1}$ | $A^b_{s_2}$ | $A^b_o$ | | | |
| Low Tidal Energy (LTE) path | $E$ | $E_{sh}$ | | $E_s$ | | $E_o$ | | | - |

"

This constant work is extremely distracting and seriously detracts from the manuscript. See line 258, for example, where XLD is noted as deeper than MLD except at stations S8, S10, and S25. S8 and S10 are neighboring stations near the slope in the south and S25 is way off to the north in deep water. Every sentence requires similar parsing. This is a simple example.

R: Thanks for your comments and suggestions.
We have systematically revised sentence structures throughout the manuscript to ensure consistent grammatical parsing and improved readability. This comprehensive editing approach maintains technical precision while enhancing clarity across all sections.

Here is a longer example from the discussion: "Stations were located in the most energetic regions of IT, specifically at sites Aa, Ab, and D, covering stations S2 to S14, as identified in previous studies (Magalhaes et al., 2016; Tchilibou et al., 2022; Assene et al., 2024). Stations S19 to S21 were positioned in less energetic IT generation areas at site E, while stations S24 and S25 were located outside the influence of the IT fields (site G). Stations were distributed across different areas, including the shelf (e.g., S4, S9, and S19), the shelf-break (e.g., S3, S6, and S10), and the open ocean (e.g., S14, S24, and S20)." It's complicated and does not have to be.

R: Thanks for your remarks.
We have accordingly revised the discussion section (lines 429-545) in the revised manuscript.

105 - what was done with the pre- and post-cruise calibrations?

R: Thanks for your remarks.
Pre- and post-cruise CTD-$O_2$ calibrations were conducted to ensure accurate dissolved oxygen measurements during the survey. This is now specified in lines 105-106 of the revised manuscript, as shown below:
"

The 24 Hz CTD-O2 sensors were calibrated before and after the cruise to ensure accurate dissolved oxygen measurements throughout the survey.
"

106 - means?
R: Thanks for your remarks.
we have rewritten the sentence in line 107 of the revised manuscript, as shown below:
"
The temperature, salinity, and oxygen standard deviation between the CTD-O$_2$ and the bottle samples was 0.003 °C, 0.003 PSU, and 0.05 ml l$^{-1}$, respectively.
"

107 - lag effects- does this refer to S spikes from mismathed times of T and C?
R: Thanks for your remarks.
Indeed, lag effects can produce salinity spikes due to temporal mismatches between temperature and conductivity measurements..

123 - Great
R: Thanks

151- tracer variance not energy
R: Thanks for your remarks.
We have decided to reorganize this subsection ("the vertical eddy diffusivity coefficient") in the revised manuscript.

152-153 - unclear
R: Thanks for your remarks.
We have decided to reorganize this subsection ("the vertical eddy diffusivity coefficient") in the revised manuscript.

161 - Gregg (2003) showed mixing is dramatcially reduced with decreasing latitude for a given internal wave energy level compared to higher latitudes.
R: Thanks for your remarks.
Yes, of course.

170 - "XLD is specified as the depth where ε drops from its first minimum value." I don't understand this definition.
R: Thanks for your remarks.
Indeed, this definition means that XLD is defined as the depth at which ε decreases from its first minimum value, e.g., 10$^{-9}$ W kg$^{-1}$ at S7.

186 - define along-/across-shore
R: Thanks for your remarks.
We have defined in lines 180-185 of the revised manuscript, as reported below:
"

The baroclinic mean velocities $[\overline{u'}, \overline{v'}]$, calculated to estimate mean circulation along IT paths, are decomposed into along-shore $\overline{u'}_l$ and cross-shore $\overline{u'}_c$ velocities. The overbar denotes the average over the $M_2$ tidal period. Similarly, the components $[u'', v'']$ are decomposed into along-shelf $u''_l$ and cross-shelf $u''_c$ velocities. The along-shelf velocity component is defined parallel to the 200 m isobath (treated as the coastline), with positive values indicating northwestward flow and negative values indicating southeastward flow. The cross-shelf velocity component is defined perpendicular to the 200 m isobath, with positive values indicating northeastward flow and negative values indicating southwestward flow.

"

188 - alternately?
R: Thanks for your remarks.
Indeed, CTD-O₂ measurements were performed alternately. We have removed the terminal word in line 187 of the revised manuscript, as shown below:
"

The similar processing is applied to the CTD-O$_2$ data.

"

200 - energy is E, dissipation is epsilon
R: Thanks for your remarks.
We have decided to remove this sentence in the revised manuscript..

204 - "measured bathymetry from CTD-O2"- what?
R: Thanks for your remarks.
Indeed, these represent bathymetry measurements at sampling stations.
We have revised the description in lines 166-188 of the revised manuscript.

206 - In this ratio the N's cancel- I don't understand this. Why N*S? Is S either S or S^2? See line 208.
R: Thanks for your remarks.
The formulation in lines 196-197 of the revised manuscript has been revised as follows:
"

The individual contributions of semi-diurnal ITs and mean circulation are then expressed as follows: $S^{2''}/(\overline{S^{2'}} + S^{2''})$ for tidal contribution and $\overline{S^{2'}}/(\overline{S^{2'}} + S^{2''})$ for mean circulation contribution.

"

225 - combined not glued. Has some ffort been made to avoid discontinuities in the profiles?

R: Thanks for your comments
Specific efforts were made to minimize profile discontinuities in the measurements.

226 - there needs to be some minimal description of this model
R: Thanks for your comments

We have added some minimal description of this model in line 216-225 of the revised manuscript, as reported below:
"

Amazon36 is a specific configuration, specifically designed to cover the western tropical Atlantic from the mouth of the Amazon River to the open sea (see Tchilibou et al., 2022; Assene et al., 2024; for configuration details and model description). The NEMO model's fine horizontal resolution (1/36°) and 75 vertical levels allow for accurate simulation of low-mode ITs generated along the Brazilian shelf break. Key inputs include bathymetric data from the 2020 General Bathymetric Chart of the Oceans, surface forcing from ERA-5 atmospheric reanalysis (Hersbach et al., 2020), and river runoff data from the ISBA (Interaction Sol-Biosphère-Atmosphère; https://www.umr-cnrm.fr/spip.php?article146&lang=en) model. Open boundary conditions were driven by 15 major tidal constituents (M2, S2, N2, K2, 2N2, MU2, NU2, L2, T2, K1, O1, Q1, P1, S1, and M4) and barotropic currents from the FES2014 atlas (Lyard et al., 2021), supplemented by temperature, salinity, and velocity data from the MERCATOR-GLORYS12v1 assimilation product (Lellouche et al., 2018).

"

Fig 3- What is plotted? Two selected profiles about 6 hrs apart to emphasize displacements at each station? Up- and downcasts? This is not necessarily an indication of mixing. A linear wave can displace isopycnals. Displacements can be calculated as (rho - mean rho)/(vertical density gradient) and plotted as a figure with shaded colors. Then you can also calculate strain = d(displacement)/dz which will highlight finer scales and maybe show internal wave propagation along the section

R: Thanks for your comments.
This figure (updated and shown below) in the revised manuscript plots, for long stations, two selected profiles (up-cast and down-cast) separated by 6 hours (half the semi-diurnal tidal period) to highlight station displacements. This sampling interval optimally captures phase reversals of internal tides, facilitating observation of isopycnal vertical displacements (Abyssal recipes II: energetics of tidal and wind mixing - ScienceDirect).
How internal tides propagate and how their vertical displacements can be observed through repeated density profiles (Internal Tide Generation in the Deep Ocean | Semantic Scholar).
These internal tides can contribute to mixing in the ocean, leading to the formation of step-like structures (Abyssal recipes II: energetics of tidal and wind mixing - ScienceDirect).

Figure RC1.4 illustrates two key aspects:

1) The observed isopycnal displacements at 6-hour intervals result from internal wave propagation, specifically through vertical advection (the advective component of wave motion)."

2) The step-like structures, consisting of vertically stacked homogeneous layers, represent clear signatures of mixing. These features result from wave-induced mixing during propagation. Notably, their spatial distribution is heterogeneous across stations, reflecting the localized nature of the mixing processes.

[Figure]

Figure RC1.4: Density profiles ($\sigma_\theta$, kg m$^{-3}$) from CTD-O$_2$ measurements during the AMAZOMIX 2021 cruise along transects: (a) $A^a$, (b) $A^b$, and (c) E. For long stations ($A^b_{s_2}$, $A^b_o$, $A^a_s$, $A^a_{o_1}$, $A^a_{o_2}$, $A^a_{o_3}$, A$_{isw}$, E$_s$, and E$_o$), two density profiles recorded ~6 hrs apart (half the M$_2$ tidal period) are shown to highlight step-like structures and vertical isopycnal displacements along the transects. Colored lines represent stations on the slope (red) and open ocean (blue, sky-blue, cyan, and light-orange). The subpanel in panel b depicts a step-like structure, where $L_{\rho_c}$ represents the vertical extent of homogeneous regions and $\rho_c$ denotes the density structure. The subpanel in panel c illustrates vertical displacements ($\Delta d$) of density structures, with $\rho_{t_1}$ and $\rho_{t_2}$ representing density structures at times t$_1$ and t$_2$, respectively

252-253- Maybe plot these on a different scale than the others.
R: Thanks for your remarks
We have plotted these on a different scale, as shown below (Figure RC1.5):
"

[Figure]

Figure RC1.5: Station-averaged dissipation rate profiles ($\varepsilon$, in W kg$^{-1}$, logarithmic scale) from VMP measurements during the AMAZOMIX 2021 cruise along transects: (a)-(b) Aa, (c)-(d) Ab, and (e)-(f) E. Colored lines represent stations on the shelf (green, lime-green), slope (red), and open ocean (blue, sky-blue, cyan, and light-orange). Vertical dashed and solid black lines are included for comparison.

"

256 - diffusivity not mixing coeff
R: Thanks for your remarks
We have updated this subsection in lines 262-280 of the revised manuscript.

258 - "It is important to note that the XLD is typically deeper than the MLD at all stations" but then there is no further mention of XLD or MLD in ths subsection.
R: Thanks for your remarks
We have updated this subsection in lines 262-280 of the revised manuscript..

Fig 4a - what do the symbols mean? Also plotting the max epsilon seemds like it will produce widely varying results. Better to use a depth- and time-mean value over some depth or density ranges.
R: Thanks for your remarks
In this figure, colored circles and stars denote short and long CTD-O$_2$/LADCP stations, respectively (see caption in Figure 10 of the revised manuscript).

For our analysis, we have decided to use both depth-integrated and depth-maximum values derived from time-averaged dissipation profiles at each station. To isolate tide-induced mixing in the mid-water column, we exclude the surface mixing layer and bottom boundary layer from these calculations. We reported the figure below (Figure RC1.6):
"

Figure RC1.6: Depth-integrated (in mW kg$^{-1}$, logarithmic scale) and maximum values (in W kg$^{-1}$, logarithmic scale) of station-averaged dissipation rates ($\varepsilon$) from VMP measurements during the AMAZOMIX 2021 cruise. Solid black lines depict transects ($A^a$, $A^b$, and and E) along high tidal energy (HTE) and low tidal energy (LTE) paths. Data are from below the wind-influenced surface layer and above the friction-dominated bottom boundary layer. Colored circles and stars represent short and long stations, respectively. Small and large colored circles indicate depth-integrated and maximum values of $\varepsilon$, respectively, with ranges shown by the color bar. Similarly, small and large colored stars indicate depth-integrated and maximum values of $\varepsilon$, respectively, with ranges shown by the color bar. Stations are grouped into five areas: $A_s$ ($A^a_s$ and $A^b_{s_2}$), $A_o$ ($A^b_o$, $A^a_{o_1}$, $A^a_{o_2}$, and $A^a_{o_3}$), $A_{isw}$ ($A_{isw}$), $E_s$ ($E_s$), and $E_o$ ($E_o$). The five blue boxes indicate these defined areas. Subscripts denote locations: 's' for slope ($A_s$), 'o' for offshore ($A_o$, and $E_o$), and 'isw' for ISW regions ($A_{isw}$).
"

270 - Some stations are plotted in the Appendix and some in the main text. This means after reading this sentence, I have to go to 2 widely separated figures.
R: Thanks for your remarks
We have completely reorganized and removed some materials in the revised manuscript.

Fig A5 - With a 2-day record, the frequency resolution is 1/2 cycles per day. The individual semidiurnal constituents cannot be distinguished and neither can the individual diurnal constituents. A fit of 1 cpd and 2 cpd and 4 cpd is sufficient- my previous review comment was unclear on this point.

R: Thanks for this clarity.

305 - 3-5 tidal modes - are these vertical normal modes? If vertical modes, no info has been provided on the mdal decomposition.

R: Thanks for your remarks

This study did not perform a formal modal decomposition. Instead, we analyzed the vertical structure of baroclinic tidal velocities to identify characteristic mode patterns by examining zero-crossings and amplitude variations (Abyssal Mixing: Where It Is Not in: Journal of Physical Oceanography Volume 26 Issue 10 (1996) ; Internal tides in the ocean - Wunsch - 1975 - Reviews of Geophysics - Wiley Online Library).

The identification criteria were:

Mode 1: Single zero-crossing with opposing velocity directions above/below

Mode 2: Two zero-crossings with two directional reversals

Higher modes: Additional zero-crossings (though typically weaker and less observable)

Thus, modes were classified by their zero-crossing counts (Mode 1 = 1, Mode 2 = 2, etc.).

Fig 6- $S^2$" in m s-1?

R: Thanks for your remarks

We have revised it in figure 9 of the revised manuscript.

Fig 7 - arrange panels better.

R: Thanks for your remarks

We have thoroughly restructured the revised manuscript, including the removal of some content to improve clarity and focus.

366 and Fig 7- At S10, turbulence appears unrelated to Ri calculated from $N^2$ and $S^2$"

R: Thanks for your comments

In this figure, we have added the linear lines representing Richardson numbers (Ri) following the relation $N^2 = Ri \times S^2$, with Ri values of 0.25 and 1.".

We have thoroughly restructured the revised manuscript, including the removal of some content to improve clarity and focus.

Fig 8a - the entire slope appears critical. So up- or downward and offshore beams could be emanating from any location on the slope in this figure. Beams if present would be visible not in the mean velocity (because tides are oscillating) but in the variance. Dissipation appears largest in the upper 100 m where currents are strong. Possibly this is a front at S6 and S10.

R: Thanks for your comments

By comparing ray slopes with topographic slopes using bathymetry from both the NEMO-AMAZON36 model and GEBCO, we not find that the entire slope appears critical in our sensitivity tests (see Figures RC1.7 and RC1.8; with topographic steepness γ). However, these results might vary with different bathymetric products.

This figure 8a displayed total (rather than mean) along- and cross-shore velocities. Indeed, the strongest dissipation occurs in the upper 100 m, may be associated with strong currents and a frontal feature near stations S6 and S10.
In the revised manuscript (Figure 12), we have removed total velocities and focused exclusively on internal tide rays to facilitate comparison with dissipation rates. This approach helps explain the intense mixing hotspot at Aisw, which contrasts with values typically found in the open ocean..

[Figure]

Figure RC1.7: Example of sensitivity tests with different cross-sectional measurements of $N^2$ along the transect T1 $N^2$. colors are used to distinguish different cross-shore measurements of $N^2$ for corresponding stations on T1. Topography steepness (gamma = ray slope / topography slope) for T1 using measured $N^2$ of S10. Gamma is illustrated by the colored bar (named gamma S10).

[Figure]

Figure RC1.8: Sensitivity tests of M2 IT ray-tracing along the transects Aa, conducted by varying the location of the critical topography slope. The tests use mean buoyancy frequency squared ($N^2$, in $s^{-2}$) obtained from CTD-$O_2$ data (September 2021) and NEMO-Amazon36 model data (2012-2016). Dashed colored lines represent IT beams calculated for different seasons (April, August, October, and September)

and for varying locations of the critical topography slope. Grey areas indicate local topography. Panel also includes dissipation rate profiles ($\varepsilon$, in W kg$^{-1}$, shown as vertical colored bars on a logarithmic scale) from

413- I think there is little evidence showing elevated turbulence along ray paths. I can't see where interference/interaction between waves along these ray paths occurs.
R: Thanks for your comments
While not explicitly examined in this study, we propose wave-wave interference as the primary mechanism potentially explaining the enhanced hotspots at station $A_{isw}$. This hypothesis requires verification through future targeted studies.

553- "The most relevant finding of this study was the relative increase in mixing within the pycnocline layer, observed at S14 in the
open ocean, far from the IT generation sites." Agreed. This is a possible focus of the manuscript. Everything in the paper should support this statement. Everything that is unrelated to this statement can be removed.
R: Thanks for your agreement.
We have substantially revised the manuscript to focus exclusively on this central finding, removing all extraneous material to sharpen the study's narrative and conclusions.

561 - I don't understand? - "...with large-amplitude ISWs
exceeding 100 m clearly visible in satellite records..."
R: Thanks for your remarks
We have updated the text in lines 356-358 of the revised manuscript to clarify this point.